# Finite-Time Analysis of Decentralized Single-Timescale Actor-Critic

**Qijun Luo**                                                    *qijunluo@link.cuhk.edu.cn*
*School of Science and Engineering*
*Shenzhen Research Institute of Big Data (SRIBD)*
*The Chinese University of Hong Kong, Shenzhen*
*Shenzhen, China*

**Xiao Li**                                                       *lixiao@cuhk.edu.cn*
*School of Data Science*
*Shenzhen Institute of Artificial Intelligence and Robotics for Society (AIRS)*
*The Chinese University of Hong Kong, Shenzhen*
*Shenzhen, China*

**Reviewed on OpenReview:** *https://openreview.net/forum?id=KQRvOO8iW4*

## Abstract

Decentralized Actor-Critic (AC) algorithms have been widely utilized for multi-agent reinforcement learning (MARL) and have achieved remarkable success. Apart from its empirical success, the theoretical convergence property of decentralized AC algorithms is largely unexplored. Most of the existing finite-time convergence results are derived based on either double-loop update or two-timescale step sizes rule, and this is the case even for centralized AC algorithm under a single-agent setting. In practice, the *single-timescale* update is widely utilized, where actor and critic are updated in an alternating manner with step sizes being of the same order. In this work, we study a decentralized *single-timescale* AC algorithm. Theoretically, using linear approximation for value and reward estimation, we show that the algorithm has sample complexity of $\tilde{\mathcal{O}}(\varepsilon^{-2})$ under Markovian sampling, which matches the optimal complexity with a double-loop implementation (here, $\tilde{\mathcal{O}}$ hides a logarithmic term). When we reduce to the single-agent setting, our result yields new sample complexity for centralized AC using a single-timescale update scheme. The central to establishing our complexity results is *the hidden smoothness of the optimal critic variable* we revealed. We also provide a local action privacy-preserving version of our algorithm and its analysis. Finally, we conduct experiments to show the superiority of our algorithm over the existing decentralized AC algorithms.

## 1 Introduction

Multi-agent reinforcement learning (MARL) (Littman, 1994; Vinyals et al., 2019) has been successful in various models of multi-agent systems, such as robotics (Lillicrap et al., 2015), autonomous driving (Yu et al., 2019), Go (Silver et al., 2017), etc. MARL has been extensively explored in the past decades; see, e.g., (Lowe et al., 2017; Omidshafiei et al., 2017; Zhang et al., 2021; Son et al., 2019; Espeholt et al., 2018; Rashid et al., 2018). These works either focus on the setting where a central controller is available, or assuming a common reward function for all agents. Among the many cooperative MARL settings, the work (Zhang et al., 2018) proposed the fully decentralized MARL with networked agents. In this setting, each agent maintains a private heterogeneous reward function, and agents can only access local/neighboring information through communicating with its neighboring agents on the network. Then, the objective of all agents is to jointly maximize the average long-term reward through interacting with environment modeled by multi-agent Markov decision process (MDP). They proposed the decentralized Actor-Critic (AC) algorithm to solve this

MARL problem, and showed its impressive performance. However, the theoretical convergence properties of such class of decentralized AC algorithms are largely unexplored; see (Zhang et al., 2021) for a comprehensive survey. In this work, our goal is to establish the finite-time convergence results under this fully decentralized MARL setting. We first review some recent progresses on this line of research below.

**Related works and motivations.** The first fully decentralized AC algorithm with provable convergence guarantee was proposed by (Zhang et al., 2018), and they achieved asymptotic convergence results under two-timescale step sizes, which requires actor's step sizes to diminish in a faster scale than the critic's step sizes. The sample complexities of decentralized AC were established recently. In particular, (Chen et al., 2022) and (Hairi et al., 2022) independently proposed two communication efficient decentralized AC algorithms with optimal sample complexity of $\mathcal{O}(\varepsilon^{-2}\log(\varepsilon^{-1}))$ under Markovian sampling scheme. Nevertheless, their analysis are based on *double-loop* implementation, where each policy optimization step follows a nearly accurate critic optimization step (a.k.a. policy evaluation), i.e., solving the critic optimization subproblem to $\varepsilon$-accuracy. Such a double-loop scheme requires careful tuning of two additional hyper-parameters, which are the batch size and inner loop size. In particular, the batch size and inner loop size need to be of order $\mathcal{O}(\varepsilon^{-1})$ and $\mathcal{O}(\log(\varepsilon^{-1}))$ in order to achieve their sample complexity results, respectively. In practice, single-loop algorithmic framework is often utilized, where one updates the actor and critic in an alternating manner by performing any constant algorithmic iterations for both subproblems; see, e.g., (Schulman et al., 2017; Lowe et al., 2017; Lin et al., 2019; Zhang et al., 2020). The work (Zeng et al., 2022) proposed a new decentralized AC algorithm based on such a single-loop alternative update. However, they have to adopt *two-timescale* step sizes rule to ensure convergence, which requires actor's step sizes to diminish in a faster scale than the critic's step sizes. Due to the separation of the step sizes, the critic optimization subproblem is solved exactly when the number of iterations tends to $\infty$. Such a restriction on the step size will slow down the convergence speed of the algorithm. As a consequence, they only obtain sub-optimal sample complexity of $\mathcal{O}(\varepsilon^{-\frac{5}{2}})$. In practice, most algorithms are implemented with *single-timescale* step sizes rule, where the step sizes for the actor's and critic's updates are of the same order. Though there are some theoretical achievements for single-timescale update in other areas such as TDC (Wang et al., 2021) and bi-level optimization (Chen et al., 2021a), similar theoretical understanding under AC setting is largely unexplored.

Indeed, even when reducing to single-agent setting, the convergence property of single-timescale AC algorithm is not well established. The works (Fu et al., 2021; Guo et al., 2021) established the finite-time convergence result under a special single-timescale implementation, where they attained the sample complexity of $\mathcal{O}(\varepsilon^{-2})$. Their analysis is based on an algorithm where the critic optimization step is formulated as a least-square temporal difference (LSTD) at each iteration, which requires to sample the transition tuples for $\tilde{\mathcal{O}}(\varepsilon^{-1})$ times to form the data matrix in the LSTD subproblem. Then, they solve the LSTD subproblem in a closed-form fashion by inverting a matrix of large size. Later, (Chen et al., 2021a) obtained the same sample complexity using TD(0) update for critic variables under i.i.d. sampling. Their analysis highly relies on the assumption that the Jacobian of the stationary distribution is Lipschitz continuous, which is not justified in their work.

The above observations motivate us to ask the following question:

*Can we establish finite-time convergence result for decentralized AC algorithm with single-timescale step sizes rule?*[1]

**Main contributions.** By answering this question positively, we have the following contributions:

- We design a decentralized AC algorithm, which employs a *single-timescale* step sizes rule and adopts Markovian sampling scheme. The proposed algorithm allows communication between agents for every $K_c$ iterations with $K_c$ being any integer lies in $[1, \mathcal{O}(\varepsilon^{-\frac{1}{2}})]$, rather than communicating at each iteration as adopted by previous single-loop decentralized AC algorithms (Zeng et al., 2022; Zhang et al., 2018).

- Using linear approximation for value and reward estimation, we establish the *finite-time* convergence result for the proposed algorithm under standard assumptions. In particular, we show that the algorithm has a sample complexity of $\tilde{\mathcal{O}}(\varepsilon^{-2})$, which matches the optimal complexity up to a

---

[1]As convention in (Fu et al., 2021), when we use "single-timescale", it means we utilize a single-loop algorithmic framework with single-timescale step sizes rule.

logarithmic term. In addition, we show that the logarithmic term hidden in the "$\tilde{\mathcal{O}}$" can be removed under the i.i.d. sampling scheme. These convergence results are valid for all the above mentioned choices for $K_c$.

- To preserve privacy of local actions, we propose a variant of our algorithm which utilizes noisy local rewards for estimating global rewards. We show that such an algorithm will maintain the optimal sample complexity at the expense of communicating at each iteration.

Our key technical result is to reveal *the hidden smoothness of the optimal critic variable*, so that we can derive a sufficient descent on the averaged critic's optimal gap under the single-timescale update. Consequently, we can resort to the classic convergence analysis for alternating optimization algorithms to establish the approximate ascent property of the overall optimization process, which leads to the final sample complexity results. We also designed a Lyapunov function to analyze the descent of the objective function with a single-timescale update under the decentralized setting.

When we reduce to the non-decentralized case, i.e., the single-agent setting, our results yield *new* sample complexity guarantees for the classic centralized AC algorithm using a *single-timescale* update scheme.

**Discussion on a concurrent work**. We note that there is a concurrent work (Olshevsky & Gharesifard, 2023) which also analyzes the single-timescale AC algorithm and achieves similar complexity results. Their analysis is based on the small gain theorem, which is different from ours. These two analysis frameworks provide useful insights for the AC algorithm from different perspectives. (Olshevsky & Gharesifard, 2023) shows that the coupled expression on the errors of actor and critic can be fit into a non-linear small gain theorem framework, which bounds the actor's error by desired order. Our analysis reveals the hidden smoothness of the optimal critic variable so that approximate descent on the critic's objective can be achieved. In addition, (Olshevsky & Gharesifard, 2023) considers the single-agent setting while our analysis deals with the more general decentralized setting. Moreover, (Olshevsky & Gharesifard, 2023) analyzes the i.i.d. sampling scheme where the single agent is assumed to have access to the transition tuples from the stationary distribution and the discounted state-visitation distribution. By contrast, our setting considers the practical Markovian sampling scheme, where the transition tuples are from the trajectory generated during the update of the agents.

## 2 Preliminary

In this section, we introduce the problem formulation and the policy gradient theorem, which serves as the preliminary for the analyzed decentralzed AC algorithm.

Suppose there are multiple agents aiming to independently optimize a common global objective, and each agent can communicate with its neighbors through a network. To model the topology, we define the graph as $\mathcal{G} = (\mathcal{N}, \mathcal{E})$, where $\mathcal{N}$ is the set of nodes with $|\mathcal{N}| = N$ and $\mathcal{E}$ is the set of edges with $|\mathcal{E}| = E$. In the graph, each node represents an agent, and each edge represents a communication link. The interaction between agents follows the networked multi-agent MDP.

### 2.1 Markov decision process

A networked multi-agent MDP is defined by a tuple $(\mathcal{G}, \mathcal{S}, \{\mathcal{A}^i\}_{i \in [N]}, \mathcal{P}, \{r^i\}_{i \in [N]}, \gamma)$. $\mathcal{G}$ denotes the communication topology (the graph), $\mathcal{S}$ is the finite state space observed by all agents, $\mathcal{A}^i$ represents the finite action space of agent $i$. Let $\mathcal{A} := \mathcal{A}^1 \times \cdots \times \mathcal{A}^N$ denote the joint action space and $\mathcal{P}(s'|s, a) : \mathcal{S} \times \mathcal{A} \times \mathcal{S} \to [0, 1]$ denote the transition probability from any state $s \in \mathcal{S}$ to any state $s' \in \mathcal{S}$ for any joint action $a \in \mathcal{A}$. $r^i : \mathcal{S} \times \mathcal{A} \to \mathbb{R}$ is the local reward function that determines the reward received by agent $i$ given transition $(s, a)$; $\gamma \in [0, 1]$ is the discount factor.

For simplicity, we will use $a := [a^1, \cdots, a^N]$ to denote the joint action, and $\theta \in \mathbb{R}^{Nd_\theta}$ to denote concatenation of all actor's joint parameters of all actors, with $\theta^i \in \mathbb{R}^{d_\theta}$. Here, without loss of generality, we assume that every agent has the same number of parameters for notation brevity. The MDP goes as follows: For a given state $s$, each agent make its decision $a^i$ based on its policy $a^i \sim \pi_{\theta^i}(\cdot|s)$. The state transits to the next state $s'$ based on the joint action of all the agents: $s' \sim \mathcal{P}(\cdot|s, a)$. Then, each agent will receive its own reward

$r^i(s,a)$. For the notation brevity, we assume that the reward function mapping is deterministic and does not depend on the next state without loss of generality. The stationary distribution induced by the policy $\pi_\theta$ and the transition kernel is denoted by $\mu_{\pi_\theta}(s)$.

Our objective is to find a set of policies that maximize the accumulated discounted mean reward received by agents

$$\theta^* = \arg\max_\theta J(\theta) := \mathbb{E}\left[\sum_{k=0}^\infty \gamma^k \bar{r}(s_k, a_k)\right]. \tag{1}$$

Here, $k$ represents the time step. $\bar{r}(s_k, a_k) := \frac{1}{N}\sum_{i=1}^N r^i(s_k, a_k)$ is the mean reward among agents at time step $k$. The randomness of the expectation comes from the initial state distribution $\mu_0(s)$, the transition kernel $\mathcal{P}$, and the stochastic policy $\pi_{\theta^i}(\cdot|s)$.

### 2.2 Policy gradient Theorem

Under the discounted reward setting, the global state-value function, action-value function, and advantage function for policy set $\theta$, state $s$, and action $a$, are defined as

$$V_{\pi_\theta}(s) := \mathbb{E}\left[\sum_{k=0}^\infty \gamma^k \bar{r}(s_k, a_k)|s_0 = s\right] \tag{2}$$

$$Q_{\pi_\theta}(s,a) := \mathbb{E}\left[\sum_{k=0}^\infty \gamma^k \bar{r}(s_k, a_k)|s_0 = s, a_0 = a\right]$$

$$A_{\pi_\theta}(s,a) := Q_{\pi_\theta}(s,a) - V_{\pi_\theta}(s).$$

To maximize the objective function defined in (1), the policy gradient (Sutton et al., 2000) can be computed as follow

$$\nabla_\theta J(\theta) = \mathbb{E}_{s\sim d_{\pi_\theta}, a\sim\pi_\theta}\left[\frac{1}{1-\gamma}A_{\pi_\theta}(s,a)\psi_{\pi_\theta}(s,a)\right],$$

where $d_{\pi_\theta}(s) := (1-\gamma)\sum_{k=0}^\infty \gamma^k \mathbb{P}(s_k = s)$ is the discounted state visitation distribution under policy $\pi_\theta$, and $\psi_{\pi_\theta}(s,a) := \nabla \log \pi_\theta(s,a)$ is the score function.

Following the derivation of (Zhang et al., 2018), the policy gradient for each agent under discounted reward setting can be expressed as

$$\nabla_{\theta^i} J(\theta) = \mathbb{E}_{s\sim d_{\pi_\theta}, a\sim\pi_\theta}\left[\frac{1}{1-\gamma}A_{\pi_\theta}(s,a)\psi_{\pi_{\theta^i}}(s,a^i)\right]. \tag{3}$$

## 3 Algorithms

### 3.1 Decentralized single-timescale Actor-Critic

We introduce the decentralized single-timescale AC algorithm; see Algorithm 1. In the remaining parts of this section, we will explain the updates in the algorithm in details.

In fully-decentralized MARL, each agent can only observe its local reward and action, while trying to maximize the global reward (mean reward) defined in (1). The decentralized AC algorithm solves the problem by updating actor and critic variables alternatively on an online trajectory. Specifically, we have $N$ pairs of actor and critic. In order to maximize $J(\theta)$, each critic tries to estimate the *global* state-value function $V_{\pi_\theta}(s)$ defined in (2). Then, each actor updates its policy parameter based on approximated policy gradient. We now provide more details about the algorithm.

**Critics' update.** We will use $\omega^i \in \mathbb{R}^{d_\omega}$ to denote the $i_{th}$ critic's parameter and $\bar{\omega} := \frac{1}{N}\sum_{i=1}^N \omega^i$ to represent the averaged parameter of critic. Each critic approximates the global value function as $V_{\pi_\theta}(s) \approx \hat{V}_{\omega^i}(s)$.

---

**Algorithm 1:** Decentralized single-timescale AC (reward estimator version)

---

1: **Initialize:** Actor parameter $\theta_0$, critic parameter $\omega_0$, reward estimator parameter $\lambda_0$, initial state $s_0$.
2: **for** $k = 0, \cdots, K - 1$ **do**
3:  **Option 1: i.i.d. sampling:**
4:  $s_k \sim \mu_{\theta_k}(s), a_k \sim \pi_{\theta_k}(\cdot|s_k), s_{k+1} \sim \mathcal{P}(\cdot|s_k, a_k)$.
5:  **Option 2: Markovian sampling:**
6:  $a_k \sim \pi_{\theta_k}(\cdot|s_k), s_{k+1} \sim \mathcal{P}(\cdot|s_k, a_k)$.
7:
8:  **Periodical consensus:** Compute $\tilde{\omega}_k^i$ and $\tilde{\lambda}_k^i$ by (4) and (7).
9:
10:  **for** $i = 0, \cdots, N$ **in parallel do**
11:    **Reward estimator update:** update $\lambda_{k+1}^i$ by (8).
12:    **Critic update:** Update $\omega_{k+1}^i$ by (5).
13:    **Actor update:** Update $\theta_{k+1}^i$ by (6).
14:  **end for**
15: **end for**

---

The critic's approximation error can be categorized into two parts, namely, the consensus error $\frac{1}{N}\sum_{i=1}^{N}\|\omega^i - \bar{\omega}\|$, which measures how close the critics' parameters are; and the approximation error $\|\bar{\omega} - \omega^*(\theta)\|$, which measures the approximation quality of averaged critic.

In order for critics to reach consensus, each critic exchanges its parameters with neighbors and perform the following update

$$\tilde{\omega}_k^i = \begin{cases} \sum_{j=1}^{N} W^{ij}\omega_k^j & \text{if } k \bmod K_c = 0 \\ \omega_k^i & \text{otherwise.} \end{cases} \tag{4}$$

Here, $K_c$ denotes the consensus frequency. The communication matrix $W \in \mathbb{R}^{n \times n}$ is usually determined artificially in practice and can be sparse, which means that the number of neighbors for each agent is much fewer than the total number of agents. Thus, the cost for each consensus step is usually much lower than a full synchronization over the network. The detailed requirements of matrix $W$ will be discussed in Assumption 5.

To reduce the approximation error, we will perform the local TD(0) update (Tsitsiklis & Van Roy, 1997) as

$$\omega_{k+1}^i = \Pi_{R_\omega}(\tilde{\omega}_k^i + \beta_k g_c^i(\xi_k, \omega_k^i)), \tag{5}$$

where $\xi := (s, a, s')$ represents a transition tuple, $g_c^i(\xi, \omega) := \delta^i(\xi, \omega)\nabla\hat{V}_\omega(s)$ is the update direction, $\delta^i(\xi, \omega) := r^i(s, a) + \gamma\hat{V}_\omega(s') - \hat{V}_\omega(s)$ is the local temporal difference error (TD-error). $\beta_k$ is the step size for critic at iteration $k$. $\Pi_{R_\omega}$ projects the parameter into a ball of radius of $R_\omega$ containing the optimal solution, which will be explained when discussing Assumptions 1 and 2.

**Actors' update.** We will use stochastic gradient ascent to update the policy's parameter, which is calculated based on policy gradient theorem in (3). The advantage function $A_{\pi_\theta}(s, a)$ can be estimated by

$$\delta(\xi, \theta) := \bar{r}(s, a) + \gamma V_{\pi_\theta}(s') - V_{\pi_\theta}(s),$$

with $a$ sampled from $\pi_\theta(\cdot|s)$. However, to preserve the privacy of each agents, the local reward cannot be shared to other agents under the fully decentralized setting. Thus, the averaged reward $\bar{r}(s_k, a_k)$ is not directly attainable. To this end, we adopt the strategy proposed in (Zhang et al., 2018) to approximate the averaged reward. In particular, each agent $i$ will have a local reward estimator with parameter $\lambda^i \in \mathbb{R}^{d_\lambda}$, which estimates the global averaged reward as $\bar{r}(s_k, a_k) \approx \hat{r}_{\lambda^i}(s_k, a_k)$.

Thus, the update of the $i_{th}$ actor is given by

$$\theta_{k+1}^i = \theta_k^i + \alpha_k\hat{\delta}(\xi_k, \omega_{k+1}^i, \lambda_{k+1}^i)\psi_{\pi_{\theta_k^i}}(s_k, a_k^i), \tag{6}$$

---

**Algorithm 2:** Decentralized single-timescale AC (noisy reward version)

---

1: **Initialize:** Actor parameter $\theta_0$, critic parameter $\omega_0$, initial state $s_0$.
2: **for** $k = 0, \cdots, K-1$ **do**
3:     **Option 1: i.i.d. sampling:**
4:     $s_k \sim \mu_{\theta_k}(s), a_k \sim \pi_{\theta_k}(\cdot|s_k), s_{k+1} \sim \mathcal{P}(\cdot|s_k, a_k)$.
5:     **Option 2: Markovian sampling:**
6:     $a_k \sim \pi_{\theta_k}(\cdot|s_k), s_{k+1} \sim \mathcal{P}(\cdot|s_k, a_k)$.
7:
8:     **Periodical consensus:** Compute $\tilde{\omega}_k^i$ by (4).
9:
10:     **for** $i = 0, \cdots, N$ **in parallel do**
11:        **Global reward estimation:** estimate $\bar{r}_k(s_k, a_k)$ by (9).
12:        **Critic update:** Update $\omega_{k+1}^i$ by (5).
13:        **Actor update:** Update $\theta_{k+1}^i$ by (10).
14:     **end for**
15: **end for**

---

where $\hat{\delta}(\xi, \omega, \lambda) := \hat{r}_\lambda(s, a) + \gamma \hat{V}_\omega(s') - \hat{V}_\omega(s)$ is the approximated advantage function. $\alpha_k$ is the step size for actor's update at iteration $k$.

**Reward estimators' update.** Similar to critic, each reward estimator's approximation error can be decomposed into consensus error and the approximation error.

For each local reward estimator, we perform the consensus step to minimize the consensus error as

$$\tilde{\lambda}_k^i = \begin{cases} \sum_{j=1}^N W^{ij} \lambda_k^j & \text{if } k \mod K_c = 0 \\ \lambda_k^i & \text{otherwise.} \end{cases} \tag{7}$$

To reduce the approximation error, we perform a local update of stochastic gradient descent.

$$\lambda_{k+1}^i = \Pi_{R_\lambda}(\tilde{\lambda}_k^i + \eta_k g_r^i(\xi_k, \lambda_k^i)), \tag{8}$$

where $g_r^i(\xi, \lambda) := (r^i(s, a) - \hat{r}_\lambda(s, a))\nabla \hat{r}_\lambda(s, a)$ is the update direction. $\eta_k$ is the step size for reward estimator at iteration $k$. Note the calculation of $g_r^i(\xi, \lambda)$ does not depend on the next state $s'$; we use $\xi$ in (8) just for notation brevity. Similar to critic's update, $\Pi_{R_\lambda}$ projects the parameter into a ball of radius of $R_\lambda$ containing the optimal solution.

In our Algorithm 1, we will use the same order for $\alpha_k$, $\beta_k$, and $\eta_k$ and hence, our algorithm is in *single-timescale*.

**Linear approximation for analysis.** In our analysis, we will use linear approximation for both critic and reward estimator variables, i.e. $\hat{V}_\omega(s) := \phi(s)^T \omega$; $\hat{r}_\lambda(s, a) := \varphi(s, a)^T \lambda$, where $\phi(s) : \mathcal{S} \to \mathbb{R}^{d_\omega}$ and $\varphi(s, a) : \mathcal{S} \times \mathcal{A} \to \mathbb{R}^{d_\lambda}$ are two feature mappings, whose property will be specified in the discussion of Assumption 1.

**Remarks on sampling scheme.** Acquiring unbiased stochastic gradients for critic and actor variables requires sampling from $\mu_{\pi_\theta}$ and $d_{\pi_\theta}$, respectively. However, in practical implementations, states are usually collected from an online trajectory (Markovian sampling), whose distribution is generally different from $\mu_{\pi_\theta}$ and $d_{\pi_\theta}$. Such a distribution mismatch will inevitably cause biases during the update of critic and actor variables. One has to bound the corresponding error terms when analyzing the algorithm.

### 3.2 Variant for preserving local action

Note that in Algorithm 1, the reward estimators need the knowledge of joint actions in order to estimate the global rewards. Inspired by (Chen et al., 2022), we further propose a variant of Algorithm 1 to preserve the privacy of local actions. It estimates the global rewards by communicating noisy local rewards. As a trade-off, the approach requires $\mathcal{O}(\log(\varepsilon^{-1}))$ communication rounds for each iteration; see Algorithm 2.

Let $r_k^i$ represents $r_k^i(s_k, a_k)$ for brevity. The reward estimation process goes as follow: for each agent $i$, we first produce a noisy local reward $\tilde{r}_k^i = r_k^i(1+z)$, with $z \sim \mathcal{N}(0, \sigma^2)$. Thus, the noise level is controlled by the variance $\sigma^2$, which is chosen artificially. When the noise level $\sigma^2$ increases, the local reward's privacy will be strengthen. In the meantime, the variance of the estimated global reward will increase. To estimate the global reward, each agent $i$ first initialize the estimation as $\tilde{r}_{k,0}^i = \tilde{r}_k^i$. Then, each agent $i$ perform the following consensus step for $K_r$ times, i.e.

$$\tilde{r}_{k,l+1}^i = \sum_{j=1}^N W^{ij} \tilde{r}_{k,l}^i, \quad l = 0, 1, \cdots, K_r - 1. \tag{9}$$

The reward $\tilde{r}_{k,K_r}^i$ will be used for estimating the global reward for agent $i$ at $k_{th}$ iteration. As we will see, the error $|\tilde{r}_{t,l+1}^i - \frac{1}{N}\sum_{i=1}^N \tilde{r}_k^i|$ will converge to 0 linearly. Hence, to reduce the error to $\varepsilon$, we need $K_r = \mathcal{O}(\log(\varepsilon^{-1}))$ rounds of communications for each iteration. Based on the estimated global reward, the $i_{th}$ actor's update is given by

$$\theta_{k+1}^i = \theta_k^i + \alpha_k(\tilde{r}_{k,K_r}^i + \gamma \hat{V}_{\omega^i}(s') - \hat{V}_{\omega^i}(s))\psi_{\pi_{\theta_k^i}}(s_k, a_k^i). \tag{10}$$

## 4 Main results

In this section, we first introduce the technical assumptions used for our analysis, which are standard in the literature. Then, we present the convergence results for both actor and critic variables.

### 4.1 Assumptions

**Assumption 1** (boundedness of rewards and feature vectors)**.** *The local rewards are uniformly bounded, i.e., there exists a positive constant $r_{\max}$ such that for all feasible $(s,a)$ and $i \in [N]$, we have $|r^i(s,a)| \leq r_{\max}$. The norm of feature vectors are bounded such that for all $s \in \mathcal{S}$, $a \in \mathcal{A}$, $\|\phi(s)\| \leq 1, \|\varphi(s,a)\| \leq 1$.*[2]

Assumption 1 is standard and commonly adopted; see, e.g., (Bhandari et al., 2018; Xu et al., 2020; Zeng et al., 2022; Shen et al., 2020; Qiu et al., 2019). This assumption can be achieved via normalizing the feature vectors.

**Assumption 2** (sufficient exploration)**.** *There exists two positive constants $\lambda_\phi, \lambda_\varphi$ such that for all policy $\pi_\theta$, the following two matrices are negative definite*

$$A_{\theta,\phi} := \mathbb{E}_{s \sim \mu_\theta(s)}[\phi(s)(\gamma\phi(s')^T - \phi(s)^T)]$$
$$A_{\theta,\varphi} := \mathbb{E}_{s \sim \mu_\theta(s), a \sim \pi_\theta(\cdot|s)}[-\varphi(s,a)\varphi(s,a)^T],$$

*with $\lambda_{\max}(A_{\theta,\phi}) \leq \lambda_\phi, \lambda_{\max}(A_{\theta,\varphi}) \leq \lambda_\varphi$, where $\lambda_{\max}(\cdot)$ represents the largest eigenvalue.*

The Assumption 2 characterizes a strong convexity-like property of critic and reward estimator's objective function, and thereby ensures sufficient decrease of the estimation error for each update. It will be satisfied when $\inf_{\theta,s,a} \pi_\theta(a|s) \geq c$ for all policy $\pi_\theta, s \in \mathcal{S}, a \in \mathcal{A}$ with $c$ being positive. Thus, it can be understood as an exploration assumption on policy $\pi_\theta$. (see Proposition 3.1 of (Olshevsky & Gharesifard, 2023) for more detail). This assumption is widely seen in analysis of AC algorithms; see, e.g. (Shen et al., 2020; Xu & Liang, 2021; Zeng et al., 2022). Together with Assumption 1, we can show that $\|\omega^*(\theta)\| \leq R_\omega := \frac{r_{\max}}{\lambda_\phi}$, $\|\lambda^*(\theta)\| \leq R_\lambda := \frac{r_{\max}}{\lambda_\varphi}$, which justifies the projection step. In practice, one can estimate $R_\omega$ and $R_\lambda$ online; see Section 8.2 of (Bhandari et al., 2018) for one approach. We provide more details for the projection in Appendix C.

**Assumption 3** (Lipschitz properties of policy)**.** *There exists constants $C_\psi, L_\psi, L_\pi$ such that for all policy parameter $\theta, \theta'$, $s \in \mathcal{S}$ and $a \in \mathcal{A}$, we have (1). $|\pi_\theta(a|s) - \pi_{\theta'}(a|s)| \leq L_\pi\|\theta - \theta'\|$; (2). $\|\psi_\theta(s,a) - \psi_{\theta'}(s,a)\| \leq L_\psi\|\theta - \theta'\|$; (3). $\|\psi_\theta(s,a)\| \leq C_\psi$.*

---

[2]Through out the paper, we will use $\|\cdot\|$ to represent the Euclidean norm for vectors and Frobenius norm for matrices.

Assumption 3 is common for analyzing policy-based algorithms; see, e.g., (Xu et al., 2019; Wu et al., 2020; Hairi et al., 2022). The assumption implies the smoothness of objective function $J(\theta)$. It holds for policy classes such as tabular softmax policy (Agarwal et al., 2020), Gaussian policy (Doya, 2000), and Boltzmann policy (Konda & Borkar, 1999).

**Assumption 4** (mixing of Markov chain). *There exists constants $\kappa > 0$ and $\rho \in (0,1)$ such that*

$$\sup_{s \in \mathcal{S}} d_{TV}\ (\mathbb{P}(s_k \in \cdot | s_0 = s, \pi_\theta), \mu_\theta) \leq \kappa \rho^k, \ \forall k.$$

Assumption 4 is a standard assumption; see, e.g. (Bhandari et al., 2018; Wu et al., 2020; Xu et al., 2019). The assumption always holds for irreducible and aperiodic Markov chain. It ensures the geometric convergence of state to the stationary distribution.

**Assumption 5** (doubly stochastic weight matrix). *The communication matrix $W$ is doubly stochastic, i.e. each column/row sum up to 1. Moreover, the second largest singular value $\nu$ is smaller than 1.*

Assumption 5 is a common assumption in decentralized optimization and multi-agent reinforcement learning; see, e.g., (Sun et al., 2020; Chen et al., 2021b; 2022). It ensures the convergence of consensus error for critic and reward estimator variables.

### 4.2 Sample complexity for Algorithm 1

**Theorem 1.** *Suppose Assumptions 1-5 hold. Consider the update of Algorithm 1 under Markovian sampling. Let $\alpha_k = \frac{\bar{\alpha}}{\sqrt{K}}$ for some positive constant $\bar{\alpha}$, $\beta_k = \frac{C_9}{2\lambda_\phi}\alpha_k$, and $\eta_k = \frac{C_{10}}{2\lambda_\varphi}\alpha_k$ and $K_c \leq \mathcal{O}(K^{1/4})$, where $K$ is the total number of iterations. Then, we have*

$$\frac{1}{K}\sum_{k=1}^{K}\sum_{i=1}^{N} \mathbb{E}\left[\left\|\omega_k^i - \omega^*(\theta_k)\right\|^2\right] \leq \mathcal{O}\left(\frac{\log^2 K}{\sqrt{K}}\right)$$

$$\frac{1}{K}\sum_{k=1}^{K}\sum_{i=1}^{N} \mathbb{E}\left[\left\|\nabla_{\theta^i} J(\theta_k)\right\|^2\right] \leq \mathcal{O}\left(\frac{\log^2 K}{\sqrt{K}}\right) + \mathcal{O}\left(\varepsilon_{app} + \varepsilon_{sp}\right), \tag{11}$$

*where $C_9, C_{10}$ are positive constants defined in proof.*

The proof of Theorem 1 can be found in Appendix D.1. It establishes the iteration complexity of $\mathcal{O}(\log^2 K/\sqrt{K})$, or equivalently, sample complexity of $\tilde{\mathcal{O}}(\varepsilon^{-2})$ for Algorithm 1. Note that actors, critics, and reward estimators use the step size of the same order. The rate matches the state-of-the-art sample complexity of decentralized AC algorithms up to a logarithmic term, which are implemented in double-loop fashion (Hairi et al., 2022; Chen et al., 2022). The approximation error is defined as

$$\varepsilon_{app} := \max_{\theta, a} \mathbb{E}_{s \sim \mu_\theta}\left[\left|V_{\pi_\theta}(s) - \hat{V}_{\omega^*(\theta)}(s)\right|^2 + \left|\bar{r}(s,a) - \hat{r}_{\lambda^*(\theta)}(s,a)\right|^2\right]. \tag{12}$$

The error $\varepsilon_{app}$ captures the approximation power of critic and reward estimator. When using function approximation, such an error is inevitable. Similar terms also appear in the literature (see, e.g., (Xu et al., 2020; Agarwal et al., 2020; Qiu et al., 2019)). $\varepsilon_{app}$ becomes zero in tabular case. The error $\varepsilon_{sp}$ represents the mismatch between the discounted state visitation distribution $d_{\pi_\theta}$ and stationary distribution $\mu_{\pi_\theta}$. It is defined as

$$\varepsilon_{sp} := 4C_\theta^2 \left(\log_\rho \kappa^{-1} + \frac{1}{\rho}\right)^2 (1-\gamma)^2.$$

By policy gradient theorem (3), the states should be sampled from discounted state visitation distribution in order to attain unbiased estimation of policy gradient. Nevertheless, the state distribution converges to stationary distribution $\mu_{\pi_\theta}$ due to Markov chain's mixing, which inevitably introduces the sampling error $\varepsilon_{sp}$. Similar terms also appear in (Zeng et al., 2022; Shen et al., 2020). When $\gamma$ is close to 1, the error becomes small. This is because $d_{\pi_\theta}$ approaches to $\mu_{\pi_\theta}$ when $\gamma$ goes to 1. In the literature, some works assume that sampling from $d_{\pi_\theta}$ is permitted, thus eliminate this error; see, e.g., (Chen et al., 2021a).

**Complexity result under i.i.d. sampling.** Under the i.i.d. sampling scheme, state can be directly sampled from $\mu_{\pi_\theta}$ and $d_{\pi_\theta}$. In this case, the logarithmic term caused by the Markovian mixing time, and the error $\varepsilon_{sp}$ caused by the distribution mismatch, can be avoided. In this sense, one can attain the iteration complexity of $\mathcal{O}(1/\sqrt{K})$, or equivalently, sample complexity of $\mathcal{O}(\varepsilon^{-2})$.

### 4.3 Sample complexity for Algorithm 2

**Theorem 2.** *Suppose Assumptions 1-5 hold. Consider the update of Algorithm 2 under Markovian sampling. Let $\alpha_k = \frac{\bar{\alpha}}{\sqrt{K}}$ for some positive constant $\bar{\alpha}$ and $\beta_k = \frac{C_9}{2\lambda_\phi}\alpha_k$, $K_r = \log(K^{1/2})$, $K_c \leq \mathcal{O}(K^{1/4})$, where $K$ is the total number of iterations. Then, we have*

$$\frac{1}{K}\sum_{k=1}^{K}\sum_{i=1}^{N}\mathbb{E}\left[\|\omega_k^i - \omega^*(\theta_k)\|^2\right] \leq \mathcal{O}\left(\frac{\log^2 K}{\sqrt{K}}\right)$$

$$\frac{1}{K}\sum_{k=1}^{K}\sum_{i=1}^{N}\mathbb{E}\left[\|\nabla_{\theta^i}J(\theta_k)\|^2\right] \leq \mathcal{O}\left(\frac{\log^2 K}{\sqrt{K}}\right) + \mathcal{O}(\varepsilon_{app}^c + \varepsilon_{sp}), \tag{13}$$

*where the constants are defined in proof.*

The proof of Theorem 2 can be found in Appendix D.2. It establishes the sample complexity of $\tilde{\mathcal{O}}(\varepsilon^{-2})$ for Algorithm 2. The $\varepsilon_{app}^c$ captures the approximation error of the critic variables, which is defined as

$$\varepsilon_{app}^c := \max_\theta \mathbb{E}_{s\sim\mu_\theta}\left[\left|V_{\pi_\theta}(s) - \hat{V}_{\omega^*(\theta)}(s)\right|^2\right].$$

The Algorithm 2 preserves the privacy of local actions and requires less parameters than Algorithm 1 since there is no reward estimator. The cost is that it needs to communicate $\mathcal{O}(\log(\varepsilon^{-1}))$ times for each iteration.

### 4.4 Proof sketch

We present the main elements for the proof of Theorem 1, which helps in understanding the difference between classical two-timescale/double-loop analysis and our single-timescale analysis. The proof of Theorem 2 follows the similar framework.

Under Markovian sampling, it is possible to show the following inequality, which characterizes the ascent of the objective.

$$\mathbb{E}[J(\theta_{k+1})] - J(\theta_k) \geq \sum_{i=1}^{N}\left[\frac{\alpha_k}{2}\mathbb{E}\|\nabla_{\theta^i}J(\theta_k)\|^2 + \frac{\alpha_k}{2}\mathbb{E}\|g_a^i(\xi_k, \omega_{k+1}^i, \lambda_{k+1}^i)\|^2\right.$$
$$\left. -8C_\psi^2\alpha_k\mathbb{E}\|\omega^*(\theta_k) - \omega_{k+1}^i\|^2 - 4C_\psi^2\alpha_k\mathbb{E}\|\lambda^*(\theta_k) - \lambda_{k+1}^i\|^2\right]$$
$$-\mathcal{O}(\log^2(K)\alpha_k^2) - \mathcal{O}((\varepsilon_{app} + \varepsilon_{sp})\alpha_k). \tag{14}$$

To analyze the errors of critic $\|\omega^*(\theta_k) - \omega_{k+1}^i\|^2$ and reward estimator $\|\lambda^*(\theta_k) - \lambda_{k+1}^i\|^2$, the two-timescale analysis requires $\mathcal{O}(\alpha_k) < \min\{\mathcal{O}(\beta_k), \mathcal{O}(\eta_k)\}$ in order for these two errors to converge. The double-loop approach runs lower-level update for $\mathcal{O}(\log(\varepsilon^{-1}))$ times with batch size $\mathcal{O}(\varepsilon^{-1})$ to drive these errors below $\varepsilon$ and hence, they cannot allow inner loop size and bath size to be $\mathcal{O}(1)$ simultaneously. To obtain the convergence result for *single-timescale* update, the idea is to further upper bound these two lower-level errors by the quantity $\mathcal{O}(\alpha_k\mathbb{E}\|g_a^i(\xi_k, \omega_{k+1}^i, \lambda_{k+1}^i)\|^2)$ (through a series of derivations), and then eliminate these errors by the ascent term $\frac{\alpha_k}{2}\mathbb{E}\|g_a^i(\xi_k, \omega_{k+1}^i, \lambda_{k+1}^i)\|^2$.

We mainly focus on the analysis of critic's error through the proof sketch. The analysis for reward estimator's error follows similar procedure. We start by decomposing the error of critic as

$$\sum_{i=1}^{N}\|\omega_{k+1}^i - \omega^*(\theta_k)\|^2 = \sum_{i=1}^{N}(\|\omega_{k+1}^i - \bar{\omega}_{k+1}\|^2 + \|\bar{\omega}_{k+1} - \omega^*(\theta_k)\|^2). \tag{15}$$

The first term represents the consensus error, which can be bounded by the next lemma.

**Lemma 1.** *Suppose Assumptions 1 and 5 hold. Consider the sequence $\{\omega_k^i\}$ generated by Algorithm 1, then the following holds*

$$\|Q\boldsymbol{\omega}_{k+1}\| \leq \nu^{\frac{k}{K_c}-1}\|\boldsymbol{\omega}_0\| + 4\sqrt{N}C_\delta \sum_{t=0}^{k} \nu^{\frac{k-t}{K_c}-1}\beta_t,$$

*where $\boldsymbol{\omega}_k := [\omega_k^1, \cdots, \omega_k^N]^T$, $Q := I - \frac{1}{N}\mathbf{1}\mathbf{1}^T$, $\nu \in (0,1)$ is the second largest singular value of $W$.*

Based on Lemma 1 and follow the step size rule of Theorem 1, it is possible to show $\|Q\boldsymbol{\omega}_{k+1}\|^2 = \sum_{i=1}^{N}\|\omega_{k+1}^i - \bar{\omega}_{k+1}\|^2 = \mathcal{O}(K_c^2\beta_k^2)$. Let $K_c = \mathcal{O}(\beta_k^{-\frac{1}{2}})$, we have $\|Q\boldsymbol{\omega}_{k+1}\|^2 = \mathcal{O}(\beta_k)$, which maintains the optimal rate.

To analyze the second term in (15), we first construct the following Lyapunov function

$$\mathbb{V}_k := -J(\theta_k) + \|\bar{\omega}_k - \omega^*(\theta_k)\|^2 + \|\bar{\lambda}_k - \lambda^*(\theta_k)\|^2. \tag{16}$$

Then, it remains to derive an approximate descent property of the term $\|\bar{\omega}_k - \omega^*(\theta_k)\|^2$ in (16). Towards that end, our key step lies in establishing the *smoothness of the optimal critic variables* shown in the next lemma.

**Lemma 2** (Smoothness of optimal critic)**.** *Suppose Assumptions 1-3 hold, under the update of Algorithm 1, there exists a positive constant $L_{\omega,2}$ such that for any policy parameter $\theta_1, \theta_2$, it holds that*

$$\|\nabla\omega^*(\theta_1) - \nabla\omega^*(\theta_2)\| \leq L_{\omega,2}\|\theta_1 - \theta_2\|,$$

This smoothness property is essential for achieving our $\tilde{\mathcal{O}}(1/\sqrt{K})$ convergence rate.

To the best of our knowledge, the smoothness of $\omega^*(\theta)$ has not been justified in the literature. Equipped with Lemma 2, we are able to establish the following lemma.

**Lemma 3** (Error of critic)**.** *Under Assumptions 1-5, consider the update of Algorithm 1. Then, it holds that*

$$\mathbb{E}[\|\bar{\omega}_{k+1} - \omega^*(\theta_{k+1})\|^2] \leq (1 + C_9\alpha_k)\|\bar{\omega}_{k+1} - \omega^*(\theta_k)\|^2$$

$$+ \frac{\alpha_k}{4}\sum_{i=1}^{N}\|\mathbb{E}[g_a^i(\xi_k, \omega_{k+1}^i, \lambda_{k+1}^i)]\|^2 + \mathcal{O}(\alpha_k^2). \tag{17}$$

$$\mathbb{E}[\|\bar{\omega}_{k+1} - \omega^*(\theta_k)\|^2] \leq (1 - 2\lambda_\phi\beta_k)\|\bar{\omega}_k - \omega^*(\theta_k)\|^2$$

$$+ C_{K_1}\beta_k\beta_{k-Z_K} + C_{K_2}\alpha_{k-Z_K}\beta_k. \tag{18}$$

*Here, $Z_K := \min\{z \in \mathbb{N}^+ | \kappa\rho^{z-1} \leq \min\{\alpha_K, \beta_K, \eta_K\}\}$, $C_9, \lambda_\phi$ are constants specified in appendix, and $C_{K_1}$ and $C_{K_2}$ are of order $\mathcal{O}(\log(K))$ and $\mathcal{O}(\log^2(K))$ respectively.*

Plug (18) into (17), we can establish the approximate descent property of $\|\bar{\omega}_k - \omega^*(\theta_k)\|^2$ in (16):

$$\mathbb{E}[\|\bar{\omega}_{k+1} - \omega^*(\theta_{k+1})\|^2] \leq (1 + C_9\alpha_k)(1 - 2\lambda_\phi\beta_k)\|\bar{\omega}_k - \omega^*(\theta_k)\|^2$$

$$+ \frac{\alpha_k}{4}\sum_{i=1}^{N}\|\mathbb{E}[g_a^i(\xi_k, \omega_{k+1}^i, \lambda_{k+1}^i)]\|^2 + \mathcal{O}(C_{K_1}\beta_k\beta_{k-Z_K} + C_{K_2}\alpha_{k-Z_K}\beta_k). \tag{19}$$

Finally, plugging (14), (17), and (19) into (16) gives the ascent of the Lyapunov function, which leads to our convergence result through steps of standard arguments.

**Remarks on update step.** In Algorithms 1 and 2, the actor and critic update once for each iteration. This update scheme can be generalized to the case where actor and critic update arbitrary number of constant steps without affecting the order of the sample complexity. In particular, suppose that actor updates $C_a$ steps per iteration, and let $g_{a,k}^i$ be the actor's update direction at iteration $k$. The bounds (14) and (19) become

$$\mathbb{E}[J(\theta_{k+1})] - J(\theta_k) \geq \sum_{i=1}^{N}\left[\frac{\alpha_k}{2}\mathbb{E}\|\nabla_{\theta^i}J(\theta_k)\|^2 + \frac{\alpha_k}{2}\mathbb{E}\|g_{a,k}^i\|^2 - 8C_\psi^2\alpha_k\mathbb{E}\|\omega^*(\theta_k) - \omega_{k+1}^i\|^2\right.$$

$$\left. - 4C_\psi^2\alpha_k\mathbb{E}\|\lambda^*(\theta_k) - \lambda_{k+1}^i\|^2\right] - \mathcal{O}(C_a^2\log^2(K)\alpha_k^2) - \mathcal{O}((\varepsilon_{app} + \varepsilon_{sp})\alpha_k)$$

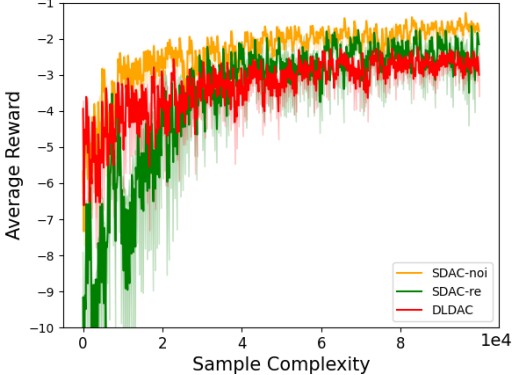 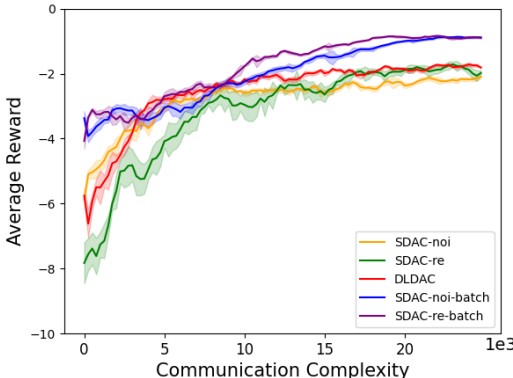

Figure 1: Averaged reward versus sample complexity and communication complexity. The vertical axis is the averaged reward over all the agents. The result is averaged over 10 Monte Carlo runs.

$$\mathbb{E}[\|\bar{\omega}_{k+1} - \omega^*(\theta_{k+1})\|^2] \leq (1 + C_9\alpha_k)(1 - 2\lambda_\phi\beta_k)\|\bar{\omega}_k - \omega^*(\theta_k)\|^2$$
$$+ \frac{\alpha_k}{4}\sum_{i=1}^{N}\|\mathbb{E}[g_{a,k}^i]\|^2 + \mathcal{O}(C_{K_1}\beta_k\beta_{k-Z_K} + C_a C_{K_2}\alpha_{k-Z_K}\beta_k),$$

where we replace the norm bound $\alpha_k\|g_a^i(\xi_k, \omega_{k+1}^i, \lambda_{k+1}^i)\| = \mathcal{O}(\alpha_k)$ with $\|g_{a,k}^i\|$ and apply Cauchy-Schwartz inequality: $\|g_{a,k}^i\| \leq C_a\|g_a^i(\xi_k, \omega_{k+1}^i, \lambda_{k+1}^i)\|$. When $C_a$ is a constant that is not related to $K$, these two bounds recovers (14) and (19). Hence, we can follow exactly the same proof procedure and obtain the $\tilde{\mathcal{O}}(\varepsilon^{-2})$ sample complexity result as before. When critic update $C_c > 1$ steps per iteration, the expected temporal difference error will decrease for each step by controlling step size, so that the bound in (19) still holds. Thus, updating critic for multiple steps will not affect the sample complexity.

### 4.5 Convergence of single-timescale decentralized NAC

The natural Actor-Critic (NAC) (Peters & Schaal, 2008) is a popular variant of AC algorithm, which enjoys the convergence to a global optimum (with compatible function approximation error) instead of a local stationary point. While our main focus is the convergence of the single-timescale AC algorithm, we find that the proof technique can be directly extended to establish the global convergence of single-timescale decentralized NAC. For reference, we design such an algorithm and provide its convergence result in Appendix E as a by-product of our single-timescale AC's analysis. To the best of our knowledge, this is the first convergence result of single-timescale NAC. However, our analysis only establishes a $\mathcal{O}(\varepsilon^{-6})$ rate for the algorithm. This result is sub-optimal compared with the existing best complexity of $\mathcal{O}(\varepsilon^{-3})$ (Chen et al., 2022), which is based on the double-loop implementation. The main reason for the sub-optimality is that in comparison with the double-loop update, the critic variables under the single-timescale update will inevitably converge slower due to the change of the actor's parameter in each iteration. Based on the classical NAC's analysis, the slower convergence of critic variables will result in a worse convergence rate of the optimality gap. Please refer to Appendix E for more discussions on the sub-optimality.

## 5 Numerical results

### 5.1 Experiment setting

We adopt the grounded communication environment proposed in (Mordatch & Abbeel, 2018). Our task consists of $N$ agents and the corresponding $N$ landmarks inhabited in a two-dimension world, where each agent can observe the relative position of other agents and landmarks. For every discrete time step, agents take actions to move along certain directions, and receive their rewards. Agents are rewarded based on the distance to their own landmark, and penalized if they collide with other agents. The objective is to maximize

the long-term averaged reward over all agents. Since we focus on decentralized setting, each agent shall not know the target landmark of others, i.e., the reward function of others. To exchange information, each agent is allowed to send their local information via a fixed communication link. Through all the experiments, the agent number $N$ is set to be 5, and the discount factor $\gamma$ is set to be 0.95.

## 5.2 Comparison with existing decentralized AC algorithms

In this section, we compare the proposed algorithm with existing decentralized AC algorithms under the cooperative MARL setting (Chen et al., 2022; Zeng et al., 2022) in terms of sample complexity and communication complexity. In the sequel, we refer Algorithm 1 as "SDAC-re" and Algorithm 2 as "SDAC-noi" (see Appendix 2). The algorithm proposed in (Chen et al., 2022) is referred as "DLDAC", which is based on double-loop implementation. The algorithm proposed in (Zeng et al., 2022) is denoted by "TDAC-re", which is based on two-timescale step size implementation. For comparison, we also implement a noisy reward version of "TDAC-re" and denote it by "TDAC-noi".

**Comparison to double-loop decentralized AC.** For "SDAC-re" and "SDAC-noi", we set $\alpha_k = 0.01(k+1)^{-0.5}$, $\beta_k = 0.1(k+1)^{-0.5}$, $\eta_k = 0.1(k+1)^{-0.5}$, $K_c = 5$, $\sigma = 0.5$, $K_r = 2$. For "DLDAC", we fix $T_c = 50$, $T'_c = 10$, $T' = 5$, $N_c = 10$, $N = 100$, $\sigma = 0.1$ [3], which is adopted by their paper (see comparisons under different hyper-parameters in Appendix A). We set $\alpha = 0.01, \beta = 0.1$ for "DLDAC" since we observe that larger step sizes will result in divergence. We have to mention that such a inner loop size $T_c = 50$ in "DLDAC" is not necessarily consistent with the theory of a double-loop algorithm, in which the loop size should be proportional to $\mathcal{O}(\varepsilon^{-1})$. The sample complexity and communication complexity results are shown in Figure 1. For the sample complexity, "SDAC-noi" enjoys a faster convergence compared with "DLDAC". In terms of communication complexity, "DLDAC" achieves better performance as it applies mini-batch technique and thereby requires less communication rounds when using the same amount of samples. Such a mini-batch approach can also be adopted to our proposed algorithms. Thus, we implement a mini-batch version of our proposed algorithms, which we refer as "SDAC-noi-batch" and "SDAC-re-batch", respectively. We set 10 as the batch size for actor, critic, and reward estimator. We can see that by applying mini-batch update, these two variants achieve significantly better communication complexity compared with "DLDAC". This is because our algorithm updates actor for more times compared with "DLDAC" under the same communication rounds.

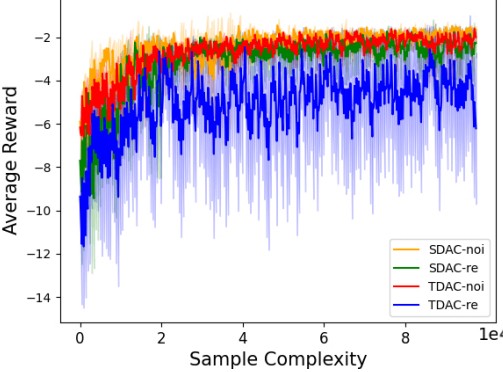

Figure 2: Comparison between the proposed algorithms and two-timescale decentralized AC algorithms (Zeng et al., 2022). The results are averaged over 10 Monte Carlo runs.

**Comparison with two-timescale decentralized AC.** We fix $K_c = 1$, $K_r = 5$ for this experiment. We set $\alpha_k = 0.01(k+1)^{-0.5}$, $\beta_k = 0.1(k+1)^{-0.5}$, and $\eta_k = 0.1(k+1)^{-0.5}$ for "SDAC-re" and "SDAC-noi"; we set $\alpha_k = 0.01(k+1)^{-0.6}$, $\beta_k = 0.1(k+1)^{-0.4}$, and $\eta_k = 0.1(k+1)^{-0.4}$ for "TDAC-re" and "TDAC-noi". The sample complexity is presented in Figure 2. We can observe that the convergence speed of "SDAC-noi" is

---

[3]Note that we adopt the notations in (Chen et al., 2022). Here, $T_c$ is the inner loop size, $T'_c$ is the communication number for each outer loop, $T'$ is the communication number for reward consensus, $N$ is the batch size for actor's update, and $N_c$ is the batch size for critic's update.

slightly better than that the two-timescale counterpart "TDAC-noi". In addition, when using reward estimator for the global reward estimation, we see that "SDAC-re" has much more stable convergence behavior than "TDAC-re", and achieves significantly higher rewards.

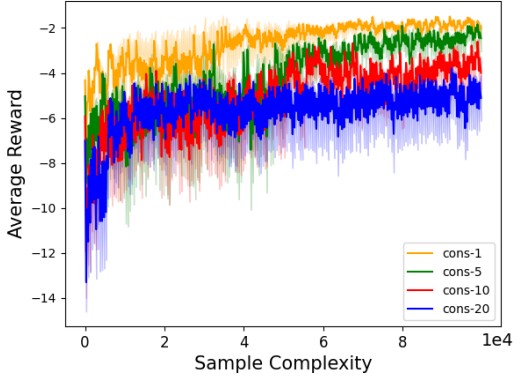 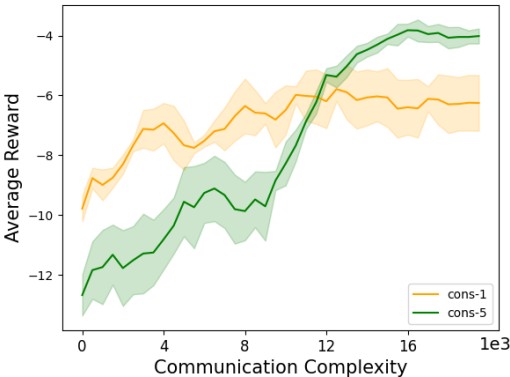

Figure 3: Ablation study on the consensus periods. The results are averaged over 10 Monte Carlo runs.

### 5.3 Ablation study on different choices of $K_c$

We compare the performance of "SDAC-noi" under different choices of consensus periods $K_c$. In particular, we set $\alpha_k = 0.01(k+1)^{-0.5}$, $\beta_k = 0.1(k+1)^{-0.5}$, $K_r = 2$, $\sigma = 0.5$ and examine the consensus periods $K_c$ of 1, 5, 10, and 20, respectively. The corresponding sample complexity and communication complexity results are summarized in Figure 3. Evidently, in terms of sample complexity, the convergence becomes slower and relatively unstable as the consensus period $K_c$ increases. Therefore, when the communication cost is low, choosing a small $K_c$ will yield a better performance. We also plot the communication complexity under the consensus periods of 1 and 5. We can see that the communication complexity of "cons-5" outperforms "cons-1" after $12 \times 10^3$ communications. Thus, when the communication cost is expensive and high averaged reward is required, one may use large $K_c$ and run the algorithm for a relatively large number of iterations.

## 6 Conclusion and future direction

In this paper, we studied the convergence of fully decentralized AC algorithm under practical single-timescale update. We showed that the algorithm will maintain the optimal sample complexity of $\tilde{\mathcal{O}}(\varepsilon^{-2})$ and is communication efficient. We also proposed a variant to preserve the privacy of local actions by communicating noisy rewards. Extensive simulation results demonstrate the superiority of our algorithms' empirical performance over existing decentralized AC algorithms. However, directly extending our single-timescale AC's analysis technique to single-timescale NAC will result in a sub-optimal sample complexity. We leave the study on improving the convergence rate and design a more efficient single-timescale NAC algorithm as promising future directions.

## Acknowledgement

The authors would like to thank the Action Editor and anonymous reviewers for their detailed and constructive comments, which have helped greatly to improve the quality and presentation of the manuscript.

X. Li was partially supported by the National Natural Science Foundation of China (NSFC) under Grant No. 12201534 and 72150002, by the Shenzhen Science and Technology Program under Grant No. RCBS20210609103708017, and by the Shenzhen Institute of Artificial Intelligence and Robotics for Society (AIRS) under Grant No. AC01202101108.

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

# Contents

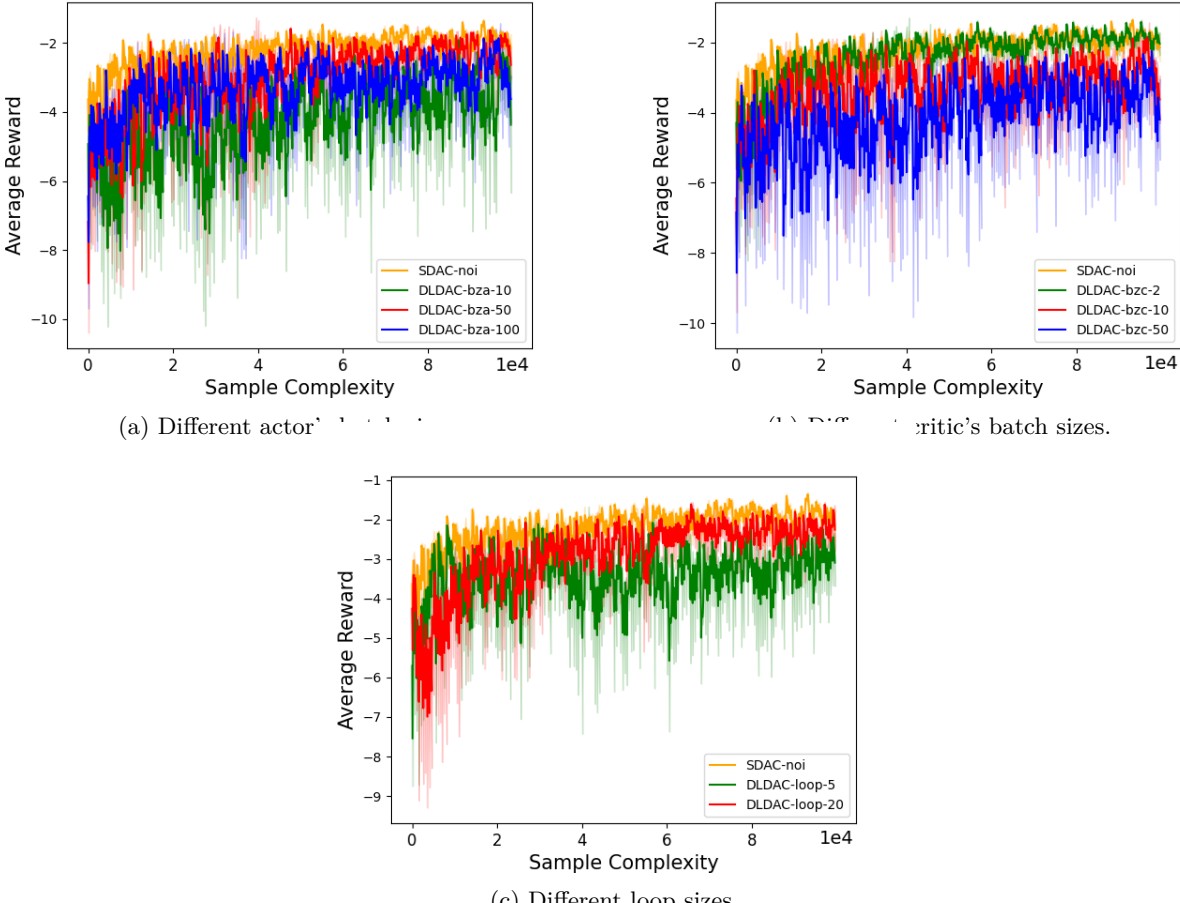

(a) Different actor's batch sizes.

(b) Different critic's batch sizes.

(c) Different loop sizes.

Figure 4: Comparison between the proposed algorithms and the double-loop decentralized AC algorithm that uses mini-batch update. The results are averaged over 10 Monte Carlo runs.

# A    Additional simulation results

In this section, we provide more experiments which compare the proposed algorithms with double-loop based decentralized AC algorithm under different batch sizes and inner loop sizes.

1. **Actor's batch size.** We fix $T_c = 50$, $T'_c = 10$, $N_c = 10$, [4] which is adopted by (Chen et al., 2022). We examine values of $N$ in $\{10, 50, 100\}$. The results are in Figure 4a. We observe that the best choice of actor's batch size $N$ is 50, and the proposed "SDAC-noi" converges faster than it in terms of sample complexity.

2. **Critic's batch size.** We fix $T_c = 50$, $T'_c = 10$, $N = 100$, which is adopted by (Chen et al., 2022). We examine values of $N_c$ in $\{2, 10, 50\}$. The results are shown in Figure 4b. As we can see, "DLDAC" with smaller critic's batch sizes can achieve better sample complexity, indicating that the variance of critic's update is relatively small and the mini-batch update is not needed for this task. Our proposed "SDAC-noi" achieves better convergence compared with the double-loop decentralized AC under different choices of $N_c$.

---

[4]Note that we adopt the notations in (Chen et al., 2022). Here, $T_c$ is the inner loop size, $T'_c$ is the communication number for each outer loop, $N$ is the batch size for actor's update, and $N_c$ is the batch size for critic's update.

3. **Inner loop size.** We fix $T_c' = 10$, $N = 100$, $N_c = 10$, which is adopted by (Chen et al., 2022). We examine values of $T_c$ in $\{5, 20\}$. The results are shown in Figure 4c. We can see that the proposed "SDAC-noi" enjoys a better convergence in terms of sample complexity.

## B  Auxiliary lemmas

In this section, we provide some auxiliary lemmas, which serves as the preliminary for the proof of main theorems and lemmas.

The following two lemmas present the Lipschitz properties of the objective function and value function.

**Lemma 4** ((Zhang et al., 2019), Lemma 3.2). *Suppose Assumption 3 holds, then there exists a positive constant $L$ such that for any policy parameter $\theta_1$ and $\theta_2$, we have $\|\nabla J(\theta_1) - \nabla J(\theta_2)\| \leq L\|\theta_1 - \theta_2\|$.*

**Lemma 5** ((Shen et al., 2020), Lemma 4). *Suppose Assumption 3 holds, for any policy parameter $\theta_1, \theta_2$ and $s \in \mathcal{S}$, there exits a positive constant $L_V$ such that*

$$\|\nabla V_{\pi_{\theta_1}}(s)\| \leq L_V$$
$$|V_{\pi_{\theta_1}}(s) - V_{\pi_{\theta_2}}(s)| \leq L_V\|\theta_1 - \theta_2\|.$$

The next lemma shows that the stationary distribution is Lipschitz continuous with respect to policy.

**Lemma 6** ((Wu et al., 2020), Lemma B.1). *For any policy parameter $\theta_1$ and $\theta_2$, it holds that*

$$d_{TV}(\mu_{\theta_1}, \mu_{\theta_2}) \leq |\mathcal{A}|L_\pi(\log_\rho \kappa^{-1} + (1-\rho)^{-1})\|\theta_1 - \theta_2\|$$
$$d_{TV}(\mu_{\theta_1} \otimes \pi_{\theta_1}, \mu_{\theta_2} \otimes \pi_{\theta_2}) \leq |\mathcal{A}|L_\pi(1 + \log_\rho \kappa^{-1} + (1-\rho)^{-1})\|\theta_1 - \theta_2\|$$
$$d_{TV}(\mu_{\theta_1} \otimes \pi_{\theta_1} \otimes \mathcal{P}, \mu_{\theta_2} \otimes \pi_{\theta_2} \otimes \mathcal{P}) \leq |\mathcal{A}|L_\pi(1 + \log_\rho \kappa^{-1} + (1-\rho)^{-1})\|\theta_1 - \theta_2\|.$$

*We will define $L_\mu := |\mathcal{A}|L_\pi(\log_\rho \kappa^{-1} + (1-\rho)^{-1})$ for the proof of main theorems and lemmas.*

The following lemma characterizes the geometric mixing of the Markov chain.

**Lemma 7** ((Shen et al., 2020), Lemma 1). *Suppose Assumption 4 holds, then there exists $\kappa > 0, \rho \in [0, 1]$ such that for any policy parameter $\theta$ we have*

$$\sup_{s_0 \in \mathcal{S}} d_{TV}(\mathbb{P}((s_k, a_k, s_{k+1}) \in \cdot | s_0, \pi_\theta), \mu_\theta \otimes \pi_\theta \otimes \mathcal{P}) \leq \kappa\rho^k,$$

*where $\mu_\theta$ is the stationary distribution induced by $\pi_\theta$ and transition kernel $\mathcal{P}(\cdot|s, a)$.*

The next lemma bounds the error of the discounted state-visitation distribution and the stationary distribution.

**Lemma 8** ((Shen et al., 2020), Lemma 2). *Suppose Assumption 4 holds, then for any policy parameter $\theta$, there exists $\kappa > 0, \rho \in [0, 1]$ such that*

$$d_{TV}(d_{\pi_\theta}, \mu_{\pi_\theta}) \leq 2(\log_\rho \kappa^{-1} + \frac{1}{1-\rho})(1-\gamma).$$

The following lemma bounds the total variation distance between state distribution under a fixed policy and that under an updating policy. The lemma is used for analyzing the sampling error.

**Lemma 9.** *Consider the Markov chain:*

$$s_{k-z} \xrightarrow{\theta_{k-z}} a_{k-z} \xrightarrow{\mathcal{P}} s_{k-z+1} \xrightarrow{\theta_{k-z+1}} a_{k-z+1} \cdots \xrightarrow{\theta_{k-1}} a_{k-1} \xrightarrow{\mathcal{P}} s_k \xrightarrow{\theta_k} a_k \xrightarrow{\mathcal{P}} s_{k+1}.$$

*Also consider the auxiliary Markov chain with fixed policy:*

$$s_{k-z} \xrightarrow{\theta_{k-z}} a_{k-z} \xrightarrow{\mathcal{P}} s_{k-z+1} \xrightarrow{\theta_{k-z}} \tilde{a}_{k-z+1} \cdots \xrightarrow{\theta_{k-z}} \tilde{a}_{k-1} \xrightarrow{\mathcal{P}} \tilde{s}_k \xrightarrow{\theta_{k-z}} \tilde{a}_k \xrightarrow{\mathcal{P}} \tilde{s}_{k+1}.$$

*Let $\xi_k := (s_k, a_k, s_{k+1})$ be sampled from chain 1, and $\tilde{\xi}_k := (\tilde{s}_k, \tilde{a}_k, \tilde{s}_{k+1})$ be sampled from chain 2. Then we have*

$$d_{TV}(\mathbb{P}(\xi_k \in \cdot | \theta_{k-z}, s_{k-z+1}), \mathbb{P}(\tilde{\xi}_k \in \cdot | \theta_{k-z}, s_{k-z+1})) \leq \frac{1}{2} \sum_{m=0}^{z-1} |\mathcal{A}|L_\pi \|\theta_{k-m} - \theta_{k-z}\|.$$

*Proof.*

$$d_{TV}(\mathbb{P}(\xi_k \in \cdot), \mathbb{P}(\tilde{\xi}_k \in \cdot))$$

$$= \frac{1}{2} \int_{s \in \mathcal{S}} \int_{s' \in \mathcal{S}} \sum_{a \in \mathcal{A}} |\mathbb{P}(s_k = ds, a_k = a, s_{k+1} = ds') - \mathbb{P}(\tilde{s}_k = ds, \tilde{a}_k = a, \tilde{s}_{k+1} = ds')|$$

$$= \frac{1}{2} \int_{s \in \mathcal{S}} \sum_{a \in \mathcal{A}} |\mathbb{P}(s_k = ds, a_k = a) - \mathbb{P}(\tilde{s}_k = ds, \tilde{a}_k = a)| \int_{s' \in \mathcal{S}} \mathbb{P}(s_{k+1} = ds'|s_k = ds, a_k = a)$$

$$= \frac{1}{2} \int_{s \in \mathcal{S}} \sum_{a \in \mathcal{A}} |\mathbb{P}(s_k = ds, a_k = a) - \mathbb{P}(\tilde{s}_k = ds, \tilde{a}_k = a)|$$

$$= \frac{1}{2} \int_{s \in \mathcal{S}} \sum_{a \in \mathcal{A}} |\mathbb{P}(s_k = ds)\pi_{\theta_k}(a|ds) - \mathbb{P}(\tilde{s}_k = ds)\pi_{\theta_{k-z}}(a|ds)|$$

$$\leq \frac{1}{2} \int_{s \in \mathcal{S}} \sum_{a \in \mathcal{A}} |\mathbb{P}(s_k = ds)\pi_{\theta_k}(a|ds) - \mathbb{P}(s_k = ds)\pi_{\theta_{k-z}}(a|ds)|$$

$$+ \frac{1}{2} \int_{s \in \mathcal{S}} \sum_{a \in \mathcal{A}} |\mathbb{P}(s_k = ds)\pi_{\theta_{k-z}}(a|ds) - \mathbb{P}(\tilde{s}_k = ds)\pi_{\theta_{k-z}}(a|ds)|$$

$$\leq \frac{1}{2} \int_{s \in \mathcal{S}} |\mathcal{A}| L_\pi \|\theta_k - \theta_{k-z}\| \mathbb{P}(s_k = ds)$$

$$+ \frac{1}{2} \int_{s \in \mathcal{S}} |\mathbb{P}(s_k = ds) - \mathbb{P}(\tilde{s}_k = ds)| \sum_{a \in \mathcal{A}} \pi_{\theta_{k-z}}(a|ds)$$

$$= \frac{1}{2} |\mathcal{A}| L_\pi \|\theta_k - \theta_{k-z}\| + d_{TV}(\mathbb{P}(s_k \in \cdot), \mathbb{P}(\tilde{s}_k \in \cdot)). \tag{20}$$

The second term can be bounded as

$$d_{TV}(\mathbb{P}(s_k \in \cdot), \mathbb{P}(\tilde{s}_k \in \cdot))$$

$$= \frac{1}{2} \int_{s' \in \mathcal{S}} |\mathbb{P}(s_k = ds) - \mathbb{P}(\tilde{s}_k = ds)|$$

$$= \frac{1}{2} \int_{s' \in \mathcal{S}} |\sum_{a \in \mathcal{A}} \int_{s \in \mathcal{S}} \mathbb{P}(s_{k-1} = ds, a_{k-1} = a, s_k = ds') - \mathbb{P}(\tilde{s}_{k-1} = ds, \tilde{a}_{k-1} = a, \tilde{s}_k = ds')|$$

$$\leq \frac{1}{2} \int_{s' \in \mathcal{S}} \sum_{a \in \mathcal{A}} \int_{s \in \mathcal{S}} |\mathbb{P}(s_{k-1} = ds, a_{k-1} = a, s_k = ds') - \mathbb{P}(\tilde{s}_{k-1} = ds, \tilde{a}_{k-1} = a, \tilde{s}_k = ds')|$$

$$= d_{TV}(\mathbb{P}(\xi_{k-1} \in \cdot), \mathbb{P}(\tilde{\xi}_{k-1} \in \cdot)). \tag{21}$$

Combined (20) and (21), we obtain

$$d_{TV}(\mathbb{P}(\xi_k \in \cdot), \mathbb{P}(\tilde{\xi}_k \in \cdot)) \leq d_{TV}(\mathbb{P}(\xi_{k-1} \in \cdot), \mathbb{P}(\tilde{\xi}_{k-1} \in \cdot)) + \frac{1}{2} |\mathcal{A}| L_\pi \|\theta_k - \theta_{k-z}\|.$$

Sum over $z - 1$ steps, we obtain

$$d_{TV}(\mathbb{P}(\xi_k \in \cdot), \mathbb{P}(\tilde{\xi}_k \in \cdot)) \leq d_{TV}(\mathbb{P}(\xi_{k-z} \in \cdot), \mathbb{P}(\tilde{\xi}_{k-z} \in \cdot)) + \frac{1}{2} \sum_{m=0}^{z-1} |\mathcal{A}| L_\pi \|\theta_{k-m} - \theta_{k-z}\|$$

$$= \frac{1}{2} \sum_{m=0}^{z-1} |\mathcal{A}| L_\pi \|\theta_{k-m} - \theta_{k-z}\|.$$

$\square$

Next, we present some mathematical facts that are useful in our analysis.

**Lemma 10** ((Chen et al., 2021b), Lemma F.3). *For a doubly stochastic matrix $W \in \mathbb{R}^{N \times N}$ and the difference matrix $Q := I - \frac{1}{N} \mathbf{1} \mathbf{1}^T$, it holds that for any matrix $H \in \mathbb{R}^{N \times N}$, $\|W^k H\| \leq \nu^k \|QH\|$, where $\nu$ is the second largest singular value of $W$.*

**Lemma 11** (descent lemma in high dimension). *Consider the mapping $F : \mathbb{R}^n \rightarrow \mathbb{R}^m$. If there exists a positive constant $L$ such that*

$$\|\nabla F(x) - \nabla F(y)\|_F \leq L\|x - y\|, \ \forall x, y \in dom(F), \tag{22}$$

*then the following holds*

$$\|F(y) - F(x) - \nabla F(x)(y - x)\| \leq \frac{L}{2}\sqrt{m}\|y - x\|^2.$$

*Proof.* Observe that (22) directly implies the smoothness of each entry $F_i$:

$$\|\nabla F_i(x) - \nabla F_i(y)\| \leq \|\nabla F(x) - \nabla F(y)\|_F \leq L\|x - y\|.$$

Define

$$z_i(x, y) := F_i(y) - F_i(x) - \nabla F_i(x)^T (y - x).$$

We have

$$\|F(y) - F(x) - \nabla F(x)(y - x)\| = \sqrt{\sum_{i=1}^m z_i(x, y)^2}$$

$$\leq \sqrt{m(\frac{L}{2}\|y - x\|^2)^2}$$

$$= \frac{L_1}{2}\sqrt{m}\|y - x\|^2,$$

where the inequality follows the descent lemma. $\qquad\square$

**Lemma 12** (Lipschitz property of multiplication). *Suppose $f(x)$ and $g(x)$ are two functions bounded by $C_f$ and $C_g$, and are $L_f$- and $L_g$-Lipschitz continuous, then $f(x)g(x)$ is $C_f L_g + C_g L_f$-Lipschitz continuous.*

*Proof.*

$$\|f(x_1)g(x_1) - f(x_2)g(x_2)\| = \|f(x_1)g(x_1) - f(x_1)g(x_2) + f(x_1)g(x_2) - f(x_2)g(x_2)\|$$

$$\leq \|f(x_1)\|\|g(x_1) - g(x_2)\| + \|f(x_1) - f(x_2)\|\|g(x_2)\|$$

$$\leq (C_f L_g + C_g L_f)\|x_1 - x_2\|.$$

$\qquad\square$

**Lemma 13** (invertible property of matrix). *If a square matrix $A$ satisfies $\lim_{t \to \infty} A^t = 0$, or equivalently, $|\lambda(A)| < 1$, then $I - A$ is invertible.*

*Proof.*

$$(I - A)\lim_{t \to \infty}\sum_{i=0}^t A^t = \lim_{t \to \infty}[\sum_{i=0}^t A^t - \sum_{i=1}^{t+1} A^t]$$

$$= I - \lim_{t \to \infty} A^{t+1}$$

$$= I$$

Since $I$ is invertible, by the rank inequality $\text{rank}(AB) \leq \min(\text{rank}(A), \text{rank}(B))$, $I - A$ and $\lim_{t \to \infty}\sum_{i=0}^t A^t$ will be full rank and thereby invertible. $\qquad\square$

## C   Supporting lemmas

Before proceeding to the analysis of critic variables, we justify the uniqueness of fix point for critic and reward estimator variables under the update (5) and (8), respectively. Define the following notations

$$
\begin{aligned}
A_{\theta,\phi} &:= \mathbb{E}[\phi(s)(\gamma\phi(s')^T - \phi(s)^T)], \qquad\qquad\qquad (23)\\
A_{\theta,\varphi} &:= \mathbb{E}[\varphi(s,a)\varphi(s,a)^T],\\
b_{\theta,\phi} &:= \mathbb{E}[\phi(s)\bar{r}(s,a)],\\
b_{\theta,\varphi} &:= \mathbb{E}[\varphi(s,a)\bar{r}(s,a)],
\end{aligned}
$$

with expectation taken from $s \sim \mu_\theta(s), a \sim \pi_\theta, s' \sim \mathcal{P}$. The optimal critic and reward estimator variables given policy $\theta$ will satisfy $A_{\theta,\phi}\omega^*(\theta) + b_{\theta,\phi} = 0; A_{\theta,\varphi}\lambda^*(\theta) + b_{\theta,\varphi} = 0$. By Assumption 2, $A_{\theta,\phi}$ and $A_{\theta,\varphi}$ are negative definite with largest eigenvalue $\lambda_\phi$ and $\lambda_\varphi$, which ensures the unique solution $\omega^*(\theta) = -A_{\theta,\phi}^{-1}b_{\theta,\phi}; \lambda^*(\theta) = -A_{\theta,\varphi}^{-1}b_{\theta,\varphi}$. Let $R_\omega := \frac{r_{\max}}{\lambda_\phi}, R_\lambda := \frac{r_{\max}}{\lambda_\varphi}$. Then the norm of optimal solutions will be bounded as $\|\omega^*(\theta)\| \leq R_\omega, \|\lambda^*(\theta)\| \leq R_\lambda$, which justifies the projection step of the Algorithm 1.   In practice, the knowledge of $\lambda_\phi$ and $\lambda_\varphi$ may not be available. One can estimate projection radius online using the methods proposed in Section 8.2 of (Bhandari et al., 2018).

We slightly abuse the notation by overwriting $V_{\pi_\theta}$ as $V_\theta$. To study the error of critic, we introduce the following notations (crf. $\xi := (s,a,s')$)

$$
\begin{aligned}
\delta^i(\xi,\theta) &:= r^i(s,a) + \gamma V_\theta(s') - V_\theta(s)\\
\delta(\xi,\theta) &:= \bar{r}(s,a) + \gamma V_\theta(s') - V_\theta(s)\\
\tilde{\delta}(\xi,\omega) &:= \bar{r}(s,a) + \gamma\phi(s')^T\omega - \phi(s)^T\omega\\
\hat{\delta}(\xi,\omega,\lambda) &:= \varphi(s,a)^T\lambda + \gamma\phi(s')^T\omega - \phi(s)^T\omega, \qquad\qquad (24)
\end{aligned}
$$

For the ease of expression, we further define

$$
\begin{aligned}
g_a^i(\xi,\omega,\lambda) &:= \hat{\delta}(\xi,\omega,\lambda)\psi_{\theta^i}(s,a^i),\\
g_c^i(\xi,\omega) &:= \delta^i(\xi,\omega)\phi(s),\\
\bar{g}_c(\xi,\omega) &:= \tilde{\delta}(\xi,\omega)\phi(s),\\
g_c(\theta,\omega) &:= \mathbb{E}_{\xi\sim\mu_\theta}[\bar{g}_c(\xi,\omega)]. \qquad\qquad\qquad\qquad (25)
\end{aligned}
$$

### C.1   Error of critic

The following lemmas and propositions serves as the preliminary for establishing the approximate descent property of the critic variables' optimal gap.

**Proposition 1** (Lipschitz continuity of $\omega^*(\theta)$ (Wu et al., 2020)). *Suppose Assumptions 1, 2, 3, and 4 hold, then there exists a positive constant $L_\omega$ such that for any $\theta_1, \theta_2 \in \mathbb{R}^{Nd_\theta}$, we have*

$$
\|\omega^*(\theta_1) - \omega^*(\theta_2)\| \leq L_\omega\|\theta_1 - \theta_2\|.
$$

**Lemma 14** (smoothness of stationary distribution). *For any $\theta, \theta' \in \mathbb{R}^d$, there exists a positive constant $L_{\mu,2}$ such that $\|\nabla\mu_\theta(s) - \nabla\mu_{\theta'}(s)\| \leq L_{\mu,2}\|\theta - \theta'\|$.*

The proof of this Lemma consists of two main steps: 1) Derive the expression of the gradient and 2) establish that the gradient is Lipschitz continuous. For the first part, we follow the main idea in (Baxter & Bartlett, 2001).

*Proof.* For a given policy $\pi_\theta$, we define the transition probability $P_\theta(s|s') := \sum_a \pi_\theta(a|s')P(s|s',a)$. By the Assumption 4, there exists a stationary distribution $\mu_\theta(s)$ which satisfies for all state $s$

$$\mu_\theta(s) = \sum_{s' \in \mathcal{S}} \mu_\theta(s') P_\theta(s|s') \tag{26}$$

Define the following notations

$$\mu_\theta := [\mu_\theta(s_1), \mu_\theta(s_2), \cdots, \mu_\theta(s_n)]^T \qquad \mathbb{R}^{|\mathcal{S}| \times 1}$$

$$P_\theta(s) := [P_\theta(s|s_1), P_\theta(s|s_2), \cdots, P_\theta(s|s_n)]^T \qquad \mathbb{R}^{|\mathcal{S}| \times 1}$$

$$P(\theta) := [P_\theta(s_1), P_\theta(s_2), \cdots, P_\theta(s_n)] \qquad \mathbb{R}^{|\mathcal{S}| \times |\mathcal{S}|}$$

$$\nabla \mu_\theta := [\nabla \mu_\theta(s_1), \nabla \mu_\theta(s_2), \cdots, \nabla \mu_\theta(s_n)] \qquad \mathbb{R}^{d_\theta \times |\mathcal{S}|}$$

$$\nabla P_\theta(s) := [\nabla P_\theta(s|s_1), \nabla P_\theta(s|s_2), \cdots, \nabla P_\theta(s|s_n)] \qquad \mathbb{R}^{d_\theta \times |\mathcal{S}|}$$

Upon taking derivative with respect to $\theta$ on both sides of (26), we have

$$\nabla \mu_\theta(s) = \sum_{s' \in \mathcal{S}} \nabla \mu_\theta(s') P_\theta(s|s') + \mu_\theta(s') \nabla_\theta P_\theta(s|s')$$
$$= \nabla \mu_\theta P_\theta(s) + \nabla P_\theta(s) \mu_\theta \tag{27}$$

(27) can be written in compact form as

$$\nabla \mu_\theta = \nabla \mu_\theta P(\theta) + [\nabla P_\theta(s_1) \mu_\theta, \cdots, \nabla P_\theta(s_n) \mu_\theta] \tag{28}$$

Therefore, we have

$$[\nabla P_\theta(s_1) \mu_\theta, \cdots, \nabla P_\theta(s_n) \mu_\theta] = \nabla \mu_\theta(I - P(\theta))$$
$$= \nabla \mu_\theta(I - (P(\theta) - e\mu_\theta^T)),$$

where the second inequality is due to $\nabla \mu_\theta e = \nabla(\mu_\theta e) = \nabla 1 = 0$.

We now show that $I - (P(\theta) - e\mu_\theta^T)$ is invertible. The first step is to show $\lim_{t \to \infty} (P(\theta) - e\mu_\theta^T)^t = 0$. Let $P, \mu$ represent $P(\theta), \mu_\theta$ for simplicity, we first show $(P - e\mu^T)^t = P^t - P^{t-1}e\mu^T$ by induction. Observe that when $t = 1$, this is trivially satisfied. Suppose the equality holds for $t = k$, then

$$(P - e\mu^T)^{k+1} = (P^k - P^{k-1}e\mu^T)P - (P^k - P^{k-1}e\mu^T)e\mu^T$$
$$= P^{k+1} - P^{k-1}e\mu^T - P^k e\mu^T + P^{k-1}(e\mu^T)^2$$
$$= P^{k+1} - P^k e\mu^T,$$

where the second equality is due to (26) such that $e\mu^T P = e\mu^T$ and the last equality is due to $\mu^T e = 1$.

Therefore, we have

$$\lim_{t \to \infty} (P(\theta) - e\mu_\theta^T)^t = \lim_{t \to \infty} (P(\theta)^t - P(\theta)^{t-1}e\mu_\theta^T) = e\mu_\theta^T - e\mu_\theta^T = 0,$$

which together with Lemma 13 justifies that $I - (P(\theta) - e\mu_\theta^T)$ is invertible. Thus, we have

$$\nabla \mu_\theta = (I - (P(\theta) - e\mu_\theta^T))^{-1} [\nabla P_\theta(s_1) \mu_\theta, \cdots, \nabla P_\theta(s_n) \mu_\theta]. \tag{29}$$

We will utilize Lemma 12 to prove the Lipschitz property of $\nabla \mu_\theta$. We first show the Lipschitz continuous of the first term. Let $A(\theta)$ to represent $I - (P(\theta) - e\mu_\theta^T)$, then we have

$$
\begin{aligned}
\|A(\theta_1) - A(\theta_2)\| &= \|P(\theta_1) - P(\theta_2) + e(\mu_{\theta_2} - \mu_{\theta_1})^T\| \\
&\leq \|P(\theta_1) - P(\theta_2)\| + \|e(\mu_{\theta_2} - \mu_{\theta_1})^T\| \\
&= \sqrt{\sum_{s,s'\in\mathcal{S}} |\sum_{a\in\mathcal{A}} (\pi_{\theta_1}(a|s') - \pi_{\theta_2}(a|s'))P(s|s',a)|^2} + \sqrt{|\mathcal{S}|}\|\mu_{\theta_2} - \mu_{\theta_1}\| \\
&\leq \sqrt{\sum_{s,s'\in\mathcal{S}} (\sum_{a\in\mathcal{A}} |(\pi_{\theta_1}(a|s') - \pi_{\theta_2}(a|s'))P(s|s',a)|)^2} + \sqrt{|\mathcal{S}|}\|\mu_{\theta_2} - \mu_{\theta_1}\| \\
&\leq \sqrt{\sum_{s'\in\mathcal{S}} |\mathcal{A}|^2 L_\pi^2 \|\theta_1 - \theta_2\|^2 \sum_{s\in\mathcal{S}} P(s|s',a)^2} + \sqrt{|\mathcal{S}|}L_\mu\|\theta_1 - \theta_2\| \\
&= \sqrt{|\mathcal{S}|}(|\mathcal{A}|L_\pi + L_\mu)\|\theta_1 - \theta_2\|.
\end{aligned}
$$

where the second inequality uses triangle inequality. The last inequality is due to Lipschitz continuous of the policy specified in Assumption 3, and Lipschitz continuous of $\mu_\theta$ implied by Lemma 5.

To see that $A^{-1}(\theta)$ is Lipschitz continuous and bounded, observe that

$$
\begin{aligned}
\|A^{-1}(\theta_1) - A^{-1}(\theta_2)\| &= \|A^{-1}(\theta_2)(A(\theta_2) - A(\theta_1))A^{-1}(\theta_1)\| \\
&\leq \|A^{-1}(\theta_2)\|\|A^{-1}(\theta_1)\|\|A(\theta_2) - A(\theta_1)\| \\
&\leq \sqrt{|\mathcal{S}|}(|\mathcal{A}|L_\pi + L_\mu)\|A^{-1}(\theta_2)\|\|A^{-1}(\theta_1)\|\|\theta_2 - \theta_1\|, \quad (30)
\end{aligned}
$$

where the first inequality uses Cauchy-Schwartz inequality, and the last inequality uses the Lipschitz continuous of $A(\theta)$ in (30). Since $\|A(\theta)\|$ is bounded, $\|A^{-1}(\theta)\|$ is also bounded (due to invertibility), which justifies that the first term in (29) is Lipschitz continuous and bounded.

We now consider the second term in (29). For any state $s$

$$
\begin{aligned}
\|\nabla P_{\theta_1}(s)\mu_{\theta_1} - \nabla P_{\theta_2}(s)\mu_{\theta_2}\| &= \|\nabla P_{\theta_1}(s)(\mu_{\theta_1} - \mu_{\theta_2}) + (\nabla P_{\theta_1}(s) - \nabla P_{\theta_2}(s))\mu_{\theta_2}\| \\
&\leq \|\nabla P_{\theta_1}(s)(\mu_{\theta_1} - \mu_{\theta_2})\| + \|(\nabla P_{\theta_1}(s) - \nabla P_{\theta_2}(s))\mu_{\theta_2}\| \\
&\leq \|\nabla P_{\theta_1}(s)\|\|\mu_{\theta_1} - \mu_{\theta_2}\| + \|\nabla P_{\theta_1}(s) - \nabla P_{\theta_2}(s)\|\|\mu_{\theta_2}\| \\
&\leq \sum_{s'\in\mathcal{S}}\sum_{a\in\mathcal{A}} \|\nabla \pi_{\theta_1}(a|s')P(s|s',a)\|L_\mu\|\theta_1 - \theta_2\| \\
&\quad + \sum_{s'\in\mathcal{S}}\sum_{a\in\mathcal{A}} \|(\nabla \pi_{\theta_1}(a|s') - \nabla \pi_{\theta_2}(a|s'))P(s|s',a)\| \\
&\leq |\mathcal{S}||\mathcal{A}|(C_\pi L_\mu + L_\pi)\|\theta_1 - \theta_2\|,
\end{aligned}
$$

which justifies the Lipschitz continuous of $\nabla P_\theta(s)\mu_\theta$. Define $B(\theta) := [\nabla P_\theta(s_1)\mu_\theta, \cdots, \nabla P_\theta(s_n)\mu_\theta]$, we have

$$
\|B(\theta_1) - B(\theta_2)\| \leq |\mathcal{S}|^{3/2}|\mathcal{A}|(C_\pi L_\mu + L_\pi)\|\theta_1 - \theta_2\|.
$$

Since $\nabla \mu_\theta = A^{-1}(\theta)B(\theta)$, with $A^{-1}(\theta)$ and $B(\theta)$ being Lipschitz continuous and bounded. Therefore, according to Lemma 12, there exists a positive constant $L_{\mu,2}$ which satisfies

$$
\|\nabla \mu_{\theta_1} - \nabla \mu_{\theta_2}\| \leq L_{\mu,2}\|\theta_1 - \theta_2\|.
$$

$\square$

**Proposition 2** (restatement of Lemma 2, Lipschitz continuity of $\nabla_\theta \omega^*(\theta)$ (Chen et al., 2021a))**.** *Suppose Assumptions 1-4 holds, then there exists a positive constant $L_{\omega,2}$ such that*

$$
\|\nabla_\theta \omega^*(\theta_1) - \nabla_\theta \omega^*(\theta_2)\|_F \leq L_{\omega,2}\|\theta_1 - \theta_2\|.
$$

*Proof.* The proof follows the derivation of Proposition 8 of (Chen et al., 2021a). However, they make assumption that $\mu_\theta(s)$ is Lipschitz continuous, which we have justified in Lemma 14. We present the proof for the completeness.

We have $\omega^*(\theta) = -A_{\theta,\phi}^{-1} b_{\theta,\phi}$, where $A_{\theta,\phi}$ is defined in (23). The Jacobian of $\omega^*(\theta)$ can be calculated as

$$\nabla_\theta \omega^*(\theta) = -\nabla_\theta(A_{\theta,\phi}^{-1} b_{\theta,\phi})$$
$$= -A_{\theta,\phi}^{-1}(\nabla_\theta A_{\theta,\phi}) A_{\theta,\phi}^{-1} b_{\theta,\phi} - A_{\theta,\phi}(\nabla_\theta b_{\theta,\phi}). \tag{31}$$

We can utilize Lemma 12 to show the Lipschitz continuity of $\nabla \omega^*(\theta)$. We have to verify the Lipschitz continuity and boundedness of $A_{\theta,\phi}^{-1}, b_{\theta,\phi}, \nabla_\theta A_{\theta,\phi}$, and $\nabla_\theta b_{\theta,\phi}$.

The Lipschitz continuity and boundedness of $A_{\theta,\phi}^{-1}$ has been shown in (30). Let $b_1$ and $b_2$ represent $b_{\theta_1,\phi}, b_{\theta_2,\phi}$, we have

$$\|b_1 - b_2\| = \|\mathbb{E}[\bar{r}(s,a,s')\phi(s)] - \mathbb{E}[r(\tilde{s},\tilde{a},\tilde{s}')\phi(\tilde{s})]\|$$
$$\leq \sup_{s,a,s'} \|r(s,a,s')\phi(s)\| \|\mathbb{P}((s,a,s' \in \cdot)) - \mathbb{P}((\tilde{s},\tilde{a},\tilde{s}' \in \cdot))\|_{TV}$$
$$\leq r_{\max}\|\mathbb{P}((s,a,s' \in \cdot)) - \mathbb{P}((\tilde{s},\tilde{a},\tilde{s}' \in \cdot))\|_{TV}$$
$$\leq 2|\mathcal{A}|L_\pi(1 + \log_\rho \kappa^{-1} + (1-\rho)^{-1}\|\theta_1 - \theta_2\|,$$

where the last inequality follows Lemma 6.

We now analyze $\nabla_\theta A_{\theta,\phi}$. We first define

$$A(s,s') := \phi(s)(\gamma\phi(s') - \phi(s))^T, \quad b(s,a,s') := r(s,a,s')\phi(s).$$

as

$$\nabla_\theta A_{\theta,\phi} = \nabla_\theta \left( \sum_{s,a,s'} \mu_\theta(s)\pi_\theta(a|s)P(s'|s,a)A(s,s') \right)$$
$$= \sum_{s,a,s'} \left[ \nabla_\theta\mu_\theta(s)\pi_\theta(a|s)P(s'|s,a)A(s,s') + \mu_\theta\nabla_\theta\pi_\theta(a|s)P(s'|s,a)A(s,s') \right].$$

By Lemma 14 and Lemma 6, and Assumption 3, $\mu_\theta(s), \pi_\theta(a|s), \nabla_\theta\mu_\theta(s), \nabla_\theta\pi_\theta(a|s)$ are Lipschitz continuous and bounded. Therefore, $\nabla_\theta A_{\theta,\phi}$ is Lipschitz and bounded.

Finally, we analyze $\nabla_\theta b_{\theta,\phi}$ by following the same technique.

$$\nabla_\theta b_{\theta,\phi} = \nabla_\theta \left( \sum_{s,a,s'} \mu_\theta(s)\pi_\theta(a|s)P(s'|s,a)b(s,a,s') \right)$$
$$= \sum_{s,a,s'} \left[ \nabla_\theta\mu_\theta(s)\pi_\theta(a|s)P(s'|s,a)b(s,a,s') + \mu_\theta(s)\nabla_\theta\pi_\theta(a|s)P(s'|s,a)b(s,a,s') \right].$$

By Lemma 14 and Lemma 6, and Assumption 3, $\mu_\theta(s), \pi_\theta(a|s), \nabla_\theta\mu_\theta(s), \nabla_\theta\pi_\theta(a|s)$ are Lipschitz continuous and bounded. Thus, $\nabla_\theta b_{\theta,\phi}$ is bounded and Lipschitz continuous.

We have shown the Lipschitz continuity and boundedness of $A_{\theta,\phi}^{-1}, b_{\theta,\phi}, \nabla_\theta A_{\theta,\phi}$, and $\nabla_\theta b_{\theta,\phi}$. Therefore, by applying Lemma 12, we conclude that there exists a positive constant $L_{\omega,2}$ such that $\nabla_\theta\omega^*(\theta)$ in (31) is $L_{\omega,2}$-Lipschitz continuous. $\square$

**Lemma 15** (descent of critic's optimal gap (Markovian sampling)). *Under Assumptions 1-5, with $\omega_{k+1}$ generated by Algorithm 1 given $\omega_k$ and $\theta_k$ under Markovian sampling, then the following holds*

$$\mathbb{E}[\|\bar{\omega}_{k+1} - \omega^*(\theta_{k+1})\|^2|\theta_k] \leq \left( 1 + 4L_\omega N\alpha_k + \frac{L_{\omega,2}^2}{2}C_\theta^2 N\sqrt{d_\theta}\alpha_k^2 \right) \mathbb{E}\|\bar{\omega}_{k+1} - \omega^*(\theta_k)\|^2$$

$$+ \left( \frac{L_{\omega,2}^2}{2}C_\theta^2 N + L_\omega^2 C_\theta^2 N \right)\alpha_k^2 + \frac{\alpha_k}{4}\sum_{i=1}^{N} \|\mathbb{E}[g_a^i(\xi_k,\omega_{k+1}^i,\lambda_{k+1}^i)]\|^2. \tag{32}$$

$$\mathbb{E}[\|\bar{\omega}_{k+1} - \omega^*(\theta_k)\|^2|\theta_k] \leq (1 - 2\lambda_\phi\beta_k)\mathbb{E}\|\bar{\omega}_k - \omega^*(\theta_k)\|^2 + C_{K_1}\beta_k\beta_{k-Z_K} + C_{K_2}\alpha_{k-Z_K}\beta_k. \tag{33}$$

*where the constants are defined as* $C_{K_1} := 4C_2C_\delta Z_K + C_\delta^2$, $C_{K_2} := 4C_1C_\theta Z_K + 2C_3C_\theta Z_K^2 + C_8$, $Z_K := \min\{z \in \mathbb{N}^+ | \kappa\rho^{z-1} \leq \min\{\alpha_K, \beta_K, \eta_K\}\}$.

*Proof.* We begin with the optimality gap of averaged critic variables

$$\begin{aligned}
&\|\bar{\omega}_{k+1} - \omega^*(\theta_{k+1})\|^2 \\
&= \|\bar{\omega}_{k+1} - \omega^*(\theta_k) + \omega^*(\theta_k) - \omega^*(\theta_{k+1})\|^2 \\
&= \|\bar{\omega}_{k+1} - \omega^*(\theta_k)\|^2 + \|\omega^*(\theta_k) - \omega^*(\theta_{k+1})\|^2 + 2\langle\bar{\omega}_{k+1} - \omega^*(\theta_k), \omega^*(\theta_k) - \omega^*(\theta_{k+1})\rangle \\
&\leq \|\bar{\omega}_{k+1} - \omega^*(\theta_k)\|^2 + NL_\omega^2C_\theta^2\alpha_k^2 + 2\langle\bar{\omega}_{k+1} - \omega^*(\theta_k), \nabla\omega^*(\theta_k)^T(\theta_k - \theta_{k+1})\rangle \\
&\quad + 2\langle\bar{\omega}_{k+1} - \omega^*(\theta_k), \omega^*(\theta_k) - \omega^*(\theta_{k+1}) - \nabla\omega^*(\theta_k)^T(\theta_k - \theta_{k+1})\rangle,
\end{aligned} \tag{34}$$

where the inequality is based on the Lipschitz of $\omega^*(\theta)$ implied by Proposition 1

$$\|\omega^*(\theta_k) - \omega^*(\theta_{k+1})\|^2 \leq L_\omega^2\|\theta_k - \theta_{k+1}\|^2,$$

$$\|\theta_k - \theta_{k+1}\|^2 = \sum_{i=1}^{N}\|\alpha_k\hat{\delta}(\xi_k, \omega_k^i, \lambda_k^i)\psi_{\theta_k^i}(s_k, a_k^i)\|^2 \leq N\alpha_k^2C_\theta^2, \tag{35}$$

with $C_\theta := C_\delta C_\psi$, and $C_\delta$ is defined in (39).

The third term in (34) can be bounded as

$$\begin{aligned}
&\langle\bar{\omega}_{k+1} - \omega^*(\theta_k), \nabla\omega^*(\theta_k)^T(\theta_k - \theta_{k+1})\rangle \\
&\leq \|\bar{\omega}_{k+1} - \omega^*(\theta_k)\|\|\nabla\omega^*(\theta_k)^T(\theta_k - \theta_{k+1})\| \\
&\leq \|\nabla\omega*(\theta_k)\|\|\bar{\omega}_{k+1} - \omega^*(\theta_k)\|\|\theta_k - \theta_{k+1}\| \\
&\leq L_\omega\|\bar{\omega}_{k+1} - \omega^*(\theta_k)\|\|\theta_k - \theta_{k+1}\| \\
&\leq \sum_{i=1}^{N}L_\omega\alpha_k\|\bar{\omega}_{k+1} - \omega^*(\theta_k)\|\|g_a^i(\xi_k, \omega_{k+1}^i, \lambda_{k+1}^i)\| \\
&\leq \sum_{i=1}^{N}(2L_\omega\alpha_k\|\bar{\omega}_{k+1} - \omega^*(\theta_k)\|^2 + \frac{\alpha_k}{8}\|g_a^i(\xi_k, \omega_{k+1}^i, \lambda_{k+1}^i)\|^2),
\end{aligned} \tag{36}$$

where the second inequality follows Proposition 1, the third inequality uses triangle inequality, and the last inequality uses Young's inequality.

The last term in (34) can be bounded as

$$\begin{aligned}
&\mathbb{E}\langle\bar{\omega}_{k+1} - \omega^*(\theta_k), \omega^*(\theta_k) - \omega^*(\theta_{k+1}) - \nabla\omega^*(\theta_k)^T(\theta_k - \theta_{k+1})\rangle \\
&\leq \frac{L_{\omega,2}^2}{2}\sqrt{d_\theta}\mathbb{E}\|\bar{\omega}_{k+1} - \omega^*(\theta_k)\|\|\theta_{k+1} - \theta_k\|^2 \\
&\leq \frac{L_{\omega,2}^2}{4}\sqrt{d_\theta}\mathbb{E}\|\bar{\omega}_{k+1} - \omega^*(\theta_k)\|^2\|\theta_{k+1} - \theta_k\|^2 + \frac{L_{\omega,2}^2}{4}\|\theta_{k+1} - \theta_k\|^2 \\
&\leq \frac{L_{\omega,2}^2}{4}\sqrt{d_\theta}NC_\theta^2\alpha_k^2\mathbb{E}\|\bar{\omega}_{k+1} - \omega^*(\theta_k)\|^2 + \frac{L_{\omega,2}^2}{4}NC_\theta^2\alpha_k^2.
\end{aligned} \tag{37}$$

The first inequality uses Lemma 11. The second inequality is induced by Young's inequality. The last inequality follows (35).

Plug (36) and (37) into (34) will yield (32).

We now prove (33). By the critic update rule, we have (crf. $g_c(\theta, \omega) := \mathbb{E}_{\xi \sim \mu_\theta}[\bar{g}_c(\xi, \omega)]$.)

$$
\begin{aligned}
\mathbb{E}\|\bar{\omega}_{k+1} - \omega^*(\theta_k)\|^2 &= \mathbb{E}\|\Pi_{R_\omega}(\bar{\omega}_k + \beta_k \bar{g}_c(\xi_k, \bar{\omega}_k)) - \Pi_{R_\omega}\omega^*(\theta_k)\|^2 \\
&\stackrel{(i)}{\leq} \mathbb{E}\|\bar{\omega}_k + \beta_k \bar{g}_c(\xi, \bar{\omega}_k) - \omega^*(\theta_k)\|^2 \\
&= \|\bar{\omega}_k - \omega^*(\theta_k)\|^2 + \beta_k^2 \mathbb{E}\|\bar{g}_c(\xi_k, \bar{\omega}_k)\|^2 + 2\beta_k \mathbb{E}[\langle \bar{\omega}_k - \omega^*(\theta_k), \mathbb{E}[\bar{g}_c(\xi_k, \bar{\omega}_k)]\rangle] \\
&\stackrel{(ii)}{\leq} \|\bar{\omega}_k - \omega^*(\theta_k)\|^2 + \beta_k^2 C_\delta^2 + 2\beta_k \langle \bar{\omega}_k - \omega^*(\theta_k), \mathbb{E}[\bar{g}_c(\xi_k, \bar{\omega}_k)]\rangle \\
&= \|\bar{\omega}_k - \omega^*(\theta_k)\|^2 + \beta_k^2 C_\delta^2 + 2\beta_k \langle \bar{\omega}_k - \omega^*(\theta_k), g_c(\theta_k, \bar{\omega}_k)\rangle \\
&\quad + 2\beta_k \mathbb{E}\langle \bar{\omega}_k - \omega^*(\theta_k), \mathbb{E}[\bar{g}_c(\xi_k, \bar{\omega}_k)] - g_c(\theta_k, \bar{\omega}_k)\rangle,
\end{aligned}
\tag{38}
$$

where (i) is due to the non-expansiveness of projection to convex set, and (ii) follows

$$
\|\bar{g}_c(\xi, \omega)\| \leq |r(s,a) + \gamma\phi(s')^T\omega - \phi(s)^T\omega| \leq r_{\max} + (1+\gamma)R_\omega := C_\delta.
\tag{39}
$$

The product in the third term in (38) can be bounded as

$$
\begin{aligned}
\langle \bar{\omega}_k - \omega^*(\theta_k), g_c(\theta_k, \bar{\omega}_k)\rangle &= \langle \bar{\omega}_k - \omega^*(\theta_k), \mathbb{E}_{\xi \sim \mu_{\theta_k}}[\bar{g}_c(\xi_k, \bar{\omega}_k)]\rangle \\
&= \langle \bar{\omega}_k - \omega^*(\theta_k), \mathbb{E}_{\xi \sim \mu_{\theta_k}}[\bar{g}_c(\xi_k, \bar{\omega}_k) - g_c(\theta_k, \omega^*(\theta_k))]\rangle \\
&= \beta_k \langle \bar{\omega}_k - \omega^*(\theta_k), \mathbb{E}_{\xi \sim \mu_{\theta_k}}[\phi(s)(\gamma\phi(s') - \phi(s))^T|\theta_k](\bar{\omega}_k - \omega^*(\theta_k))\rangle \\
&= \beta_k \langle \bar{\omega}_k - \omega^*(\theta_k), A_{\theta_k, \phi}(\bar{\omega}_k - \omega^*(\theta_k))\rangle \\
&\leq -\lambda_\phi \beta_k \|\bar{\omega}_k - \omega^*(\theta_k)\|^2.
\end{aligned}
\tag{40}
$$

Here the first equality is due to critic's optimality condition $g_c(\theta_k, \omega^*(\theta_k)) = \mathbb{E}_{\xi_k \sim \mu_{\theta_k}}[\bar{g}_c(\xi_k, \omega^*(\theta_k))|\theta_k] = 0$. The last inequality uses the negative definiteness of $A_{\theta_k, \phi}$ of Assumption 2.

Plug (40) to (38) gives

$$
\begin{aligned}
\mathbb{E}\|\bar{\omega}_{k+1} - \omega^*(\theta_k)\|^2 &\leq (1 - 2\lambda_\phi \beta_k)\|\bar{\omega}_k - \omega^*(\theta_k)\|^2 + \beta_k^2 C_\delta^2 \\
&\quad + 2\beta_k \langle \bar{\omega}_k - \omega^*(\theta_k), \mathbb{E}[\bar{g}_c(\xi_k, \bar{\omega}_k)] - g_c(\theta_k, \bar{\omega}_k)\rangle.
\end{aligned}
\tag{41}
$$

We now bound the last term in (41). By Lemma 16, for any $z \in \mathbb{N}^+$, we have

$$
\begin{aligned}
&\mathbb{E}\langle \bar{\omega}_k - \omega^*(\theta_k), \bar{g}_c(\xi_k, \bar{\omega}_k) - g_c(\theta_k, \bar{\omega}_k)\rangle \\
&\leq C_1 \mathbb{E}\|\theta_k - \theta_{k-z}\| + C_2 \mathbb{E}\|\bar{\omega}_k - \bar{\omega}_{k-z}\| + C_3 \sum_{m=0}^{z-1} \mathbb{E}\|\theta_{k-m} - \theta_{k-z}\| + C_8 \kappa \rho^{z-1} \\
&\stackrel{(i)}{\leq} C_1 \sum_{n=1}^{z} \mathbb{E}\|\theta_{k-n+1} - \theta_{k-n}\| + C_2 \sum_{n=1}^{z} \mathbb{E}\|\bar{\omega}_{k-n+1} - \bar{\omega}_{k-n}\| \\
&\quad + C_3 \sum_{m=0}^{z-1} \sum_{n=1}^{z-m} \mathbb{E}\|\theta_{k-m-n+1} - \theta_{k-m-n}\| + C_8 \kappa \rho^{z-1} \\
&\leq 2C_1 C_\theta \sum_{n=1}^{z} \alpha_{k-n} + 2C_2 C_\delta \sum_{n=1}^{z} \beta_{k-n} + C_3 C_\theta \sum_{m=0}^{z-1} \sum_{n=1}^{z-m} \alpha_{k-m-n} + C_8 \kappa \rho^{z-1} \\
&\stackrel{(ii)}{\leq} 2C_1 C_\theta z \alpha_{k-z} + 2C_2 C_\delta z \beta_{k-z} + C_3 C_\theta z(z-1)\alpha_{k-z} + C_8 \kappa \rho^{z-1},
\end{aligned}
\tag{42}
$$

where the $(i)$ uses triangle inequality, $(ii)$ uses the non-increasing property of step sizes.

Let $z = Z_K$, we have (crf. $Z_K := \min\{z \in \mathbb{N}^+ | \kappa \rho^{z-1} \leq \min\{\alpha_K, \beta_K, \eta_K\}\}$)

$$
\begin{aligned}
&\mathbb{E}\langle \bar{\omega}_k - \omega^*(\theta_k), \bar{g}_c(\xi_k, \bar{\omega}_k) - g_c(\theta_k, \bar{\omega}_k)\rangle \\
&\leq 2C_1 C_\theta Z_K \alpha_{k-Z_K} + 2C_2 C_\delta Z_K \beta_{k-Z_K} + C_3 C_\theta Z_K^2 \alpha_{k-Z_K} + C_8 \alpha_{k-Z_K}.
\end{aligned}
\tag{43}
$$

Plug (43) into (41) will yield

$$\|\bar{\omega}_{k+1} - \omega^*(\theta_k)\|^2 \leq (1 - 2\lambda_\phi \beta_k)\|\bar{\omega}_k - \omega^*(\theta_k)\|^2 + C_\delta^2 \beta_k^2$$
$$+ 4C_1 C_\theta Z_K \alpha_{k-Z_K} + 4C_2 C_\delta Z_K \beta_{k-Z_K} + 2C_3 C_\theta Z_K^2 \alpha_{k-Z_K} + 2C_8 \alpha_{k-Z_K}.$$

By defining $C_{K_1} := 4C_2 C_\delta Z_K + C_\delta^2$, $C_{K_2} := 4C_1 C_\theta Z_K + 2C_3 C_\theta Z_K^2 + C_8$, we complete the proof. □

**Lemma 16.** *Consider the sequence generated by Algorithm 1, for any $z \in \mathbb{N}^+$, we have*

$$\mathbb{E}\langle \bar{\omega}_k - \omega^*(\theta_k), \bar{g}_c(\xi_k, \bar{\omega}_k) - g_c(\theta_k, \bar{\omega}_k)\rangle \leq C_1\|\theta_k - \theta_{k-z}\| + C_2\|\bar{\omega}_k - \bar{\omega}_{k-z}\|$$
$$+ C_3 \sum_{m=0}^{z-1} \|\theta_{k-m} - \theta_{k-z}\| + C_8 \kappa \rho^{z-1}, \tag{44}$$

*where $C_1 := 4R_\omega C_\delta |\mathcal{A}| L_\pi (1 + \log_\rho \kappa^{-1} + (1-\rho)^{-1}) + 2C_\delta L_\omega$, $C_2 := 4(1+\gamma)R_\omega + 2C_\delta$, $C_3 := 4R_\omega C_\delta |\mathcal{A}| L_\pi$, $C_8 := 8R_\omega C_\delta$.*

Basically, this lemma shows that the term on the left hand side of (44) is of order $\tilde{\mathcal{O}}(\alpha_k + \beta_k)$.

*Proof.* Consider the Markov chain since timestep $k - z$:

$$s_{k-z} \xrightarrow{\theta_{k-z}} a_{k-z} \xrightarrow{\mathcal{P}} s_{k-z+1} \xrightarrow{\theta_{k-z+1}} a_{k-z+1} \cdots \xrightarrow{\theta_{k-1}} a_{k-1} \xrightarrow{\mathcal{P}} s_k \xrightarrow{\theta_k} a_k \xrightarrow{\mathcal{P}} s_{k+1}.$$

Also consider the auxiliary Markov chain with fixed policy since timestep $k - z$:

$$s_{k-z} \xrightarrow{\theta_{k-z}} a_{k-z} \xrightarrow{\mathcal{P}} s_{k-z+1} \xrightarrow{\theta_{k-z}} \tilde{a}_{k-z+1} \cdots \xrightarrow{\theta_{k-z}} \tilde{a}_{k-1} \xrightarrow{\mathcal{P}} \tilde{s}_k \xrightarrow{\theta_{k-z}} \tilde{a}_k \xrightarrow{\mathcal{P}} \tilde{s}_{k+1}.$$

Throughout the proof of this lemma, we will use $\theta, \theta', \bar{\omega}, \bar{\omega}', \xi, \tilde{\xi}$ as shorthand notations of $\theta_k, \theta_{k-z}, \bar{\omega}_k, \bar{\omega}_{k-z}, \xi_k, \tilde{\xi}_k$.

For the ease of expression, define

$$\Delta_1(\xi, \theta, \omega) := \langle \omega - \omega^*(\theta), \bar{g}_c(\xi, \omega) - g_c(\theta, \omega)\rangle.$$

Therefore, we have

$$\langle \bar{\omega}_k - \omega^*(\theta_k), \bar{g}_c(\xi_k, \bar{\omega}_k) - g_c(\theta_k, \bar{\omega}_k)\rangle = \Delta_1(\xi, \theta, \bar{\omega})$$
$$= \underbrace{\Delta_1(\xi, \theta, \bar{\omega}) - \Delta_1(\xi, \theta', \bar{\omega})}_{I_1} + \underbrace{\Delta_1(\xi, \theta', \bar{\omega}) - \Delta_1(\xi, \theta', \bar{\omega}')}_{I_2}$$
$$+ \underbrace{\Delta_1(\xi, \theta', \bar{\omega}') - \Delta_1(\tilde{\xi}, \theta', \bar{\omega}')}_{I_3} + \underbrace{\Delta_1(\tilde{\xi}, \theta', \bar{\omega}')}_{I_4}. \tag{45}$$

$I_1$ can be expressed as

$$I_1 = \langle \bar{\omega} - \omega^*(\theta), \bar{g}_c(\xi, \bar{\omega}) - g_c(\theta, \bar{\omega})\rangle - \langle \bar{\omega} - \omega^*(\theta'), \bar{g}_c(\xi, \bar{\omega}) - g_c(\theta', \bar{\omega})\rangle$$
$$= \langle \bar{\omega} - \omega^*(\theta), \bar{g}_c(\xi, \bar{\omega}) - g_c(\theta, \bar{\omega})\rangle - \langle \bar{\omega} - \omega^*(\theta), \bar{g}_c(\xi, \bar{\omega}) - g_c(\theta', \bar{\omega})\rangle$$
$$+ \langle \omega^*(\theta) - \omega^*(\theta'), \bar{g}_c(\xi, \bar{\omega}) - g_c(\theta', \bar{\omega})\rangle$$
$$\leq \|\bar{\omega} - \omega^*(\theta)\|\|g_c(\theta', \bar{\omega}) - g_c(\theta, \bar{\omega})\| + \|\omega^*(\theta) - \omega^*(\theta')\|\|\bar{g}_c(\xi, \bar{\omega}) - g_c(\theta', \bar{\omega})\|. \tag{46}$$

The first term can be bounded as

$$\|\bar{\omega} - \omega^*(\theta)\|\|g_c(\theta', \bar{\omega}) - g_c(\theta, \bar{\omega})\| \leq 2R_\omega \|\mathbb{E}_{\xi \sim \mu'_\theta}[\bar{g}_c(\xi, \bar{\omega})] - \mathbb{E}_{\xi \sim \mu_\theta}[\bar{g}_c(\xi, \bar{\omega})]\|$$
$$\leq 4R_\omega \sup_\xi \|\bar{g}_c(\xi, \bar{\omega})\| d_{TV}(\mu'_\theta \otimes \pi'_\theta \otimes \mathcal{P}, \mu_\theta \otimes \pi_\theta \otimes \mathcal{P})$$
$$\leq 4R_\omega C_\delta d_{TV}(\mu'_\theta \otimes \pi'_\theta \otimes \mathcal{P}, \mu_\theta \otimes \pi_\theta \otimes \mathcal{P})$$
$$\leq 4R_\omega C_\delta |\mathcal{A}| L_\pi (1 + \log_\rho \kappa^{-1} + (1-\rho)^{-1})\|\theta - \theta'\|, \tag{47}$$

where the first inequality is due to $\omega \leq R_\omega$ induced by the projection step of critic's update. The third inequality is due to $\|\bar{g}_c(\xi, \bar{\omega})\| \leq C_\delta$, and the last inequality follows Lemma 6.

By the Lipschitz conitinuous of $\omega^*(\theta)$ proposed in Proposition 1, the second term in (46) can be bounded as

$$\|\omega^*(\theta) - \omega^*(\theta')\|\|\bar{g}_c(\xi, \bar{\omega}) - g_c(\theta, \bar{\omega})\| \leq L_\omega\|\theta - \theta'\|\|\bar{g}_c(\xi, \bar{\omega}) - g_c(\theta, \bar{\omega})\|$$
$$\leq 2C_\delta L_\omega\|\theta - \theta'\| \tag{48}$$

Plug (47) and (48) into (46), we can bound $I_1$ as

$$I_1 \leq (4R_\omega C_\delta|\mathcal{A}|L_\pi(1 + \log_\rho \kappa^{-1} + (1-\rho)^{-1}) + 2C_\delta L_\omega)\|\theta - \theta'\|. \tag{49}$$

$I_2$ can be decomposed by

$$\begin{aligned}
I_2 &= \langle \bar{\omega} - \omega^*(\theta'), \bar{g}_c(\xi, \bar{\omega}) - g_c(\theta', \bar{\omega})\rangle - \langle \bar{\omega}' - \omega^*(\theta'), \bar{g}_c(\xi, \bar{\omega}') - g_c(\theta', \bar{\omega}')\rangle \\
&= \langle \bar{\omega} - \omega^*(\theta'), \bar{g}_c(\xi, \bar{\omega}) - g_c(\theta', \bar{\omega})\rangle - \langle \bar{\omega}' - \omega^*(\theta'), \bar{g}_c(\xi, \bar{\omega}) - g_c(\theta', \bar{\omega})\rangle \\
&\quad + \langle \bar{\omega}' - \omega^*(\theta'), \bar{g}_c(\xi, \bar{\omega}) - \bar{g}_c(\xi, \bar{\omega}') - g_c(\theta', \bar{\omega}) + g_c(\theta', \bar{\omega}')\rangle \\
&= \langle \bar{\omega} - \omega^*(\theta'), \bar{g}_c(\xi, \bar{\omega}) - g_c(\theta', \bar{\omega})\rangle \\
&\quad + \langle \bar{\omega}' - \omega^*(\theta'), \bar{g}_c(\xi, \bar{\omega}) - \bar{g}_c(\xi, \bar{\omega}') - g_c(\theta', \bar{\omega}) + g_c(\theta', \bar{\omega}')\rangle \\
&\leq 2C_\delta\|\bar{\omega} - \omega^*(\theta')\| + \langle \bar{\omega}' - \omega^*(\theta'), \bar{g}_c(\xi, \bar{\omega}) - \bar{g}_c(\xi, \bar{\omega}') - g_c(\theta', \bar{\omega}) + g_c(\theta', \bar{\omega}')\rangle.
\end{aligned}$$

The last term can be bounded as

$$\begin{aligned}
&\langle \bar{\omega}' - \omega^*(\theta'), \bar{g}_c(\xi, \bar{\omega}) - \bar{g}_c(\xi, \bar{\omega}') - g_c(\theta', \bar{\omega}) + g_c(\theta', \bar{\omega}')\rangle \\
&\leq \|\bar{\omega} - \omega^*(\theta')\|(\|\bar{g}_c(\xi, \bar{\omega}) - \bar{g}_c(\xi, \bar{\omega}')\| + \|g_c(\theta', \bar{\omega}') - g_c(\theta', \bar{\omega})\|) \\
&\leq 2R_\omega(\|\bar{g}_c(\xi, \bar{\omega}) - \bar{g}_c(\xi, \bar{\omega}')\| + \|g_c(\theta', \bar{\omega}') - g_c(\theta', \bar{\omega})\|) \\
&\leq 2R_\omega(\|\bar{g}_c(\xi, \bar{\omega}) - \bar{g}_c(\xi, \bar{\omega}')\| + \mathbb{E}_{\xi \sim \mu_{\theta'}}\|\bar{g}_c(\xi, \bar{\omega}') - g_c(\xi, \bar{\omega})\|) \\
&\leq 4R_\omega(1 + \gamma)\|\bar{\omega} - \bar{\omega}'\|, \tag{50}
\end{aligned}$$

where the first inequality applies Cauchy-Schwartz inequality and triangle inequality, the second inequality follows the projection of each critic step. The last inequality is due to

$$\begin{aligned}
\|\bar{g}_c(\xi, \bar{\omega}) - \bar{g}_c(\xi, \bar{\omega}')\| &= \|\phi(s)(\gamma\phi(s')^T(\bar{\omega} - \bar{\omega}') - \phi(s)^T(\bar{\omega} - \bar{\omega}'))\| \\
&\leq \gamma\|\phi(s')^T(\bar{\omega} - \bar{\omega}')\| + \|\phi(s)^T(\bar{\omega} - \bar{\omega}')\| \\
&\leq (1 + \gamma)\|\bar{\omega} - \bar{\omega}'\|.
\end{aligned}$$

Thus, $I_2$ can be bounded as

$$I_2 \leq (4(1 + \gamma)R_\omega + 2C_\delta)\|\bar{\omega} - \bar{\omega}'\|. \tag{51}$$

We bound $I_3$ as

$$\begin{aligned}
\mathbb{E}[I_3|\theta', s_{k-z+1}] &= \mathbb{E}[\Delta_1(\xi, \theta', \bar{\omega}') - \Delta_1(\tilde{\xi}, \theta', \bar{\omega}')|\theta', s_{k-z+1}] \\
&\leq 2\sup_\xi |\Delta_1(\xi, \theta', \bar{\omega}')| \, d_{TV}(\mathbb{P}(\xi \in \cdot|\theta', s_{k-z+1}), \mathbb{P}(\tilde{\xi} \in \cdot|\theta', s_{k-z+1})) \\
&\leq 8R_\omega C_\delta d_{TV}(\mathbb{P}(\xi \in \cdot|\theta', s_{k-z+1}), \mathbb{P}(\tilde{\xi} \in \cdot|\theta', s_{k-z+1})) \\
&\leq 4R_\omega C_\delta|\mathcal{A}|L_\pi \sum_{m=0}^{z-1} \|\theta_{k-m} - \theta_{k-z}\|. \tag{52}
\end{aligned}$$

Here, the second inequality is due to $\|\Delta_1(\xi, \theta', \bar{\omega}')\| \leq \|\omega' - \omega^*(\theta')\|\|\bar{g}_c(\xi, \omega') - g_c(\theta', \omega')\| \leq 4R_\omega C_\delta$, and the last inequality is according to Lemma 9.

We now bound $I_4$

$$
\begin{aligned}
\mathbb{E}[I_4|\theta', \bar{\omega}', s_{k+z-1}] &= \mathbb{E}[\Delta_1(\tilde{\xi}, \theta', \bar{\omega}')|\theta', \bar{\omega}', s_{k-z+1}] \\
&\leq \sup_{\xi} |\Delta_1(\xi, \theta', \bar{\omega}')| \|\mathbb{P}(\xi \in \cdot|\theta', s_{k-z+1}) - \mu_{\theta'} \otimes \pi_{\theta'} \otimes \mathcal{P}\| \\
&\leq 8R_\omega C_\delta d_{TV}(\mathbb{P}(\tilde{x} \in \cdot|\theta', s_{t-z+1}), \mu_{\theta'} \otimes \pi_{\theta'} \otimes \mathcal{P}) \\
&\leq 8R_\omega C_\delta \kappa \rho^{z-1},
\end{aligned}
\tag{53}
$$

where the last inequality follows Lemma 7.

Plug (49), (51), (52), and (53) into (45), we get

$$
\begin{aligned}
\mathbb{E}[\Delta_1(\xi, \theta, \bar{\omega})] &\leq (4R_\omega C_\delta |\mathcal{A}| L_\pi (1 + \log_\rho \kappa^{-1} + (1-\rho)^{-1}) + 2C_\delta L_\omega)\mathbb{E}\|\theta_k - \theta_{k-z}\| \\
&\quad + (4(1+\gamma)R_\omega + 2C_\delta)\mathbb{E}\|\bar{\omega}_k - \bar{\omega}_{k-z}\| \\
&\quad + (4R_\omega C_\delta |\mathcal{A}| L_\pi) \sum_{m=0}^{z-1} \mathbb{E}\|\theta_{k-m} - \theta_{k-z}\| \\
&\quad + 8R_\omega C_\delta \kappa \rho^{z-1},
\end{aligned}
$$

which completes the proof. $\qquad\square$

## C.2  Error of reward estimator

The analysis for the error of reward estimator is similar to critic. To see this, we only need to change $\bar{g}_c(\xi, \bar{\omega})$ into $\bar{g}_r(\xi, \bar{\lambda}) := (r(s,a) - \varphi(s,a)^T \bar{\lambda})\varphi(s,a)$ to recover most of the proofs.

Lemma 17 and 18 are the counter parts of Lemma 15 and 16 for reward estimator.

**Lemma 17.** *Suppose Assumptions 1 , 2, 3, 4 hold, with $\lambda_{k+1}$ generated by Algorithm 1 given $\lambda_k$ and $\theta_k$ under Markovian sampling, then the following holds*

$$
\begin{aligned}
\mathbb{E}[\|\bar{\lambda}_{k+1} - \lambda^*(\theta_{k+1})\|^2|\theta_k] &\leq \left(1 + 4L_\lambda N\alpha_k + \frac{L_{\lambda,2}^2}{2}C_\theta^2 N\sqrt{d_\theta}\alpha_k^2\right)\mathbb{E}\|\bar{\lambda}_{k+1} - \lambda^*(\theta_k)\|^2 \\
&\quad + \left(\frac{L_{\lambda,2}^2}{2}C_\theta^2 N + L_\lambda^2 C_\theta^2 N\right)\alpha_k^2 + \frac{\alpha_k}{4}\sum_{i=1}^N \|\mathbb{E}[g_a^i(\xi_k, \lambda_{k+1}^i, \lambda_{k+1}^i)]\|^2. \quad (54)
\end{aligned}
$$

$$
\mathbb{E}[\|\bar{\lambda}_{k+1} - \lambda^*(\theta_k)\|^2|\theta_k] \leq (1 - 2\eta_k\lambda_\varphi)\|\bar{\lambda}_k - \lambda^*(\theta_k)\|^2 + C_{K_3}\eta_k\eta_{k-Z_K} + C_{K_4}\eta_k\alpha_{k-Z_K}, \tag{55}
$$

*where $C_{K_3} := 4C_6C_\lambda Z_K + C_\lambda^2$, $C_{K_4} := 4C_5C_\theta Z_K + 2C_7C_\theta Z_K^2 + C_8, Z_K := \min\{z \in \mathbb{N}^+|\kappa\rho^{z-1} \leq \min\{\alpha_K, \beta_K, \eta_K\}\}, C_\lambda := r_{\max} + R_\lambda \geq \max_{s,a,\lambda}\|(r(s,a) - \lambda^T\varphi(s,a)\varphi(s,a))\|$.*

**Lemma 18.** *Consider the sequence generated by Algorithm 1, for any $z \in \mathbb{N}^+$, we have*

$$
\begin{aligned}
\mathbb{E}[\langle \bar{\lambda}_k - \lambda^*(\theta), \bar{g}_r(\xi_k, \bar{\lambda}_k) - g_r(\theta_k, \bar{\lambda}_k)\rangle] &\leq C_5\|\theta_k - \theta_{k-z}\| + C_6\|\lambda_k - \lambda_{k-z}\| \\
&\quad + C_7\sum_{m=0}^{z-1}\|\theta_{k-m} - \theta_{k-z}\| + C_8\kappa\rho^{z-1}, \quad (56)
\end{aligned}
$$

*where $C_5 := 4R_\lambda C_\lambda |\mathcal{A}| L_\pi (1 + \log_\rho \kappa^{-1} + (1-\rho)^{-1}) + 2C_\lambda L_\lambda, C_6 := 4R_\lambda + 2C_\lambda, C_7 := 4R_\lambda C_\lambda |\mathcal{A}| L_\pi, C_8 := 8R_\lambda C_\lambda$.*

## C.3  Consensus error

**Lemma 19** (restatement of Lemma 1, bound of consensus error)**.** *Define the matrix representation of critics and reward estimators' parameters as $\boldsymbol{\omega}_k := [\omega_k^1, \cdots, \omega_k^N]^T, \boldsymbol{\lambda}_k := [\lambda_k^1, \cdots, \lambda_k^N]^T$. Let $Q := I - \frac{1}{N}\mathbf{1}\mathbf{1}^T$, then*

*the consensus error can be expressed as* $\sum_{i=1}^{N}\|\omega_k^i - \bar{\omega}_k\|^2 = \|Q\boldsymbol{\omega}_k\|^2$, $\sum_{i=1}^{N}\|\lambda_k^i - \bar{\lambda}_k\|^2 = \|Q\boldsymbol{\lambda}_k\|^2$. *Suppose Asssumption 5 holds. Let* $\{\omega_k^i\}_k, \{\lambda_k^i\}_k$ *be the sequence generated by the Algorithm 1, then for any* $k \geq 1$, *the following inequalities hold*

$$\|Q\boldsymbol{\omega}_k\| \leq \nu^{\frac{k}{K_c}-1}\|\boldsymbol{\omega}_0\| + 4\sqrt{N}C_\delta \sum_{t=0}^{k} \beta_t \nu^{\frac{k-t}{K_c}-1} \tag{57}$$

$$\|Q\boldsymbol{\lambda}_k\| \leq \nu^{\frac{k}{K_c}-1}\|\boldsymbol{\lambda}_0\| + 4\sqrt{N}C_\lambda \sum_{t=0}^{k} \beta_t \nu^{\frac{k-t}{K_c}-1}, \tag{58}$$

*where* $\nu \in (0,1)$ *is the second largest singular value of* $W$.

*Proof.* We will prove the bound in (57) for the critic variables. The analysis for reward estimator in (58) follows the same routine. To simplify the notation, we will use $g_k^i$ to represent $g_c^i(\xi_k, \omega_k^i)$ throughout the proof of this lemma. We also use $e_k^i$ to represent the projection update $e_k^i := \Pi_{R_\omega}(\omega_k^i - \beta_k g_k^i) - (\omega_k^i - \beta_k g_k^i)$. Define $\bar{g}_k := \frac{1}{N}\sum_{i=1}^{N} g_k^i$; $\bar{e}_k := \frac{1}{N}\sum_{i=1}^{N} e_k^i$, and the corresponding matrix exressions as

$$G_k := \begin{bmatrix} (g_k^1)^T, \\ \vdots \\ (g_k^N)^T \end{bmatrix}, E_k := \begin{bmatrix} (e_k^1)^T, \\ \vdots \\ (e_k^N)^T \end{bmatrix}.$$

According to the update rule of critic variables, the following equalities holds

$$\boldsymbol{\omega}_{k+1} = \begin{cases} W\boldsymbol{\omega}_k - \beta_k G_k + E_k, & \text{if } k \bmod K_c = 0 \\ \boldsymbol{\omega}_k - \beta_k G_k + E_k, & \text{otherwise.} \end{cases} \tag{59}$$

To bound the consensus error, We first bound the consensus error of critic's update as

$$\|QG_k\| = \sqrt{\sum_{i=1}^{N}\|g_k^i - \bar{g}_k\|^2} \overset{(i)}{\leq} \sqrt{\sum_{i=1}^{N} 2\|g_k^i\|^2 + 2\|\bar{g}_k\|^2} \leq 2\sqrt{N}C_\delta. \tag{60}$$

$$\|QE_k\| = \sqrt{\sum_{i=1}^{N}\|e_t^i - \bar{e}_t\|^2} \leq \sqrt{\sum_{i=1}^{N} 2\|e_k^i\|^2 + 2\|\bar{e}_k\|^2} \overset{(ii)}{\leq} \sqrt{\sum_{i=1}^{N} 2\beta_k^2\|g_k^i\|^2 + 2\beta_k^2\|\bar{g}_k\|^2} \leq 2\beta_k\sqrt{N}C_\delta, \tag{61}$$

where $(i)$ is due to $\|g_k^i\| \leq C_\delta$; $(ii)$ is ensured by the convexity of the projection set.

We now study the consensus error of critic variables. Let $k' = \lfloor \frac{k}{K_c} \rfloor * K_c$. By the update rule in (59), we have

$$\begin{aligned} Q\boldsymbol{\omega}_{k'} &= QW\boldsymbol{\omega}_{k'-1} - \beta_{k'-1}QG_{k'-1} + QE_{k'-1} \\ &= WQ\boldsymbol{\omega}_{k'-1} - \beta_{k'-1}QG_{k'-1} + QE_{k'-1} \\ &= WQ\boldsymbol{\omega}_{k'-K_c} - \sum_{t=k'-K_c}^{k'-1} \beta_t W^{\lceil k'-1-t \rceil}QG_t + \sum_{k'-K_c}^{t=k'-1} W^{\lceil k'-1-t \rceil}QE_t, \end{aligned} \tag{62}$$

where the second equality is due to the doubly stochasticity of matrix $W$ implied by Assumption 5: $QW = W - \frac{1}{N}\mathbf{1}\mathbf{1}^T W = W - \frac{1}{N}W\mathbf{1}\mathbf{1}^T = WQ$. The last equality is indicated by the update rule that

$$\boldsymbol{\omega}_{k'-1} = \boldsymbol{\omega}_{k'-K_c} - \sum_{t=k'-K_c}^{k'-2} \beta_t G_t + \sum_{t=k'-K_c}^{k'-2} E_t.$$

Expand the recursion in (62), we have

$$Q\boldsymbol{\omega}_{k'} = W^{\frac{k'}{K_c}}Q\boldsymbol{\omega}_0 - \sum_{t=0}^{k'-1} W^{c_t}\beta_t QG_t + \sum_{t=0}^{k'-1} W^{c_t}QE_t,$$

where $c_t := \lceil \frac{k'-1-t}{K_c} \rceil$. Therefore, the $k_{th}$ iteration's consensus error can be expressed as

$$
\begin{aligned}
Q\boldsymbol{\omega}_k &= Q\boldsymbol{\omega}_{k'} - \sum_{t=k'}^{k-1} \beta_t Q G_t + \sum_{t=k'}^{k-1} Q E_t \\
&= W^{\frac{k'}{K_c}} Q\boldsymbol{\omega}_0 - \sum_{t=0}^{k} W^{c_t} \beta_t Q G_t + \sum_{t=0}^{k} W^{c_t} Q E_t.
\end{aligned}
\tag{63}
$$

Take norm on the each side of (63) and apply triangle inequality, we get

$$
\begin{aligned}
\|Q\boldsymbol{\omega}_k\| &\leq \|W^{\frac{k'}{K_c}} Q\boldsymbol{\omega}_0\| + \sum_{t=0}^{k} \beta_t \|W^{c_t} Q G_t\| + \sum_{t=0}^{k} \|W^{c_t} Q E_t\| \\
&\overset{(i)}{\leq} \nu^{\frac{k'}{K_c}} \|\boldsymbol{\omega}_0\| + \sum_{t=0}^{k} \beta_t \nu^{c_t} \|G_t\| + \sum_{t=0}^{k} \nu^{c_t} \|E_t\| \\
&\overset{(ii)}{\leq} \nu^{\frac{k'}{K_c}} \|\boldsymbol{\omega}_0\| + 4\sqrt{N} C_\delta \sum_{t=0}^{k} \beta_t \nu^{c_t} \\
&\overset{(iii)}{\leq} \nu^{\frac{k}{K_c}-1} \|\boldsymbol{\omega}_0\| + 4\sqrt{N} C_\delta \sum_{t=0}^{k} \beta_t \nu^{\frac{k-t}{K_c}-1}.
\end{aligned}
$$

where $(i)$ inequality uses Lemma 10 and the fact that the spectral of $Q$ is less than 1; $(ii)$ is due to (60) and (61); $(iii)$ uses the fact that $\frac{k'}{K_c} \geq \frac{k}{K_c} - 1$ and $\lceil \frac{k'-1-t}{K_c} \rceil = \lceil \frac{k-t}{K_c} + \frac{k'-k-1}{K_c} \rceil \geq \lceil \frac{k-t}{K_c} \rceil - 1 \geq \frac{k-t}{K_c} - 1$. Thus, the proof for (57) is completed. The proof of (58) follows a similar procedure, we leave it as an exercise to reader.

$\square$

### C.4 Error of actor

The following lemma characterizes the sampling error of actor.

**Lemma 20.** *Consider the sequence generated by Algorithm 1, for any $z \geq 1$ we have*

$$
\begin{aligned}
&\|\mathbb{E}_{\xi \sim \mu_{\theta_k}}[\delta(\xi, \theta_k)\psi_{\theta_k^i}(s_k, a_k^i)] - \mathbb{E}[\delta(\xi_k, \theta_k)\psi_{\theta_k^i}(s_k, a_k^i)]\| \\
&\leq 2C_\theta \kappa \rho^{z-1} + C_{12} \sum_{m=0}^{z-1} \|\theta_{k-m} - \theta_{k-z}\| + C_{13}\|\theta_k - \theta_{k-z}\| + C_{14}\|\theta_k^i - \theta_{k-z}^i\|,
\end{aligned}
\tag{64}
$$

*where $C_{12} := 2C_\theta|\mathcal{A}|L_\pi$, $C_{13} := |\mathcal{A}|L(\log_\rho \kappa^{-1} + (1-\rho)^{-1})C_\theta + 2(1+\gamma)L_V$, $C_{14} := 2C_\delta L_\psi$.*

*Proof.* Consider the Markov chain since timestep $k - z$:

$$
s_{k-z} \xrightarrow{\theta_{k-z}} a_{k-z} \xrightarrow{\mathcal{P}} s_{k-z+1} \xrightarrow{\theta_{k-z+1}} a_{k-z+1} \cdots \xrightarrow{\theta_{k-1}} a_{k-1} \xrightarrow{\mathcal{P}} s_k \xrightarrow{\theta_k} a_k \xrightarrow{\mathcal{P}} s_{k+1}.
$$

Also consider the auxiliary Markov chain with fixed policy since timestep $k - z$:

$$
s_{k-z} \xrightarrow{\theta_{k-z}} a_{k-z} \xrightarrow{\mathcal{P}} s_{k-z+1} \xrightarrow{\theta_{k-z}} \tilde{a}_{k-z+1} \cdots \xrightarrow{\theta_{k-z}} \tilde{a}_{k-1} \xrightarrow{\mathcal{P}} \tilde{s}_k \xrightarrow{\theta_{k-z}} \tilde{a}_k \xrightarrow{\mathcal{P}} \tilde{s}_{k+1}.
$$

Throughout the proof of this lemma, we wil use $\psi_{\theta^i}$ to represent $\psi_{\theta^i}(s_k, a_k^i)$ for brevity.

We define the following notation for the ease of discussion

$$
\Delta_3(\xi, \theta) := \mathbb{E}_{\xi \sim \mu_\theta}[\delta(\xi, \theta)\psi_{\theta^i}] - \delta(\xi, \theta)\psi_{\theta^i}.
$$

Then our objective is to bound

$$
\mathbb{E}[\|\Delta_3(\xi_k, \theta_k)\| \mid \theta_{k-z}].
$$

We decompose $\|\Delta_3(\xi_k, \theta_k)\|$ by applying triangle inequality

$$
\|\Delta_3(\xi_k, \theta_k)\| \leq \underbrace{\|\Delta_3(\xi_k, \theta_k) - \Delta_3(\xi_k, \theta_{k-z})\|}_{I_1}
$$
$$
+ \underbrace{\|\Delta_3(\xi_k, \theta_{k-z}) - \Delta_3(\tilde{\xi}_k, \theta_{k-z})\|}_{I_2}
$$
$$
+ \underbrace{\|\Delta_3(\tilde{\xi}_k, \theta_{k-z})\|}_{I_3}. \tag{65}
$$

We apply triangle inequality again to bound $I_1$ as

$$
I_1 \leq \underbrace{\|\delta(\xi_k, \theta_{k-z})\psi_{\theta^i_{k-z}} - \delta(\xi_k, \theta_k)\psi_{\theta^i_k}\|}_{I_1^{(1)}}
$$
$$
+ \underbrace{\|\mathbb{E}_{\xi \sim \mu_{\theta_k}}[\delta(\xi, \theta_k)\psi_{\theta^i_k}] - \mathbb{E}_{\xi \sim \mu_{\theta_{k-z}}}[\delta(\xi, \theta_{k-z})\psi_{\theta^i_{k-z}}]\|}_{I_1^{(2)}} \tag{66}
$$

$I_1^{(1)}$ can be bounded as

$$
\begin{aligned}
I_1^{(1)} &= \|\delta(\xi_k, \theta_{k-z})\psi_{\theta^i_{k-z}} - \delta(\xi_k, \theta_k)\psi_{\theta^i_k}\| \\
&\leq \|\delta(\xi_k, \theta_{k-z})\psi_{\theta^i_{k-z}} - \delta(\xi_k, \theta_k)\psi_{\theta^i_{k-z}}\| \\
&\quad + \|\delta(\xi_k, \theta_k)\psi_{\theta^i_{k-z}} - \delta(\xi_k, \theta_k)\psi_{\theta^i_k}\| \\
&\leq \||\gamma(V_{\theta_{k-z}}(s') - V_{\theta_k}(s')) + (V_{\theta_{k-z}}(s) - V_{\theta_{k-z}}(s'))|\psi^i_{k-z}\| \\
&\quad + \|\delta(\xi_k, \theta_k)\psi_{\theta^i_{k-z}} - \delta(\xi_k, \theta_k)\psi_{\theta^i_k}\| \\
&\leq (1+\gamma)L_V\|\theta_k - \theta_{k-z}\| + \|\delta(\xi_k, \theta_k)\psi_{\theta^i_{k-z}} - \delta(\xi_k, \theta_k)\psi_{\theta^i_k}\| \\
&\leq (1+\gamma)L_V\|\theta_k - \theta_{k-z}\| + C_\delta L_\psi\|\theta^i_k - \theta^i_{k-z}\|, \tag{67}
\end{aligned}
$$

where the second last inequality follows the Lipschitz continuous of value function in Lemma 5, and the last inequality uses Lipschitz continuous of $\psi_{\theta^i}$.

$I_1^{(2)}$ can be bounded as

$$
\begin{aligned}
I_1^{(2)} &= \|\mathbb{E}_{\xi \sim \mu_{\theta_k}}[\delta(\xi, \theta_k)\psi_{\theta^i_k}] - \mathbb{E}_{\xi \sim \mu_{\theta_{k-z}}}[\delta(\xi, \theta_{k-z})\psi_{\theta^i_{k-z}}]\| \\
&= \|\mathbb{E}_{\xi \sim \mu_{\theta_k}}[\delta(\xi, \theta_{k-z})\psi_{\theta^i_{k-z}}] - \mathbb{E}_{\xi \sim \mu_{\theta_{k-z}}}[\delta(\xi, \theta_{k-z})\psi_{\theta^i_{k-z}}] \\
&\quad + \mathbb{E}_{\xi \sim \mu_{\theta_k}}[\delta(\xi, \theta_k)\psi_{\theta^i_k} - \delta(\xi, \theta_{k-z})\psi_{\theta^i_{k-z}}]\| \\
&\leq |\mathcal{A}|L(\log_\rho \kappa^{-1} + (1-\rho)^{-1})C_\theta\|\theta_k - \theta_{k-z}\| \\
&\quad + \|\mathbb{E}_{\xi \sim \mu_{\theta_k}}[\delta(\xi, \theta_k)\psi_{\theta^i_k} - \delta(\xi, \theta_{k-z})\psi_{\theta^i_{k-z}}]\| \\
&\leq |\mathcal{A}|L(\log_\rho \kappa^{-1} + (1-\rho)^{-1})C_\theta\|\theta_k - \theta_{k-z}\| \\
&\quad + (1+\gamma)L_V\|\theta_k - \theta_{k-z}\| + C_\delta L_\psi\|\theta^i_k - \theta^i_{k-z}\|, \tag{68}
\end{aligned}
$$

where the first inequality applies Lemma 6, and the last inequality uses the derivation in (67).

Combine (67) and (68), we have

$$
\begin{aligned}
I_1 &\leq |\mathcal{A}|L(\log_\rho \kappa^{-1} + (1-\rho)^{-1})C_\theta\|\theta_k - \theta_{k-z}\| \\
&\quad + 2(1+\gamma)L_V\|\theta_k - \theta_{k-z}\| + 2C_\delta L_\psi\|\theta^i_k - \theta^i_{k-z}\| \tag{69}
\end{aligned}
$$

We now bound $I_2$ as

$$
\begin{aligned}
\mathbb{E}[I_2] &= \mathbb{E}\|\delta(\tilde{\xi}_k, \theta_{k-z})\psi^i_{\theta_{k-z}} - \delta(\xi_k, \theta_{k-z})\psi^i_{\theta_{k-z}}\| \\
&\leq 2\sup_{\xi}\|\delta(\xi, \theta_{k-z})\psi^i_{\theta^i_{k-z}}\|d_{TV}(P(\tilde{\xi}_k \in \cdot|\theta_{k-z}, s_{k-z}), P(\xi_k \in \cdot|\theta_{k-z}, s_{k-z})) \\
&\leq 2C_\theta \sum_{m=0}^{z-1}|\mathcal{A}|L_\pi\|\theta_{k-m} - \theta_{k-z}\|,
\end{aligned}
\tag{70}
$$

where the last inequality follows Lemma 9.

$I_3$ can be bounded as

$$
\begin{aligned}
I_3 &= \mathbb{E}\|\mathbb{E}_{\xi \sim \mu_{\theta_{k-z}}}[\delta(\xi, \theta_{k-z})\psi^i_{k-z}] - \delta(\tilde{\xi}_k, \theta_{k-z}\psi^i_{\theta_{k-z}})\| \\
&\leq 2\sup_{\xi}\|\delta(\xi, \theta_{k-z})\psi^i_{\theta_{k-z}}\|d_{TV}(P(\tilde{\xi} \in \cdot|\theta_{k-z}, s_{k-z}), \mu_{\theta_{k-z}} \otimes \pi_{\theta_{k-z}} \otimes \mathcal{P}) \\
&\leq 2C_\theta\kappa\rho^{z-1},
\end{aligned}
\tag{71}
$$

where the last inequality follows Lemma 7.

Plug (69), (70), and (71), we have

$$
\begin{aligned}
&\|\mathbb{E}_{\xi \sim \mu_{\theta_k}}[\delta(\xi, \theta_k)\psi_{\theta^i_k}(s_k, a^i_k)] - \mathbb{E}[\delta(\xi_k, \theta_k)\psi_{\theta^i_k}(s_k, a^i_k)]\| \\
&\leq 2C_\theta\kappa\rho^{z-1} + 2C_\delta L_\psi\|\theta^i_k - \theta^i_{k-z}\| + 2C_\theta\sum_{m=0}^{z-1}|\mathcal{A}|L_\pi\|\theta_{k-m} - \theta_{k-z}\| \\
&\quad + (|\mathcal{A}|L(\log_\rho\kappa^{-1} + (1-\rho)^{-1})C_\theta + 2(1+\gamma)L_V)\|\theta_k - \theta_{k-z}\|,
\end{aligned}
$$

which completes the proof.

$\square$

# D   Proof of main results

## D.1   Proof of Theorem 1

Let $\theta_k \in \mathbb{R}^{Nd_\theta}$ be the stack of actors' parameter at timestep $k$. By Lemma 4, we have

$$
\begin{aligned}
\mathbb{E}[J(\theta_{k+1})] - J(\theta_k) &\geq \mathbb{E}[\langle \nabla J(\theta_k), \theta_{k+1} - \theta_k \rangle] - \frac{L}{2}\|\theta_{k+1} - \theta_k\|^2 \\
&= \sum_{i=1}^{N}\mathbb{E}[\langle \nabla_{\theta^i}J(\theta_k), \theta^i_{k+1} - \theta^i_k \rangle] - \frac{L}{2}\sum_{i=1}^{N}\|\theta^i_{k+1} - \theta^i_k\|^2 \\
&= \sum_{i=1}^{N}\mathbb{E}[\alpha_k\langle \nabla_{\theta^i}J(\theta_k), g^i_a(\xi_k, \omega^i_{k+1}, \lambda^i_{k+1})\rangle] - \frac{L}{2}\alpha^2_k\sum_{i=1}^{N}\mathbb{E}\|g^i_a(\xi_k, \omega^i_{k+1}, \lambda^i_{k+1})\|^2 \\
&\geq \sum_{i=1}^{N}\left[\frac{\alpha_k}{2}\|\nabla_{\theta^i}J(\theta_k)\|^2 + \frac{\alpha_k}{2}\|\mathbb{E}[g^i_a(\xi_k, \omega^i_{k+1}, \lambda^i_{k+1})]\|^2 \right. \\
&\quad \left. - \frac{\alpha_k}{2}\|\nabla_{\theta^i}J(\theta_k) - \mathbb{E}[g^i_a(\xi_k, \omega^i_{k+1}, \lambda^i_{k+1})]\|^2\right] - \frac{L}{2}NC^2_\theta\alpha^2_k,
\end{aligned}
\tag{72}
$$

where the expectation is taken over $\xi_k$ under Markovian sampling. The last inequality is due to

$$
\|g^i_a(\xi_k, \omega^i_{k+1}, \lambda^i_{k+1})\| = \left\|\hat{\delta}(\xi_k, \omega^i_k, \lambda^i_k)\psi_{\theta^i_k}(s_k, a^i_k)\right\| \leq C_\delta C_\psi := C_\theta.
\tag{73}
$$

For brevity, we will use $\psi_{\theta_k^i}$ to represent $\psi_{\theta_k^i}(s_k, a_k^i)$. The gradient bias can be bounded as

$$
\left\| \nabla_{\theta^i} J(\theta_k) - \mathbb{E}\left[ g_a^i(\xi_k, \omega_{k+1}^i, \lambda_{k+1}^i) | \omega_{k+1}^i, \lambda_{k+1}^i \right] \right\|^2
$$

$$
\leq 4 \underbrace{\left\| \nabla_{\theta^i} J(\theta_k) - \mathbb{E}\left[ \delta(\xi_k, \theta_k) \psi_{\theta_k^i} \right] \right\|^2}_{I_1} + 4 \underbrace{\left\| \mathbb{E}\left[ (\delta(\xi_k, \theta_k) - \tilde{\delta}(\xi_k, \omega^*(\theta_k))) \psi_{\theta_k^i} \right] \right\|^2}_{I_2}
$$

$$
+ 4 \underbrace{\left\| \mathbb{E}\left[ (\tilde{\delta}(\xi_k, \omega^*(\theta_k)) - \tilde{\delta}(\xi_k, \omega_{k+1}^i)) \psi_{\theta_k^i} \right] \right\|^2}_{I_3} + 4 \underbrace{\left\| \mathbb{E}\left[ (\tilde{\delta}(\xi_k, \omega_{k+1}^i) - \hat{\delta}(\xi_k, \omega_{k+1}^i, \lambda_{k+1}^i)) \psi_{\theta_k^i} \right] \right\|^2}_{I_4}, \quad (74)
$$

where the inequality uses $\|a + b + c + c\|^2 \leq 4\|a\|^2 + 4\|b\|^2 + 4\|c\|^2 + 4\|d\|^2$.

We bound $I_1$ as

$$
I_1 = \left\| \nabla_{\theta^i} J(\theta_k) - \mathbb{E}\left[ \delta(\xi_k, \theta_k) \psi_{\theta_k^i} | \theta_k \right] \right\|^2
$$

$$
= \left\| \mathbb{E}_{\xi \sim d_{\theta_k}}\left[ \delta(\xi, \theta_k) \psi_{\theta_k^i} | \theta_k \right] - \mathbb{E}\left[ \delta(\xi_k, \theta_k) \psi_{\theta_k^i} | \theta_k \right] \right\|^2
$$

$$
\leq 2 \underbrace{\left\| \mathbb{E}_{\xi \sim d_{\theta_k}}\left[ \delta(\xi, \theta_k) \psi_{\theta_k^i} | \theta_k \right] - \mathbb{E}_{\xi \sim \mu_{\theta_k}}\left[ \delta(\xi, \theta_k) \psi_{\theta_k^i} | \theta_k \right] \right\|^2}_{I_1^{(1)}} + 2 \underbrace{\left\| \mathbb{E}_{\xi \sim \mu_\theta}\left[ \delta(\xi, \theta_k) \psi_{\theta_k^i} | \theta_k \right] - \mathbb{E}\left[ \delta(\xi_k, \theta_k) \psi_{\theta_k^i} | \theta_k \right] \right\|^2}_{I_1^{(2)}}
$$

$$
(75)
$$

From now on, we will use $\xi \sim d_\theta$ to denote $s \sim d_{\pi_\theta}, a \sim \pi(\cdot|s), s' \sim \mathcal{P}$ for notational simplicity. $I_1$ is the sampling error under perfect value function estimation of critic. It can be bounded as

$$
\mathbb{E}\left[ I_1^{(1)} | \theta_k \right] = \left\| \nabla_{\theta^i} J(\theta_k) - \mathbb{E}\left[ \delta(\xi_k, \theta_k) \psi_{\theta_k^i} | \theta_k \right] \right\|^2
$$

$$
= \left\| \mathbb{E}_{\xi \sim d_{\theta_k}}\left[ \delta(\xi, \theta_k) \psi_{\theta_k^i} | \theta_k \right] - \mathbb{E}_{\xi \sim \mu_{\theta_k}}\left[ \delta(\xi, \theta_k) \psi_{\theta_k^i} | \theta_k \right] \right\|^2
$$

$$
\leq \left( 2 \sup_\xi \left\| \delta(\xi, \theta_k) \psi_{\theta_k^i} \right\| \, d_{TV}(\mu_{\theta_k} \otimes \pi_{\theta_k} \otimes \mathcal{P}, d_{\theta_k} \otimes \pi_{\theta_k} \otimes \mathcal{P}) \right)^2
$$

$$
\overset{(i)}{\leq} (2C_\theta d_{TV}(\mu_{\theta_k}, d_{\theta_k}))^2
$$

$$
\overset{(ii)}{\leq} 16C_\theta^2 (\log_\rho \kappa^{-1} + \frac{1}{\rho})^2 (1 - \gamma^2),
$$

where $(i)$ uses (73); $(ii)$ follows Lemma 8. Define $\varepsilon_{sp} := 4C_\theta^2 (\log_\rho \kappa^{-1} + \frac{1}{\rho})^2 (1 - \gamma)^2$, then we have

$$
I_1^{(1)} \leq 4\varepsilon_{sp}. \quad (76)
$$

By Lemma 20, $I_1^{(2)}$ can be bounded as

$$
I_1^{(2)} \leq \left( 2C_\theta \kappa \rho^{z-1} + C_{12} \sum_{m=0}^{z-1} \|\theta_{k-m} - \theta_{k-z}\| + C_{13} \|\theta_k - \theta_{k-z}\| + C_{14} \|\theta_k^i - \theta_{k-z}^i\| \right)^2
$$

$$
\leq \left( 2C_\theta \kappa \rho^{z-1} + C_{12} \sum_{m=0}^{z-1} \sum_{n=1}^{z-m} \|\theta_{k-m-n+1} - \theta_{k-m}\| + C_{13} \sum_{n=1}^{z} \|\theta_{k-n+1} - \theta_{k-n}\| + C_{14} \sum_{n=1}^{z} \|\theta_{k-n+1}^i - \theta_{k-n}^i\| \right)^2
$$

$$
\leq \left( 2C_\theta \kappa \rho^{z-1} + C_{12} N C_\theta \frac{z(z+1)}{2} \alpha_{k-z} + C_{13} N z C_\theta \alpha_{k-z} + C_{14} z C_\theta \alpha_{k-z} \right)^2
$$

$$
\leq 16C_\theta^2 \kappa^2 \rho^{2z-2} + 2C_{12}^2 C_\theta^2 z^2 \alpha_{k-z}^2 + 4C_{13}^2 N^2 z^2 C_\theta^2 \alpha_{k-z}^2 + 4C_{14}^2 z^2 C_\theta^2 \alpha_{k-z}^2, \quad (77)
$$

where the second inequality uses triangle inequality, and the last inequality applies $(a + b + c + d)^2 \leq 4a^2 + 4b^2 + 4c^2 + 4d^2$. Let $z = Z_K := \min\{z \in \mathbb{N}^+ | \kappa \rho^{z-1} \leq \min\{\alpha_K, \beta_K, \eta_K\}\}$. Then we have

$$I_1^{(2)} \leq C_{K_5} \alpha_{k-Z_K}^2, \tag{78}$$

where we define $C_{K_5} := 16C_\theta^2 + 2C_{12}^2 C_\theta^2 Z_K^2 + 4C_{13}^2 N^2 Z_K^2 C_\theta^2 + 4C_{14}^2 Z_K^2 C_\theta^2$. Thus, we have

$$I_1 \leq 4\varepsilon_{sp} + C_{K_5} \alpha_{k-Z_K}^2. \tag{79}$$

The term $I_2$ describes the approximation quality of linear function class, it can be bounded as

$$
\begin{aligned}
I_2 &= \left\| \mathbb{E}\left[ (\delta(\xi_k, \theta_k) - \tilde{\delta}(\xi_k, \omega^*(\theta_k))) \psi_{\theta_k^i} \right] \right\|^2 \\
&\overset{(i)}{\leq} \mathbb{E}\left[ \left| \delta(\xi_k, \theta_k) - \tilde{\delta}(\xi_k, \omega^*(\theta_k)) \right|^2 \left\| \psi_{\theta_k^i} \right\|^2 \right] \\
&\overset{(ii)}{\leq} C_\psi^2 \mathbb{E}\left[ \gamma \left| V_{\pi_{\theta_k}}(s_{k+1}) - \hat{V}_{\omega^*(\theta_k)}(s_{k+1}) \right| + \left| V_{\pi_{\theta_k}}(s_k) - \hat{V}_{\omega^*(\theta_k)}(s_k) \right| \right] \\
&\overset{(iii)}{\leq} 2C_\psi^2 \left( \gamma^2 \mathbb{E}\left[ \left| V_{\theta_k}(s_{k+1}) - \hat{V}_{\omega^*(\theta_k)}(s_{k+1}) \right|^2 \right] + \mathbb{E}\left[ \left| V_{\theta_k}(s_k) - \hat{V}_{\omega^*(\theta_k)}(s_k) \right|^2 \right] \right) \\
&\overset{(iiii)}{\leq} 2C_\psi^2 (1 + \gamma^2) \varepsilon_{app}^c \leq 4C_\psi^2 \varepsilon_{app}^c.
\end{aligned} \tag{80}
$$

where $(i)$ applies Cauchy Schwarz inequality and triangle inequality; $(ii)$ is due to $\|\psi_{\theta_k^i}\| \leq C_\psi$, which is ensured by Assumption 3; $(iii)$ uses $|a + b|^2 \leq 2|a|^2 + 2|b|^2$; $(iiii)$ follows the definition of the critic's approximation error:

$$\varepsilon_{app}^c := \max_\theta \sqrt{\mathbb{E}_{s \sim \mu_\theta}\left[ \left| V_{\pi_\theta}(s) - \hat{V}_{\omega^*(\theta)}(s) \right|^2 \right]}. \tag{81}$$

$I_3$ captures the error of critic's estimator, which can be bounded as

$$
\begin{aligned}
\mathbb{E}[I_3] &= \left\| \mathbb{E}\left[ \left( \tilde{\delta}(\xi_k, \omega^*(\theta_k)) - \tilde{\delta}(\xi_k, \omega_{k+1}^i) \right) \psi_{\theta_k^i} \right] \right\|^2 \\
&\leq \mathbb{E}\left[ \left| \tilde{\delta}(\xi_k, \omega^*(\theta_k)) - \tilde{\delta}(\xi_k, \omega_{k+1}^i) \right|^2 \left\| \psi_{\theta_k^i} \right\|^2 \right] \\
&\leq C_\psi^2 \mathbb{E}\left[ \left| \gamma \phi(s_{k+1})^T \left( \omega^*(\theta_k) - \omega_{k+1}^i \right) - \phi(s_k)^T \left( \omega^*(\theta_k) - \omega_{k+1}^i \right) \right|^2 \right] \\
&\leq C_\psi^2 \left( 2\mathbb{E}\left[ \left| \gamma \phi(s_{k+1})^T \left( \omega^*(\theta_k) - \omega_{k+1}^i \right) \right|^2 \right] + 2\mathbb{E}\left[ \left| \phi(s_k)^T \left( \omega^*(\theta_k) - \omega_{k+1}^i \right) \right|^2 \right] \right) \\
&\leq C_\psi^2 \left( 2\gamma^2 \mathbb{E}\left[ \|\phi(s_{k+1})\|^2 \left\| \omega^*(\theta_k) - \omega_{k+1}^i \right\|^2 \right] + 2\mathbb{E}\left[ \|\phi(s_k)\|^2 \left\| \omega^*(\theta_k) - \omega_{k+1}^i \right\|^2 \right] \right) \\
&\leq 2C_\psi^2 (1 + \gamma^2) \left\| \omega^*(\theta_k) - \omega_{k+1}^i \right\|^2 \leq 4C_\psi^2 \left\| \omega^*(\theta_k) - \omega_{k+1}^i \right\|^2,
\end{aligned} \tag{82}
$$

where the last inequality is due to $\|\phi(s)\| \leq 1$, which is specified by Assumption 1.

$I_4$ characterizes the error of reward estimator, which can be bounded as

$$
\begin{aligned}
\mathbb{E}[I_4] &= \left\| \mathbb{E}\left[ \left( \tilde{\delta}(\xi_k, \omega_{k+1}^i) - \hat{\delta}(\xi_k, \omega_{k+1}^i, \lambda_{k+1}^i) \right) \psi_{\theta_k^i} | \lambda_{k+1}^i \right] \right\|^2 \\
&\leq \mathbb{E}\left[ \left| \tilde{\delta}(\xi_k, \omega_{k+1}^i) - \hat{\delta}(\xi_k, \omega_{k+1}^i, \lambda_{k+1}^i) \right|^2 \left\| \psi_{\theta_k^i} \right\|^2 | \lambda_{k+1}^i \right] \\
&\leq C_\psi^2 \mathbb{E}\left[ \left| \bar{r}(s_k, a_k) - \varphi(s_k, a_k)^T \lambda_{k+1}^i \right|^2 | \lambda_{k+1}^i \right] \\
&\leq C_\psi^2 \left( 2\mathbb{E}\left[ \left| \bar{r}(s_k, a_k) - \varphi(s_k, a_k)^T \lambda^*(\theta_k) \right|^2 \right] + 2\mathbb{E}\left[ \left| \varphi(s_k, a_k)^T \lambda^*(\theta_k) - \varphi(s_k, a_k)^T \lambda_{k+1}^i \right|^2 | \lambda_{k+1}^i \right] \right) \\
&\leq 2C_\psi^2 \varepsilon_{app}^r + 2C_\psi^2 \left\| \lambda^*(\theta_k) - \lambda_{k+1}^i \right\|^2,
\end{aligned} \tag{83}
$$

where the $\varepsilon_{app}^r$ in the last inequality is the approximation error of reward estimator, which is defined as

$$\varepsilon_{app}^r := \max_{\theta,a} \sqrt{\mathbb{E}_{s\sim\mu_\theta}\left[\left|\bar{r}(s,a) - \hat{r}_{\lambda^*(\theta)}(s,a)\right|^2\right]}.$$

Combining (79), (80), (82), and (83) gives us the bound of the gradient bias error as

$$\|\nabla_{\theta^i}F(\theta_k) - \mathbb{E}[g_a^i(\xi_k,\omega_{k+1}^i,\lambda_{k+1}^i)]\|^2 \leq 16(\varepsilon_{sp} + C_\psi^2\varepsilon_{app}) + 16C_\psi^2\|\omega^*(\theta_k) - \omega_{k+1}^i\|^2$$
$$+ 8C_\psi^2\|\lambda^*(\theta_k) - \lambda_{k+1}^i\|^2 + 4C_{K_5}\alpha_{k-Z_K}^2. \tag{84}$$

Plug (84) into (72), we get

$$\mathbb{E}[J(\theta_{k+1})] - J(\theta_k) \geq \sum_{i=1}^N \left(\frac{\alpha_k}{2}\mathbb{E}\|\nabla_{\theta^i}J(\theta_k)\|^2 + \frac{\alpha_k}{2}\mathbb{E}\|g_a^i(\xi_k,\omega_{k+1}^i,\lambda_{k+1}^i)\|^2 \right.$$
$$\left. -8C_\psi^2\alpha_k\mathbb{E}\|\omega^*(\theta_k) - \omega_{k+1}^i\|^2 - 4C_\psi^2\alpha_k\mathbb{E}\|\lambda^*(\theta_k) - \lambda_{k+1}^i\|^2\right)$$
$$- \frac{L}{2}NC_\theta^2\alpha_k^2 - 2NC_{K_5}\alpha_{k-Z_K}^2 - 8(\varepsilon_{sp} + C_\psi^2\varepsilon_{app})N\alpha_k. \tag{85}$$

Consider the Lyapunov function

$$\mathbb{V}_k := -J(\theta_k) + \|\bar{\omega}_k - \omega^*(\theta_k)\|^2 + \|\bar{\lambda}_k - \lambda^*(\theta_k)\|^2. \tag{86}$$

The difference between two Lyapunov functions will be

$$\mathbb{E}[\mathbb{V}_{k+1}] - \mathbb{E}[\mathbb{V}_k] = \mathbb{E}[J(\theta_k)] - \mathbb{E}[J(\theta_{k+1})] + \mathbb{E}\|\bar{\omega}_{k+1} - \omega^*(\theta_{k+1})\|^2 - \mathbb{E}\|\bar{\omega}_k - \omega^*(\theta_k)\|^2$$
$$+ \mathbb{E}\|\bar{\lambda}_{k+1} - \lambda^*(\theta_k)\|^2 - \mathbb{E}\|\bar{\lambda}_k - \lambda^*(\theta_k)\|^2$$
$$\leq \sum_{i=1}^N \left(-\frac{\alpha_k}{2}\|\nabla_{\theta^i}J(\theta_k)\|^2 - \frac{\alpha_k}{2}\mathbb{E}\|g_a^i(\xi_k,\omega_{k+1}^i)\|^2\right)$$
$$+ 2NC_{K_5}\alpha_{k-Z_K} + \frac{L}{2}NC_\theta^2\alpha_k^2 + 8(\varepsilon_{sp} + C_\psi^2\varepsilon_{app})N\alpha_k$$
$$+ \underbrace{\sum_{i=1}^N 8C_\psi^2\alpha_k\mathbb{E}\|\omega^*(\theta_k) - \omega_{k+1}^i\|^2 + \mathbb{E}\|\bar{\omega}_{k+1} - \omega^*(\theta_{k+1})\|^2 - \mathbb{E}\|\bar{\omega}_k - \omega^*(\theta_k)\|^2}_{I_5}$$
$$+ \underbrace{\sum_{i=1}^N 4C_\psi^2\alpha_k\mathbb{E}\|\lambda^*(\theta_k) - \lambda_{k+1}^i\|^2 + \mathbb{E}\|\bar{\lambda}_{k+1} - \lambda^*(\theta_{k+1})\|^2 - \mathbb{E}\|\bar{\lambda}_k - \lambda^*(\theta_k)\|^2}_{I_6}. \tag{87}$$

The first two terms of $I_5$ can be bounded as

$$\sum_{i=1}^N 8C_\psi^2\alpha_k\mathbb{E}\|\omega^*(\theta_k) - \bar{\omega}_{k+1} + \bar{\omega}_{k+1} - \omega_{k+1}^i\|^2 + \mathbb{E}\|\bar{\omega}_{k+1} - \omega^*(\theta_{k+1})\|^2$$
$$= \sum_{i=1}^N 8C_\psi^2\alpha_k\mathbb{E}\|\bar{\omega}_{k+1} - \omega_{k+1}^i\|^2 + 8C_\psi^2\alpha_k\mathbb{E}\|\bar{\omega}_{k+1} - \omega^*(\theta_k)\|^2 + \mathbb{E}\|\bar{\omega}_{k+1} - \omega^*(\theta_{k+1})\|^2$$
$$\leq 8C_\psi^2\alpha_k\mathbb{E}\|\bar{\omega}_{k+1} - \omega^*(\theta_k)\|^2 + \mathbb{E}\|\bar{\omega}_{k+1} - \omega^*(\theta_{k+1})\|^2 + \alpha_k M_{k_1}$$
$$\leq \left(1 + 4L_\omega N\alpha_k + 8C_\psi^2\alpha_k + \frac{L_{\omega,2}^2}{2}C_\theta^2 N\sqrt{d_\theta}\alpha_k^2\right)\mathbb{E}\|\bar{\omega}_{k+1} - \omega^*(\theta_k)\|^2$$
$$+ \left(\frac{L_{\omega,2}^2 C_\theta^2 N}{2} + L_\omega^2 C_\theta^2 N\right)\alpha_k^2 + \frac{\alpha_k}{4}\sum_{i=1}^N \left\|\mathbb{E}\left[g_a^i(\xi_k,\omega_{k+1}^i,\lambda_{k+1}^i)\right]\right\|^2 + \alpha_k M_{k_1}, \tag{88}$$

where the equality is due to

$$\sum_{i=1}^{N} \left\langle \omega^*(\theta_k) - \bar{\omega}_{k+1}, \bar{\omega}_{k+1} - \omega_{k+1}^i \right\rangle = \left\langle \omega^*(\theta_k) - \bar{\omega}_{k+1}, \bar{\omega}_{k+1} - \bar{\omega}_{k+1} \right\rangle = 0.$$

The first inequality follows the Lemma 19, where $M_{k_1}$ is defined as

$$M_{k_1} := \nu^{\frac{2k}{K_c} - 2} \|\boldsymbol{\omega}_0\|^2 + 16 N C_\delta^2 \left( \sum_{t=0}^{k} \beta_t \nu^{\frac{k-t}{K_c} - 1} \right)^2 + 8\sqrt{N} C_\delta \nu^{\frac{k}{K_c} - 1} \sum_{t=0}^{k} \beta_t \nu^{\frac{k-t}{K_c} - 1}.$$

The last inequality follows (32) in Lemma 15.

Plug (88) into (87), and define $C_9 := \min\left\{ c \mid 4 L_\omega N \alpha_k + 8 C_\psi^2 \alpha_k + \frac{L_{\omega,2}^2}{2} C_\theta^2 N \sqrt{d_\theta} \alpha_k^2 \leq c \alpha_k \right\}$, we get

$$I_5 \leq (1 + C_9 \alpha_k) \mathbb{E} \|\bar{\omega}_{k+1} - \omega^*(\theta_k)\|^2 + \left( \frac{L_{\omega,2}^2 C_\theta^2 N^2}{2} + L_\omega^2 C_\theta^2 N \right) \alpha_k^2$$

$$+ \frac{\alpha_k}{4} \sum_{i=1}^{N} \|\mathbb{E}[g_a^i(\xi_k, \omega_{k+1}^i, \lambda_{k+1}^i)]\|^2 + \alpha_k M_{k_1}$$

$$\leq [(1 + C_9 \alpha_k)(1 - 2\lambda_\phi \beta_k) - 1] \mathbb{E} \|\bar{\omega}_k - \omega^*(\theta_k)\|^2$$

$$+ (1 + C_9 \alpha_k)(C_{K_1} \beta_k \beta_{k-Z_K} + C_{K_2} \beta_k \alpha_{k-Z_K})$$

$$+ \left( \frac{L_{\omega,2}^2 C_\theta^2 N}{2} + L_\omega^2 C_\theta^2 N \right) \alpha_k^2 + \frac{\alpha_k}{4} \sum_{i=1}^{N} \left\| \mathbb{E} \left[ g_a^i \left( \xi_k, \omega_{k+1}^i, \lambda_{k+1}^i \right) \right] \right\|^2 + \alpha_k M_{k_1}, \quad (89)$$

where the last inequality follows (33) in Lemma 15.

By letting $\beta_k = \frac{C_9}{2\lambda_\phi} \alpha_k$, we can ensure

$$(1 + C_9 \alpha_k)(1 - 2\lambda_\phi \beta_k) < 1.$$

Therefore, $I_5$ can be bounded as

$$I_5 \leq \frac{\alpha_k}{4} \sum_{i=1}^{N} \|\mathbb{E}[g_a^i(\xi_k, \omega_{k+1}^i, \lambda_{k+1}^i)]\|^2 + \alpha_k M_{k_1} + \left( \frac{L_{\omega,2}^2 C_\theta^2 N^2}{2} + L_\omega^2 C_\theta^2 N \right) \alpha_k^2$$

$$+ (1 + C_9 \alpha_k)(C_{K_1} \beta_k \beta_{k-Z_K} + C_{K_2} \beta_k \alpha_{k-Z_K}). \quad (90)$$

By applying Lemma 17 and following the similar procedure, we can bound $I_6$ as

$$I_6 \leq \frac{\alpha_k}{4} \sum_{i=1}^{N} \|\mathbb{E}[g_a^i(\xi_k, \omega_{k+1}^i, \lambda_{k+1}^i)]\|^2 + \alpha_k M_{k_2} + \left( \frac{L_{\lambda,2}^2 C_\theta^2 N^2}{2} + L_\lambda^2 C_\theta^2 N \right) \alpha_k^2$$

$$+ (1 + C_{10} \alpha_k)(C_{K_3} \eta_k \eta_{k-Z_K} + C_{K_4} \eta_k \alpha_{k-Z_K}). \quad (91)$$

with $\eta_k = \frac{C_{10}}{2\lambda_\varphi} \alpha_k$. $C_{10}$ and $M_{k_2}$ are defined as

$$C_{10} := \min\left\{ c \mid 4 L_\lambda N \alpha_k + 4 C_\psi^2 \alpha_k + \frac{L_{\lambda,2}^2}{2} C_\delta^2 N \sqrt{d_\theta} \alpha_k^2 \leq c \alpha_k \right\},$$

$$M_{k_2} := \nu^{\frac{2k}{K_c} - 2} \|\boldsymbol{\lambda}_0\|^2 + 16 N C_\lambda^2 \left( \sum_{t=0}^{k} \eta_t \nu^{\frac{k-t}{K_c} - 1} \right)^2 + 8\sqrt{N} C_\lambda \nu^{\frac{k}{K_c} - 1} \sum_{t=0}^{k} \eta_t \nu^{\frac{k-t}{K_c} - 1}. \quad (92)$$

Plug (90) and (91) into (87), we have

$$
\begin{aligned}
\mathbb{E}[\mathbb{V}_{k+1}] - \mathbb{E}[\mathbb{V}_k] \leq & \sum_{i=1}^{N} -\frac{\alpha_k}{2}\|\nabla_{\theta^i} J(\theta_k)\|^2 + (M_{k_1} + M_{k_2})\alpha_k \\
& + (1 + C_9\alpha_k)(C_{K_1}\beta_k\beta_{k-Z_K} + C_{K_2}\beta_k\alpha_{k-Z_K}) \\
& + (1 + C_{10}\alpha_k)(C_{K_3}\eta_k\eta_{k-Z_K} + C_{K_4}\eta_k\alpha_{k-Z_K}) \\
& + \left(\frac{L}{2}NC_\theta^2 + C_{11}\right)\alpha_k^2 + 8(\varepsilon_{sp} + C_\psi^2\varepsilon_{app}N)\alpha_k,
\end{aligned}
\tag{93}
$$

where $C_{11} := C_\theta^2 N(\frac{L_{\omega,2}^2 + L_{\lambda,2}^2}{2} + L_\omega^2 + L_\lambda^2)$.

By letting $\alpha_k = \frac{\bar{\alpha}}{\sqrt{K}}$ for some positive constant $\bar{\alpha}$, and recall $\beta_k = \frac{C_9}{2\lambda_\phi}\alpha_k, \eta_k = \frac{C_{10}}{2\lambda_\varphi}\alpha_k$, we can telescope (93) as

$$
\begin{aligned}
\frac{1}{K}\sum_{k=0}^{K}\sum_{i=1}^{N}\mathbb{E}\|\nabla_{\theta^i} J(\theta_k)\|^2 \leq & \frac{2\mathbb{E}[\mathbb{V}_0]}{K\alpha_k} + 16(\varepsilon_{sp} + C_\psi^2\varepsilon_{app}N) + \frac{2}{K}\sum_{k=0}^{K}(M_{k_1} + M_{k_2}) \\
& + (2 + 2C_9\alpha_k)(C_{K_1}\frac{\beta_k}{\alpha_k}\beta_{k-Z_K} + C_{K_2}\frac{\beta_k}{\alpha_k}\alpha_{k-Z_K}) \\
& + (2 + 2C_{10}\alpha_k)(C_{K_3}\frac{\eta_k}{\alpha_k}\eta_{k-Z_K} + C_{K_4}\frac{\eta_k}{\alpha_k}\alpha_{k-Z_K}) \\
& + \left(LNC_\theta^2 + 2C_{11}\right)\alpha_k.
\end{aligned}
\tag{94}
$$

The summation of $M_{k_1}$ can be bounded as

$$
\begin{aligned}
\sum_{k=0}^{K}M_{k_1} &= \sum_{k=0}^{K}\left(\nu^{\frac{2k}{K_c}-2}\|\boldsymbol{\omega}_0\|^2 + 16NC_\delta^2\left(\sum_{t=0}^{k}\beta_t\nu^{\frac{k-t}{K_c}-1}\right)^2 + 8\sqrt{N}C_\delta\nu^{\frac{k}{K_c}-1}\sum_{t=0}^{k}\beta_t\nu^{\frac{k-t}{K_c}-1}\right) \\
&= \sum_{k=0}^{K}\left(\nu^{\frac{2k}{K_c}-2}\|\boldsymbol{\omega}_0\|^2 + 16NC_\delta^2\beta_k^2\left(\sum_{t=0}^{k}\nu^{\frac{k-t}{K_c}-1}\right)^2 + 8\sqrt{N}C_\delta\nu^{\frac{k}{K_c}-1}\beta_k\sum_{t=0}^{k}\nu^{\frac{k-t}{K_c}-1}\right) \\
&\overset{(i)}{\leq} \sum_{k=0}^{K}\left(\nu^{\frac{2k}{K_c}-2}\|\boldsymbol{\omega}_0\|^2 + 16NC_\delta^2\beta_k^2\frac{K_c^2}{\nu^2(1-\nu)^2} + 8\sqrt{N}C_\delta\nu^{\frac{k}{K_c}-1}\beta_k\frac{K_c}{\nu(1-\nu)}\right) \\
&\overset{(ii)}{\leq} \frac{\|\boldsymbol{\omega}_0\|^2}{\nu^2(1-\nu^{2/K_c})^2} + \frac{16NC_\delta^2\beta_k^2 K_c^2 K}{\nu^2(1-\nu)^2} + 8\sqrt{N}C_\delta\beta_k\frac{K_c}{(1-\nu^{K_c})\nu(1-\nu)} \\
&= \mathcal{O}(\beta_k^2 K_c^2) = \mathcal{O}(\sqrt{K}),
\end{aligned}
\tag{95}
$$

where the second equality is according to the step size choice. $(i)$ is due to

$$
\sum_{t=0}^{k}\nu^{\frac{k-t}{K_c}-1} \leq K_c\sum_{z=0}^{\lceil\frac{k}{K_c}\rceil}\nu^{z-1} \leq K_c\frac{1}{\nu(1-\nu)}.
$$

$(ii)$ is due to $\sum_{k=0}^{K}\nu^{\frac{k}{K_c}-1} = \frac{1}{\nu(1-\nu^{1/K_c})}$. The last equality uses $K_c = \mathcal{O}(K^{1/4})$. By following similar arguments, we can show that $\sum_{k=0}^{K}M_{k_2} = \mathcal{O}(\sqrt{K})$. Therefore, the third term in (94) is of order $\mathcal{O}(\frac{1}{\sqrt{K}})$.

Finally, by noticing $C_{K_1} = \mathcal{O}(\log\frac{1}{\alpha_k}), C_{K_2} = \mathcal{O}(\log^2\frac{1}{\alpha_k}), C_{K_3} = \mathcal{O}(\log\frac{1}{\alpha_k}), C_{K_4} = \mathcal{O}(\log^2\frac{1}{\alpha_k})$, we obtain the desired iteration complexity of $\tilde{\mathcal{O}}(\frac{1}{\sqrt{K}})$, or equivalently, the sample complexity of $\tilde{\mathcal{O}}(\varepsilon^{-2})$.

## D.2 Proof of Theorem 2

Define the update of actor $i$ using the noisy reward as

$$
g_a^i(\xi_k, \omega_{k+1}^i) := \tilde{r}_{k,K_r}^i(s_k, a_k) + \gamma\phi(s')^T\omega_{k+1}^i - \phi(s)^T\omega_{k+1}^i.
\tag{96}
$$

Following the derivation of (72), we have

$$\mathbb{E}[J(\theta_{k+1}) - J(\theta_k) \geq \sum_{i=1}^{N} \left[ \frac{\alpha_k}{2} \|\nabla_{\theta^i} J(\theta_k)\|^2 + \frac{\alpha_k}{2} \|\mathbb{E}[g_a^i(\xi_k, \omega_{k+1}^i)]\|^2 \right.$$
$$\left. - \frac{\alpha_k}{2} \|\nabla_{\theta^i} J(\theta_k) - \mathbb{E}[g_a^i(\xi_k, \omega_{k+1}^i)]\|^2 \right] - \frac{L}{2} N C_\theta^2 \alpha_k^2. \tag{97}$$

Similarly to the proof of Theorem 1, the gradient bias term can be decomposed as as

$$\|\nabla_{\theta^i} J(\theta_k) - \mathbb{E}[g_a^i(\xi_k, \omega_{k+1}^i)]\|^2 \leq 4 \underbrace{\|\nabla_{\theta^i} J(\theta_k) - \mathbb{E}[\delta(\xi_k, \theta_k) \psi_{\theta_k^i}]\|^2}_{I_1}$$
$$+ 4 \underbrace{\|\mathbb{E}[(\delta(\xi_k, \theta_k) - \tilde{\delta}(\xi_k, \omega^*(\theta_k))) \psi_{\theta_k^i}]\|^2}_{I_2}$$
$$+ 4 \underbrace{\|\mathbb{E}[(\tilde{\delta}(\xi_k, \omega^*(\theta_k)) - \tilde{\delta}(\xi_k, \omega_{k+1}^i)) \psi_{\theta_k^i}]\|^2}_{I_3}$$
$$+ 4 \underbrace{\|\mathbb{E}[(\bar{r}_k(s_k, a_k) - \tilde{r}_{k,K_r}(s_k, a_k)) \psi_{\theta_k^i}]\|^2}_{I_4} \tag{98}$$

$I_1$, $I_2$, $I_3$ can be bounded following the derivation of (84), (80), and (82), respectively. Plug these bounds into (97), we have

$$\mathbb{E}[J(\theta_{k+1})] - J(\theta_k) \geq \sum_{i=1}^{N} \left( \frac{\alpha_k}{2} \mathbb{E}\|\nabla_{\theta^i} J(\theta_k)\|^2 + \frac{\alpha_k}{2} \mathbb{E}\|g_a^i(\xi_k, \omega_{k+1}^i)\|^2 - 8C_\psi^2 \alpha_k \mathbb{E}\|\omega^*(\theta_k) - \omega_{k+1}^i\|^2 \right)$$
$$- \sum_{i=1}^{N} \frac{\alpha_k}{2} C_\psi^2 \|\bar{r}_k(s_k, a_k) - \tilde{r}_{k,K_r}^i(s_k, a_k)\|^2 - \frac{L}{2} N C_\theta^2 \alpha_k^2$$
$$- 2N C_{K_5} \alpha_{k-Z_K}^2 - 8(\varepsilon_{sp} + C_\psi^2 \varepsilon_{app}^r) N \alpha_k. \tag{99}$$

Define $\tilde{r}_{k,K_r} := [r_{k,K_r}^1, \cdots, r_{k,K_r}^N]^T$. The reward bias can be bounded as

$$\sum_{i=1}^{N} \|\bar{r}_k(s_k, a_k) - \tilde{r}_{k,K_r}^i(s_k, a_k)\|^2 = \|Q\tilde{r}_{k,K_r}\|^2$$
$$= \|QW^{K_r} \tilde{r}_{k,0}(s_k, a_k)\|^2$$
$$\leq \nu^{2K_r} \|\tilde{r}_{k,0}(s_k, a_k)\|^2$$
$$= \nu^{2K_r} \sum_{i=1}^{N} \left( \|\tilde{r}_{k,0}^i(s_k, a_k) - \bar{r}_k(s_k, a_k)\|^2 + \|\bar{r}_k(s_k, a_k)\|^2 \right)$$
$$\leq \nu^{2K_r} N(\sigma^2 + r_{\max}), \tag{100}$$

where $\sigma^2$ is the variance of the reward noise. Let $K_r = \frac{1}{2} \log_\nu \alpha_k$ and define $C_{15} := \sigma^2 + r_{\max}^2$. Plug (100) back to (99), we have

$$\mathbb{E}[J(\theta_{k+1})] - J(\theta_k) \geq \sum_{i=1}^{N} \left( \frac{\alpha_k}{2} \mathbb{E}\|\nabla_{\theta^i} J(\theta_k)\|^2 + \frac{\alpha_k}{2} \mathbb{E}\|g_a^i(\xi_k, \omega_{k+1}^i)\|^2 - 8C_\psi^2 \alpha_k \mathbb{E}\|\omega^*(\theta_k) - \omega_{k+1}^i\|^2 \right)$$
$$+ \frac{N}{2}(C_{15} + C_\theta^2 L)\alpha_k^2 - 2N C_{K_5} \alpha_{k-Z_K}^2 - 8(\varepsilon_{sp} + C_\psi^2 \varepsilon_{app}^r) N \alpha_k.$$

Consider the Lyapunov function

$$\mathbb{V}_k := -J(\theta_k) + \|\bar{\omega}_k - \omega^*(\theta_k)\|^2.$$

The difference between two Lyapunov functions is

$$
\mathbb{E}[\mathbb{V}_{k+1}] - \mathbb{E}[\mathbb{V}_k] \leq \sum_{i=1}^{N} \left( -\frac{\alpha_k}{2} \|\nabla_{\theta^i} J(\theta_k)\|^2 - \frac{\alpha_k}{2} \mathbb{E}\|g_a^i(\xi_k, \omega_{k+1}^i)\|^2 \right)
$$
$$
+ \frac{N}{2} C_{16} \alpha_k^2 - 2N C_{K_5} \alpha_{k-Z_K}^2 - 8(\varepsilon_{sp} + C_\psi^2 \varepsilon_{app}^r) N \alpha_k
$$
$$
+ \underbrace{\sum_{i=1}^{N} 8 C_\psi^2 \alpha_k \mathbb{E}\|\omega^*(\theta_k) - \omega_{k+1}^i\|^2 + \mathbb{E}\|\bar{\omega}_{k+1} - \omega^*(\theta_{k+1})\|^2 - \mathbb{E}\|\bar{\omega}_k - \omega^*(\theta_k)\|^2}_{I_5}.
$$

$I_5$ can be bounded by following the derivation of (90). Thus, we have

$$
\mathbb{E}[\mathbb{V}_{k+1}] - \mathbb{E}[\mathbb{V}_k]
$$
$$
\leq \sum_{i=1}^{N} -\frac{\alpha_k}{2} \|\nabla_{\theta^i} J(\theta_k)\|^2 + \frac{N}{2} C_{16} \alpha_k^2 - 2N C_{K_5} \alpha_{k-Z_K}^2 - 8(\varepsilon_{sp} + C_\psi^2 \varepsilon_{app}^r) N \alpha_k
$$
$$
+ (1 + C_9 \alpha_k)(C_{K_1} \beta_k \beta_{k-Z_K} + C_{K_2} \beta_k \alpha_{k-Z_K}) + M_{k_1} \alpha_k, \tag{101}
$$

where $C_{16} := C_{15} + C_\theta^2 L + \frac{L_{\omega,2}^2 C_\theta^2 N^2}{2} + L_\omega^2$.

Telescoping (101), we have

$$
\frac{1}{K} \sum_{k=0}^{K} \sum_{i=1}^{N} \mathbb{E}\|\nabla_{\theta^i} J(\theta_k)\|^2 \leq \frac{2\mathbb{E}[\mathbb{V}_0]}{K\alpha_k} + 16(\varepsilon_{sp} + C_\psi^2 \varepsilon_{app}^r N) + \frac{2}{K} \sum_{k=0}^{K} M_{k_1} + C_{16} \alpha_k
$$
$$
+ (1 + C_9 \alpha_k) \left( C_{K_1} \frac{\beta_k}{\alpha_k} \beta_{k-Z_K} + C_{K_2} \frac{\beta_k}{\alpha_k} \alpha_{k-Z_K} \right).
$$

The term $\frac{2}{K} \sum_{k=0}^{K} M_{k_1}$ has been bounded in (95). Let $\alpha_k = \frac{\bar{\alpha}}{\sqrt{K}}$ for some positive constant $\bar{\alpha}$, $\beta_k = \frac{C_9}{2\lambda_\phi} \alpha_k$ will yield the desired rate.

## E   Natural Actor-Critic variant and its convergence

In this section, we propose a natural Actor-Critic variant of Algorithm 1, where the approach of calculating the natural policy graident under the decentralized setting is mainly inspired by (Chen et al., 2022). We show that the gradient norm square of such an algorithm will converge with the optimal sample complexity of $\widetilde{\mathcal{O}}(\varepsilon^{-3})$. Moreover, the algorithm will converge to the *global optimum* with the sample complexity of $\widetilde{\mathcal{O}}(\varepsilon^{-6})$. In the rest of this section, we first explain the update of the algorithm, and then prove its convergence.

### E.1   Decentralized natural Actor-Critic

The natural policy gradient (NPG) algorithm (Kakade, 2002) can be viewed as a preconditioned policy gradient algorithm, which updates as follow:

$$
\theta_{k+1} = \theta_k - \alpha_k F(\theta_k)^\dagger \nabla J(\theta_k), \tag{102}
$$

where $F(\theta) := \mathbb{E}_{s \sim d_{\pi_\theta}, a \sim \pi_\theta} \left[ \psi_\theta(s,a) \psi_\theta(s,a)^T \right]$ is the Fisher information matrix (FIM). The natural Actor-Critic (NAC) uses the critic variable to estimate the gradient. The main challenge for implementing NAC lies in the estimation of the matrix-vector product $F(\theta_k)^\dagger \nabla J(\theta_k)$, especially under the decentralized setting. The work (Chen et al., 2022) proposes to solve the following subproblem in order to estimate the product in a decentralized way:

$$
h(\theta_k) = \arg\min_h f_{\theta_k}(h) := \frac{1}{2} h^T F(\theta_k) h - \nabla J(\theta_k)^T h. \tag{103}
$$

---

**Algorithm 3:** Decentralized single-timescale NAC

---

1: **Initialize:** Actor parameter $\theta_0$, critic parameter $\omega_0$, reward estimator parameter $\lambda_0$, initial state $s_0$, natural policy gradient estimation $h_{k,0}$.
2: **for** $k = 0, \cdots, K - 1$ **do**
3:    **Option 1: i.i.d. sampling:**
4:    $s_k \sim \mu_{\theta_k}(\cdot), a_k \sim \pi_{\theta_k}(\cdot|s_k), s_{k+1} \sim \mathcal{P}(\cdot|s_k, a_k)$.
5:    **Option 2: Markovian sampling:**
6:    $a_k \sim \pi_{\theta_k}(\cdot|s_k), s_{k+1} \sim \mathcal{P}(\cdot|s_k, a_k)$.
7:
8:    **Periodical consensus:** Compute $\tilde{\omega}_k^i$ and $\tilde{\lambda}_k^i$ by (4) and (7).
9:
10:    **for** $i = 0, \cdots, N$ **in parallel do**
11:      **Reward estimator update:** Update $\lambda_{k+1}^i$ by (8).
12:      **Critic update:** Update $\omega_{k+1}^i$ by (5).
13:      **Actor update:**
14:      Collect $N_a$ transition samples based on Markovian/i.i.d sampling.
15:      **for** $k' = 1, \cdots, K_a$ **do**
16:        Estimate $\bar{z}_{k',n}, \; \forall n \in [N_a]$ using (104).
17:        Update $h_{k,k'+1}$ by (106).
18:      **end for**
19:      Update $\theta_{k+1}^i$ by (107).
20:    **end for**
21: **end for**

---

Such a problem can be solved by using (stochastic) gradient descent, where the gradient is calculated by $F(\theta_k)h - \nabla J(\theta_k)$. For the centralized setting, the gradient w.r.t. each agent can be approximated as $\frac{1}{N_a} \sum_{n=1}^{N_a} \psi_{\theta_k}^i(s_n, a_n^i)\psi_{\theta_k}(s_n, a_n)^T h - g_a^i(\xi_n, \omega_{k+1}, \lambda_{k+1})$. However, when considering the decentralized setting, the term $\bar{z}_n := \psi_{\theta_k}(s_n, a_n)^T h = \sum_{i=1}^{N} \psi_{\theta_k}^i(s_n, a_n)^T h^i$ is not accessible for each agent. To approximate this value under the decentralized setting, agents compute $z_{n,0}^i := \psi_{\theta_k}^i(s_n, a_n)^T h^i$ locally and then perform the following communication step for $K_z$ steps:

$$z_{n,k'+1}^i = \sum_{j=1}^{N} W^{ij} z_{n,k'}^i, \; \forall n \in [N_a], \; k' = 0, \cdots, K_z - 1. \tag{104}$$

As we will see, the value $N z_{n,k'}^i$ converges to $\bar{z}_n$ linearly. Thus, the gradient of the subproblem (103) for agent $i$ can be approximated as:

$$\widetilde{\nabla} f_{\theta_k}^i(h_{k,k'}) := \frac{N}{N_a} \sum_{n=1}^{N_a} \psi_{\theta_k}^i(s_n, a_n^i) z_{n,K_z}^i - g_a^i(\xi_n, \omega_{k+1}, \lambda_{k+1}). \tag{105}$$

Then, each agent $i$ performs the following update for $K_a$ steps to estimate the natural policy gradient direction:

$$h_{k,k'+1}^i = \Pi_{C_h}(h_{k,k'}^i - \varrho \widetilde{\nabla} f_{\theta_k}^i(h_{k,k'})), \tag{106}$$

where $\varrho$ is a positive constant step size. Since the norm of optimal direction is bounded by $C_h := \lambda_{\max}(F(\theta)^{-1})C_\theta$, we project the vector into a ball of norm $C_h$ for each update. Finally, we perform the approximate natural policy gradient step as:

$$\theta_{k+1}^i = \theta_k^i - \alpha_k h_{k,K_a}^i. \tag{107}$$

## E.2 Convergence of natural Actor-Critic

In this section, we establish the sample complexity of Algorithm 3. We first introduce an additional assumption.

**Assumption 6.** *(invertible FIM) There exists a positive constant $\lambda_F$ such that for all policy $\theta$, $\lambda_{\min}(F(\theta)) \geq \lambda_F$.*

Assumption 6 ensures that $F(\theta)$ is positive definite so that the problem (103) is strongly convex for all policy. Such an assumption is also adopted by (Chen et al., 2022; Xu et al., 2021; Liu et al., 2020).

We now show the sample complexity of the Algroithm 3 in terms of gradient norm and the global optimality gap. To keep the analysis concise, we will consider the i.i.d. sampling scheme where we can directly sample transition tuples $(s, a, s')$ from the stationary distribution $\mu_{\pi_\theta}$. Extending the analysis to the Markovian sampling scheme essentially follows the similar technique as in AC's analysis, which introduces an additional $\mathcal{O}(\log(\varepsilon^{-1}))$ error terms caused by Markov chain mixing, and an error of order $\mathcal{O}(\frac{1}{1-\gamma})$ due to the mismatch between $\mu_{\pi_\theta}$ and $d_{\pi_\theta}$.

**Theorem 3.** *Suppose Assumptions 1-6 hold. Consider the update of Algorithm 3 under the i.i.d. sampling. Let $\alpha_k = \frac{\bar{\alpha}}{\sqrt{K}}$ for some positive constant $\bar{\alpha}$, $\beta_k = \frac{C_9}{2\lambda_\phi}\alpha_k$, $\varrho \leq \frac{1}{2C_\psi^2}$, $N_a = \mathcal{O}(\sqrt{K})$, $K_a = \mathcal{O}(\log(K^{1/2}))$, $K_c = \mathcal{O}(\log(K^{1/4}))$. Then, the following hold*

$$\frac{1}{K}\sum_{k=1}^{K}\sum_{i=1}^{N}\mathbb{E}\left[\|\nabla_{\theta^i}J(\theta_k)\|^2\right] \leq \mathcal{O}\left(\frac{1}{\sqrt{K}}\right) + \mathcal{O}(\varepsilon_{app} + \varepsilon_{sp}) \tag{108}$$

$$\frac{1}{K}\sum_{k=0}^{K}J(\theta^*) - J(\theta_k) \leq \mathcal{O}\left(\frac{1}{K^{1/4}}\right) + \mathcal{O}(\varepsilon_{app} + \varepsilon_{sp} + \varepsilon_{actor}). \tag{109}$$

The error $\varepsilon_{app}$ and $\varepsilon_{sp}$ are defined in (12) and (4.2), respectively. The error $\varepsilon_{actor}$ is referred as "compatible function approximation error", which is defined as:

$$\varepsilon_{actor} := \max_\theta \min_d \mathbb{E}_{s\sim d_{\pi_\theta}, a\sim \pi_\theta}[(\psi_\theta(s,a)^T d - A_{\pi_\theta}(s,a))^2].$$

Such an error captures the expressivity of the policy parameterization class: it measures the error of approximating $A_{\pi_\theta}(s,a)$ using $\psi_\theta(s,a)$ as feature. The error becomes 0 when using the softmax-tabular parameterization; see more discussions in Section 6 of (Agarwal et al., 2019).

Based on Theorem 3, Algorithm 3 needs $K = \mathcal{O}(\varepsilon^{-2})$ iterations to achieve $\varepsilon$-error for gradient norm square, and thus attains the sample complexity of $KN_aK_a = \widetilde{\mathcal{O}}(\varepsilon^{-3})$, which matches the best existing sample complexity of NAC (Xu et al., 2020; Chen et al., 2022). In terms of the global optimality gap, the algorithm requires $K = \mathcal{O}(\varepsilon^{-4})$ iterations to achieve $\varepsilon$-error, and thus has the sample complexity of $KN_aK_a = \widetilde{\mathcal{O}}(\varepsilon^{-6})$. Such a sample complexity is worse than the best existing sample complexity of $\widetilde{\mathcal{O}}(\varepsilon^{-3})$ (Xu et al., 2020; Chen et al., 2022).

We now explain the gap for the sub-optimal sample complexity. Mimicking the analysis of (Chen et al., 2022) allows to establish the following inequality:

$$\frac{1}{K}\sum_{k=0}^{K}J(\theta^*) - \mathbb{E}[J(\theta_k)] \leq \mathcal{O}\left(\frac{1}{K}\sum_{k=1}^{K}\sum_{i=1}^{N}\mathbb{E}[\|\nabla_{\theta^i}J(\theta_k)\|^2]\right)$$
$$+ \mathcal{O}\left(\frac{1}{K}\sum_{k=1}^{K}\sum_{i=1}^{N}\mathbb{E}\|\omega_k^i - \omega^*(\theta_k)\|\right) + \mathcal{O}\left(\frac{1}{K\alpha_k}\right). \tag{110}$$

While our analysis can obtain the iteration complexity of $\mathcal{O}(\frac{1}{\sqrt{K}})$ for $\|\nabla J(\theta_k)\|^2$, we can only achieve $\mathcal{O}(\frac{1}{K^{1/4}})$ iteration complexity for critic's error $\|\omega_k - \omega^*(\theta_k)\|$. This is because our algorithm uses single-timescale update, where the critic's error inevitably converges slower than that of double-loop based algorithms which have $\mathcal{O}(\frac{1}{\sqrt{K}})$ complexity for the critic's error at each iteration. Therefore, the sample complexity in terms of global optimality gap of our single-timescale NAC is dominated by this critic's error term, resulting in the final complexity of $\widetilde{\mathcal{O}}(\varepsilon^{-6})$. Nevertheless, the bound (110) is not necessarily tight. We leave the research on the tight bound of single-timescale NAC as a future work.

## E.3 Proof of Theorem 3

By Lemma 4, we have

$$
\mathbb{E}[J(\theta_{k+1})] - J(\theta_k) \geq \mathbb{E}\langle \nabla_\theta J(\theta_k), \theta_{k+1} - \theta_k \rangle - \frac{L}{2}\|\theta_{k+1} - \theta_k\|^2
$$

$$
\overset{(i)}{\geq} \alpha_k \mathbb{E}\langle \nabla_\theta J(\theta_k), h_k \rangle - \frac{L}{2} N C_h^2 \alpha_k^2
$$

$$
= \alpha_k \mathbb{E}\langle \nabla_\theta J(\theta_k), F(\theta_k)^{-1} g_a(\xi_k, \omega_{k+1}, \lambda_{k+1}) \rangle
$$

$$
+ \alpha_k \mathbb{E}\langle \nabla_\theta J(\theta_k), h_k - F(\theta_k)^{-1} g_a(\xi_k, \omega_{k+1}, \lambda_{k+1}) \rangle - \frac{L}{2} N C_h^2 \alpha_k^2
$$

$$
\overset{(ii)}{=} \alpha_k \mathbb{E}\langle F(\theta_k)^{-1/2} \nabla_\theta J(\theta_k), F(\theta_k)^{-1/2} g_a(\xi_k, \omega_{k+1}, \lambda_{k+1}) \rangle
$$

$$
+ \alpha_k \mathbb{E}\langle \nabla_\theta J(\theta_k), h_k - F(\theta_k)^{-1} g_a(\xi_k, \omega_{k+1}, \lambda_{k+1}) \rangle - \frac{L}{2} N C_h^2 \alpha_k^2
$$

$$
= \frac{\alpha_k}{2}\|F(\theta_k)^{-1/2} \nabla_\theta J(\theta_k)\|^2 + \frac{\alpha_k}{2}\|F(\theta_k)^{-1/2}\mathbb{E}[g_a(\xi_k, \omega_{k+1}, \lambda_{k+1})]\|^2
$$

$$
- \frac{\alpha_k}{2}\|F(\theta_k)^{-1/2} \nabla_\theta J(\theta_k) - F(\theta_k)^{-1/2}\mathbb{E}[g_a(\xi_k, \omega_{k+1}, \lambda_{k+1})]\|^2
$$

$$
+ \alpha_k \mathbb{E}\langle \nabla_\theta J(\theta_k), h_k - F(\theta_k)^{-1} g_a(\xi_k, \omega_{k+1}, \lambda_{k+1}) \rangle - \frac{L}{2} N C_h^2 \alpha_k^2
$$

$$
\overset{(iii)}{\geq} \frac{\alpha_k}{4} C_\psi^{-2}\|\nabla_\theta J(\theta_k)\|^2 + \frac{\alpha_k}{2}\lambda_F^{-1}\|\mathbb{E}[g_a(\xi_k, \omega_{k+1}, \lambda_{k+1})]\|^2
$$

$$
- \frac{\alpha_k}{2}\lambda_F^{-1}\underbrace{\|\nabla_\theta J(\theta_k) - \mathbb{E}[g_a(\xi_k, \omega_{k+1}, \lambda_{k+1})]\|^2}_{I_1}
$$

$$
- 4\alpha_k C_\psi^2 \underbrace{\|\mathbb{E}[h_k] - F(\theta_k)^{-1}\mathbb{E}[g_a(\xi_k, \omega_{k+1}, \lambda_{k+1})]\|^2}_{I_2} - \frac{L}{2} N C_h^2 \alpha_k^2, \tag{111}
$$

where $(i)$ is due to $\|\theta_{k+1}^i - \theta_k^i\| \leq C_h := \lambda_F C_\theta$. Note that we use $h_k^i$ to represent $h_{k,K_a}^i$ for simplifying the notation. $(ii)$ uses decomposition of positive definite (PD) matrix. Specifically, let $A$ be PD matrix, then by eigenvalue decomposition, $A = V\Lambda V^T$ for some orthonormal matrix $V$. Define $A^{-1/2} := V\Lambda^{-1/2}V^T$, then $\langle x, A^{-1}y \rangle = \langle A^{-1/2}x, A^{-1/2}y \rangle$ for any $x$ and $y$. $(iii)$ uses $C_\psi^{-2} \leq \lambda(F(\theta)^{-1}) \leq \lambda_F^{-1}$ and Young's inequality.

$I_1$ represents the error of gradient bias, which we have bounded when analyzing the error of AC. By (84), we have

$$
I_1 \leq \sum_{i=1}^{N} \left[16(\varepsilon_{sp} + C_\psi^2 \varepsilon_{app}) + 16C_\psi^2\|\omega^*(\theta_k) - \omega_{k+1}^i\|^2 + 8C_\psi^2\|\lambda^*(\theta_k) - \lambda_{k+1}^i\|^2\right]. \tag{112}
$$

To bound $I_2$, we need to bound the error of $h_{k,k'}$. We start with the gradient bias when estimating $h_{k,k'}$. Define $\overline{\nabla}f_{k,k'}(h_{k,k'}) := \nabla F(\theta_k)h_{k,k'} - \mathbb{E}[g_a(\xi_k, \omega_{k+1}^i, \lambda_{k+1}^i)]$, then it is easy to see that $\overline{\nabla}f_{k,k'}(h_{k,k'})$ is the unbiased gradient of the following problem

$$
\frac{1}{2}h_{k,k'}^T \nabla F(\theta_k)h_{k,k'} - \mathbb{E}[g_a(\xi_k, \omega_{k+1}^i, \lambda_{k+1}^i)]^T h_{k,k'}.
$$

Define the following notation for the ease of expression:

$$
\widehat{\nabla}f_{k,k'}^i(h_{k,k'}) := \frac{1}{N_a}\sum_{n=1}^{N_a} \psi_{\theta_k^i}(s_n, a_n^i)\psi_{\theta_k}(s_n, a_n)^T h_{k,k'} - g_a^i(\xi_{k,k'}, \omega_{k+1}^i, \lambda_{k+1}^i)
$$

$$
\widehat{\nabla}f_{k,k'}(h_{k,k'}) := [\widehat{\nabla}f_{k,k'}^1(h_{k,k'}), \cdots, \widehat{\nabla}f_{k,k'}^N(h_{k,k'})]
$$

$$
\widetilde{\nabla}f_{k,k'}^i(h_{k,k'}) := \frac{N}{N_a}\sum_{n=1}^{N_a} \psi_{\theta_k^i}(s_n, a_n^i)z_{n,K_z}^i - g_a^i(\xi_{k,k'}, \omega_{k+1}^i, \lambda_{k+1}^i)
$$

$$
\widetilde{\nabla}f_{k,k'}(h_{k,k'}) := [\widetilde{\nabla}f_{k,k'}^1(h_{k,k'}), \cdots, \widetilde{\nabla}f_{k,k'}^N(h_{k,k'})].
$$

We now analyze the error at outer-loop iteration $k$. For notational simplicity, we omit the subscript $k$ for the prementioned notations, e.g. we use $\widehat{\nabla} f_{k'}^i(h_{k'})$, $\widehat{\nabla} f_{k'}(h_{k'})$, $\widetilde{\nabla} f_{k'}^i(h_{k'})$, $\widetilde{\nabla} f_{k'}(h_{k'})$ to represent the above notations, respectively.

$$\|\overline{\nabla} f_{k'}(h_{k'}) - \widetilde{\nabla} f_{k'}(h_{k'})\|^2 \leq 2 \underbrace{\|\overline{\nabla} f_{k'}(h_{k'}) - \widehat{\nabla} f_{k'}(h_{k'})\|^2}_{I_3} + 2 \underbrace{\|\widehat{\nabla} f_{k'}(h_{k'}) - \widetilde{\nabla} f_{k'}(h_{k'})\|^2}_{I_4}.$$

$I_3$ can be bounded as

$$
\begin{aligned}
I_3 &= \|\sum_{n=1}^{N_a} (\frac{1}{N_a} \psi_\theta(s_n, a_n) \psi_\theta(s_n, a_n)^T - F(\theta)) h_{k'}\|^2 \\
&\leq \|\sum_{n=1}^{N_a} (\frac{1}{N_a} \psi_\theta(s_n, a_n) \psi_\theta(s_n, a_n)^T - F(\theta))\|^2 C_h^2 \\
&\leq \frac{1}{N_a} C_\psi^4 C_h^2.
\end{aligned}
\tag{113}
$$

$I_4$ can be bounded as

$$
\begin{aligned}
I_4 &= \sum_{i=1}^{N} \left\| \psi_{\theta^i}(s_n, a_n^i) \left( \frac{1}{N_a} \sum_{n=1}^{N_a} N z_{n,K_z}^i - \psi_\theta(s_n, a_n)^T h_{k'} \right) \right\|^2 \\
&\leq \frac{1}{N_a} N C_\psi^2 \sum_{i=1}^{N} \sum_{n=1}^{N_a} \|z_{n,K_z}^i - \bar{z}_{n,K_z}\|^2 \\
&= \frac{N C_\psi^2}{N_a} \sum_{n=1}^{N_a} \|Q W^{K_z} z_{n,0}\|^2 \\
&\leq \frac{N C_\psi^2}{N_a} \sum_{n=1}^{N_a} \nu^{K_z} \|z_{n,0}\|^2 \leq N C_\psi^4 C_h^2 \nu^{K_z}.
\end{aligned}
\tag{114}
$$

Let $K_z = \min\{c \in \mathbb{N}^+ | \nu^c \leq \frac{4}{N_a N}\}$, then $K_z = \mathcal{O}(\log \frac{1}{N_a})$. Combine (113) and (114) gives us

$$\|\overline{\nabla} f_{k'}(h_{k'}) - \widetilde{\nabla} f_{k'}(h_{k'})\|^2 \leq \frac{4 C_\psi^4 C_h^2}{N_a}.$$

We now analyze the error of $h_{k,k'}$. Note that we omit the subscript $k$ here for simplifying notation. Define

$$h^* = \arg\min_h \bar{f}_\theta(h) := h^T F(\theta) h := -\mathbb{E}_{\xi \sim \mu_\theta}[g_a(\xi, \omega, \lambda)]^T h. \tag{115}$$

It is easy to see that the function on the RHS is strongly convex, since $F(\theta)$ is positive definite w.r.t. $h$. We bound the optimal gap by

$$
\begin{aligned}
\mathbb{E}\|h_{k'+1} - h^*\|^2 &= \mathbb{E}\|h_{k'} - \varrho \widetilde{\nabla} f_{k'}(h_{k'}) - h^*\|^2 \\
&= \mathbb{E}\|h_{k'} - h^*\|^2 - 2\varrho \mathbb{E}\langle h_{k'} - h^*, \widetilde{\nabla} f_{k'}(h_{k'})\rangle + \varrho^2 \|\widetilde{\nabla} f_{k'}(h_{k'})\|^2 \\
&\leq \mathbb{E}\|h_{k'} - h^*\|^2 - 2\varrho \mathbb{E}\langle h_{k'} - h^*, \overline{\nabla} f_{k'}(h_{k'})\rangle + 2\varrho \mathbb{E}\langle h_{k'} - h^*, \overline{\nabla} f_{k'}(h_{k'}) - \widetilde{\nabla} f_{k'}(h_{k'})\rangle \\
&\quad + 2\varrho^2 \|\overline{\nabla} f_{k'}(h_{k'})\|^2 + 2\varrho^2 \|\widetilde{\nabla} f_{k'}(h_{k'}) - \overline{\nabla} f_{k'}(h_{k'})\|^2 \\
&\overset{(i)}{\leq} (1 - \varrho\lambda_F)\mathbb{E}\|h_{k'} - h^*\|^2 - 2\varrho(f_{k'}(h_{k'}) - \overline{f}^*) + 2\varrho\mathbb{E}\langle h_{k'} - h^*, \overline{\nabla} f_{k'}(h_{k'}) - \widetilde{\nabla} f_{k'}(h_{k'})\rangle \\
&\quad + 2\varrho^2 \|\overline{\nabla} f_{k'}(h_{k'})\|^2 + 2\varrho^2 \|\widetilde{\nabla} f_{k'}(h_{k'}) - \overline{\nabla} f_{k'}(h_{k'})\|^2
\end{aligned}
$$

$$\overset{(ii)}{\le} (1 - \varrho\lambda_F)\mathbb{E}\|h_{k'} - h^*\|^2 - 2\varrho(1 - 2\varrho C_\psi^2)(f_{k'}(h_{k'}) - \overline{f}^*)$$
$$+ 2\varrho\mathbb{E}\langle h_{k'} - h^*, \overline{\nabla}f_{k'}(h_{k'}) - \widetilde{\nabla}f_{k'}(h_{k'})\rangle + 2\varrho^2\|\widetilde{\nabla}f_{k'}(h_{k'}) - \overline{\nabla}f_{k'}(h_{k'})\|^2$$

$$\overset{(iii)}{\le} (1 - \varrho\lambda_F)\mathbb{E}\|h_{k'} - h^*\|^2 + 2\varrho\mathbb{E}\langle h_{k'} - h^*, \overline{\nabla}f_{k'}(h_{k'}) - \widetilde{\nabla}f_{k'}(h_{k'})\rangle$$
$$+ 2\varrho^2\|\widetilde{\nabla}f_{k'}(h_{k'}) - \overline{\nabla}f_{k'}(h_{k'})\|^2$$

$$\overset{(iiii)}{\le} (1 - \frac{\varrho\lambda_F}{2})\mathbb{E}\|h_{k'} - h^*\|^2 + (\frac{2\varrho}{\lambda_F} + 2\varrho^2)\|\widetilde{\nabla}f_{k'}(h_{k'}) - \overline{\nabla}f_{k'}(h_{k'})\|^2,$$

where $\overline{f}^*$ is the optimal value of $\overline{f}(h)$ defined in (115), and the inequality follows the property of $\lambda_F$-strongly convex function: $\overline{f}(h_2) \ge \overline{f}(h_1) + \langle\nabla\overline{f}(h_1), h_2 - h_2\rangle + \frac{\lambda_F}{2}\|h_1 - h_2\|^2$, $\forall h_1, h_2$. $(ii)$ uses the PL condition implied by $\lambda_F$-strong convexity: $\overline{f}(h^*) - \overline{f}(h) \le -\frac{1}{2\lambda_F}\|\nabla\overline{f}(h)\|^2$, $\forall h$. $(iii)$ is due to step size rule that $\varrho \le \frac{1}{2C_\psi^2}$. $(iiii)$ applies Young's inequality.

Use the above induction, we have

$$\mathbb{E}\|h_{K_a} - h^*\|^2 \le (1 - \frac{\varrho\lambda_F}{2})^{K_a}\|h_0 - h^*\|^2 + \sum_{t=0}^{K_a}(1 - \frac{\varrho\lambda_F}{2})^t(\frac{2\varrho}{\lambda_F} + 2\varrho^2)\|\overline{\nabla}f_{K_a-t}(h_{K_a}) - \widetilde{\nabla}f_{K_a}(h_{K_a})\|^2$$

$$\le 4C_h^2(1 - \frac{\varrho\lambda_F}{2})^{K_a} + (\frac{4\varrho}{\varrho\lambda_F^2} + \frac{4\varrho}{\lambda_F})C_\psi^4 C_h^2 \frac{4}{N_a}.$$

Let $K_a = \min\{c \in \mathbb{N}^+ | 4C_h^2(1 - \frac{\varrho\lambda_F}{2})^c = (\frac{4\varrho}{\varrho\lambda_F^2} + \frac{4\varrho}{\lambda_F})C_\psi^4 C_h^2 \frac{1}{N_a}\}$, then $K_a = \mathcal{O}(\log(\frac{1}{N_a}))$. Define $C_{18} := (\frac{16\varrho}{\varrho\lambda_F^2} + \frac{16\varrho}{\lambda_F})C_\psi^4 C_h^2$, we have

$$I_2 = \mathbb{E}\|h_{K_a} - h^*\|^2 \le \frac{2C_{18}}{N_a}. \tag{116}$$

Plug (112) and (116) back to (111), we have

$$\mathbb{E}[J(\theta_{k+1})] - J(\theta_k) \ge \sum_{i=1}^{N}[\frac{\alpha_k}{4}C_\psi^{-2}\|\nabla_{\theta^i}J(\theta_k)\|^2 + \frac{\alpha_k}{2}\lambda_F\|F(\theta_k)^{-1}\mathbb{E}[g_a^i(\xi_k, \omega_{k+1}^i, \lambda_{k+1}^i)]\|^2 + \alpha_k C_\psi^2 \frac{2C_{18}}{N_a}$$
$$+ 8\lambda_F^{-1}(\varepsilon_{sp} + C_\psi^2\varepsilon_{app}) + 8\lambda_F^{-1}C_\psi^2\|\omega^*(\theta_k) - \omega_{k+1}^i\|^2 + 4\lambda_F^{-1}C_\psi^2\|\lambda^*(\theta_k) - \lambda_{k+1}^i\|^2]$$

Consider the Lyapunov function

$$\mathbb{V}^k = -J(\theta_k) + \lambda_F^{-1}(\|\omega_k - \omega^*(\theta_k)\|^2 + \|\lambda_k - \lambda^*(\theta_k)\|^2).$$

The difference of the Lyapunov function is

$$\mathbb{E}[\mathbb{V}^{k+1}] - \mathbb{E}[\mathbb{V}^k] = \mathbb{E}[J(\theta_k)] - \mathbb{E}[J(\theta_{k+1})] + \lambda_F^{-1}(\mathbb{E}\|\omega_{k+1} - \omega^*(\theta_{k+1})\|^2 - \mathbb{E}\|\omega_k - \omega^*(\theta_k)\|^2$$
$$+ \mathbb{E}\|\lambda_{k+1} - \lambda^*(\theta_{k+1})\|^2 - \mathbb{E}\|\lambda_k - \lambda^*(\theta_k)\|^2)$$

$$\le \sum_{i=1}^{N}\left[\frac{\alpha_k}{4}C_\psi^{-2}\mathbb{E}\|\nabla_{\theta^i}J(\theta_k)\|^2 + \frac{\alpha_k}{2}\lambda_F\|F(\theta_k)^{-1}\mathbb{E}[g_a^i(\xi_k, \omega_{k+1}^i, \lambda_{k+1}^i)]\|^2 + \alpha_k C_\psi^2 \frac{2C_{18}}{N_a}\right]$$

$$+ \lambda_F^{-1}\underbrace{\left[\sum_{i=1}^{N}8C_\psi^2\alpha_k\mathbb{E}\|\omega^*(\theta_k) - \omega_{k+1}^i\|^2 + \mathbb{E}\|\bar{\omega}_{k+1} - \omega^*(\theta_{k+1})\|^2 - \mathbb{E}\|\bar{\omega}_k - \omega^*(\theta_k)\|^2\right]}_{I_5}$$

$$+ \lambda_F^{-1}\underbrace{\left[\sum_{i=1}^{N}4C_\psi^2\alpha_k\mathbb{E}\|\lambda^*(\theta_k) - \lambda_{k+1}^i\|^2 + \mathbb{E}\|\bar{\lambda}_{k+1} - \lambda^*(\theta_{k+1})\|^2 - \mathbb{E}\|\bar{\lambda}_k - \lambda^*(\theta_k)\|^2\right]}_{I_6}$$

$$+ 8N\lambda_F^{-1}(\varepsilon_{sp} + C_\psi^2\varepsilon_{app}). \tag{117}$$

By following the similar procedures through (87) to (91), we can bound $I_5$ and $I_6$ as

$$I_5 \leq (1 + C_{19}\alpha_k)C_\delta^2\beta_k^2 + \frac{\alpha_k}{4}\lambda_F^{-1}\sum_{i=1}^{N}\mathbb{E}\|F(\theta_k)^{-1}g_a^i(\xi_k, \omega_{k+1}^i, \lambda_{k+1}^i)\|^2 + \alpha_k M_{k_1} + C_{20}\alpha_k^2 \qquad (118)$$

$$I_6 \leq (1 + C_{21}\alpha_k)C_\lambda^2\eta_k^2 + \frac{\alpha_k}{4}\lambda_F^{-1}\sum_{i=1}^{N}\mathbb{E}\|F(\theta_k)^{-1}g_a^i(\xi_k, \omega_{k+1}^i, \lambda_{k+1}^i)\|^2 + \alpha_k M_{k_2} + C_{22}\alpha_k^2, \qquad (119)$$

where $C_{19}, C_{20}, C_{21}, C_{22}$ are some positive constants. Plug (118) and (119) back to (117), we have

$$\mathbb{E}[\mathbb{V}^{k+1}] - \mathbb{E}[\mathbb{V}^k] \leq \sum_{i=1}^{N}[\frac{\alpha_k}{4}C_\psi^{-2}\mathbb{E}\|\nabla_{\theta^i}J(\theta_k)\|^2 + \alpha_k C_\psi^2\frac{2C_{18}}{N_a} + \mathcal{O}(\alpha_k^2 + \beta_k^2 + \eta_k^2)$$
$$+ (M_{k_1} + M_{k_2})\alpha_k + \mathcal{O}(\varepsilon_{sp} + \varepsilon_{app})\alpha_k]. \qquad (120)$$

By telescoping (120), we can get

$$\frac{1}{K}\sum_{k=0}^{K}\sum_{i=1}^{N}\mathbb{E}\|\nabla_{\theta^i}J(\theta_k)\|^2 \leq \frac{4C_\psi^2\mathbb{V}_0}{K\alpha_k} + \mathcal{O}(\varepsilon_{sp} + \varepsilon_{app}) + \frac{8C_\psi^2 C_{18}}{N_a} + \mathcal{O}(\alpha_k + \frac{\beta_k^2}{\alpha_k} + \frac{\eta_k^2}{\alpha_k})$$
$$+ 4C_\psi^2\frac{1}{K}\sum_{k=0}^{K}(M_{k_1} + M_{k_2})$$

By (95), $M_{k_1} + M_{k_2} = \mathcal{O}(\frac{1}{\sqrt{K}})$ when $K_c \leq \mathcal{O}(K^{1/4})$. Therefore, let $C, \bar{\alpha}$ be some positive constants. Set $N_a = C\sqrt{K}$, $\alpha_k = \frac{\bar{\alpha}}{\sqrt{K}}$, $\beta_k = \frac{C_9}{2\lambda_\phi}\alpha_k$, $\eta_k = \frac{C_{10}}{2\lambda_\varphi}\alpha_k$, we obtain the desired result of (108).

We now prove (109). Let $\mathbb{E}_{\theta^*}$ denote the expectation over $s \sim d_{\pi_{\theta^*}}, a \sim \pi_{\theta^*}(\cdot|s)$. By the smoothness of $\log\pi_\theta(a|s)$, we have

$$\mathbb{E}_{\theta^*}[\log\pi_{\theta_{k+1}}(a|s) - \log\pi_{\theta_k}(a|s)]$$
$$\geq \alpha_k\mathbb{E}_{\theta^*}[\psi_{\theta_k}(s,a)^T h_k] - \frac{L_\psi\alpha_k^2}{2}C_h^2$$
$$\geq \alpha_k\mathbb{E}_{\theta^*}[\psi_{\theta_k}(s,a)^T(h_k - h^*(\theta_k))] + \alpha_k\mathbb{E}_{\theta^*}[\psi_{\theta_k}(s,a)^T h^*(\theta_k) - A_{\theta_k}(s,a)]$$
$$+ \alpha_k\mathbb{E}_{\theta^*}[A_{\theta_k}(s,a)] - \frac{L_\psi\alpha_k^2}{2}C_h^2$$
$$\geq -\alpha_k C_\psi\|h_k - h^*(\theta_k)\| - \alpha_k\sqrt{\varepsilon_{actor}} + \alpha_k(J(\theta^*) - J(\theta_k)) - \frac{L_\psi\alpha_k^2}{2}C_h^2.$$

By telescoping the above inequality and rearranging terms, we have

$$\frac{1}{K}\sum_{k=1}^{K}(J(\theta^*) - J(\theta_k)) \leq \frac{1}{K\alpha_k}\mathbb{E}_{\theta^*}[\log\pi_K(a|s) - \log\pi_0(a|s)] + \sqrt{\varepsilon_{actor}}$$
$$+ \frac{1}{K}\sum_{k=1}^{K}C_\psi\|h_k - h^*(\theta_k)\| + \frac{1}{K}\sum_{k=1}^{K}\frac{L_\psi\alpha_k}{2}.$$

The term $\|h_k - h^*(\theta_k)\| \leq \|h_k - F(\theta_k)^{-1}\mathbb{E}[g_a(\xi_k, \omega_{k+1}, \lambda_{k+1})]\| + \|\mathbb{E}[g_a(\xi_k, \omega_{k+1}, \lambda_{k+1})] - F^{-1}\nabla J(\theta_k)\|$. Since by the (116) and (84), these two terms are of order $\mathcal{O}(\frac{1}{N_a^{1/2}})$ and $\mathcal{O}(\|\omega_k - \omega_{k+1}\| + \varepsilon_{app})$, respectively, we conclude that $\|h_k - h^*(\theta_k)\|$ is of order $\mathcal{O}(\|\omega_k - \omega^*(\theta_k)\| + \varepsilon_{app})$. By following the step size rule as suggested by Theorem 3, we obtain the desired result.

