# OpenReview forum: "Finite-Time Analysis of Decentralized Single-Timescale Actor-Critic"
_TMLR — Accepted by TMLR_

### Review · Reviewer_VYyQ · 2022-11-06

**Summary Of Contributions:**

This paper studies the Finite-Time convergence of Decentralized Single-Timescale Actor-Critic and Natural Actor-Critic algorithm. In this setting, we have A agents where each of them take actions a^1, ..., a^N separately according to their local policies parameterized by theta^1, ..., \theta^N, and each agent receive separate rewards. The goal is to find the optimal joint set of parameters \theta^1, ..., \theta^N which results in the maximum cumulative discounted reward.
The authors provide two algorithms for finding the optimal policy. In the first algorithm, we estimate the reward function of the agents separately, and we use this reward function approximation to find the optimal policy. However, this algorithm requires the knowledge of individual actions of the each agent, which might violate the privacy. For that, the authors further propose algorithm 2 which adds some noise to the reward function before sharing it with the other agents. However, this algorithm requires the agents to communicate log(1/\epsilon) times more.
Finally, the authors characterize the sample complexity of Algorithms 1 and 2. At the end the authors provide experimental results for their algorithm.

**Audience:**

Yes

**Broader Impact Concerns:**

No broader impact concerns

**Claims And Evidence:**

Yes

**Requested Changes:**

- In the bounds of Theorems 1 and 2 there is term O(\epsilon_{sp}), which is essentially a constant. This constant is not acceptable when we characterize an upper bound. Even though the authors argue that this term goes to zero as \gamma-->1, but still it is a constant which is not even dependent on the function approximation error. Can the authors get rid of this term?
- Furthermore, on the paper "Actor-Critic Algorithms " by Konda and Tsitsiklis, the authors characterize convergence to stationary point of actor-critic algorithms. In this paper, however, we do not even have convergence to stationary point, and the upper bound involves \epsilon_{app} term. Can the authors get red of this term, or argue why it is essential to have this term?
- Could the authors expand more on Assumption 5? What is the requirement on the upper bound on the second largest singular value mean? Does the assumption more or less require a complete synchronization among the agents every K_c time step? In that case, we cannot assume this algorithm is decentralized, since we need a complete synchronizer, which is against decentralization.
- Could you explain how the error of Algorithm 2 depend on the \sigma of the noise added?
- How do we choose the radius of projection? In the paper you are not mentioning a rule to choose this radius.

---------------------------------------------------------------------------------------------------------------------------------------------------------
Comments after reading the review:
- The reference to the paper [Konda et al., 1999] is not accurate. As long as I understand, [Konda et al., 1999] does not require the function approximation error to be zero. So my question regarding function approximation error for convergence to the stationary point remains.
- The argument provided by the authors for why $\varepsilon_{sp}$ is inevitable, is not satisfactory. There might be ways to get rid of this term. Either by changing the algorithm, or by a finer analysis.
- In the case you need to know the horizon before hand to tune the parameters, then we do require careful design of the parameters, and the point mentioned in the paper for the advantage of single time scale algorithm is no longer valid.

---------------------------------------------------------------------------------------------------------------------------------------------------------
The second round of responses, and the changes the authors made to the paper addressed my main concerns. I do not have crucial concerns anymore.

**Strengths And Weaknesses:**

Strength:
- The authors characterize a tight convergence bound
Weakness:
- The paper lacks novelty.
- The results are for convergence to stationary point, and even for that, the upper bound involves constants which do not go to zero.
- The convergence bound provided in the paper is based on average of gradients of the algorithm over time. The authors claim that single time-scale algorithm which is studies with them do not require careful design of the parameters. However, for one, the choice of the step size \alpha_k depends on the horizon, and also the bound is on the average of the gradient of the points observed so far, which means we need a uniform sampling of time step from 1 to K. Hence, their criticism for two-time-scale algorithms is not valid.

---

> ### Author Response · Authors · 2022-11-10
> **Author's response: Part I**
>
> Dear reviewer VYyQ,
>
> Thank you for your comments. We now address your concerns in a point-by-point manner below. Major changes in the manuscript are highlighted in blue.
>
> **A. "The paper lacks novelty".** We would like to highlight that our main contribution is the analysis on Actor-Critic algorithm with *single-timescale* update for general objective function, which --- to the best of our knowledge --- remains nearly unexplored in the literature. Note that existing results are derived based on either the double-loop update or two-timescale step sizes rule, which deviates a little bit from the practical implementation where actor and critic variables are updated in an *alternating manner* with step sizes being of *the same* order, namely, the single-timescale update. **Our work’s main novelty can be summarized as:** We show that *single-timescale* Actor-Critic algorithm has finite-time complexity bound guarantees. Moreover, it attains the optimal sample complexity that matches the previous double-loop based analysis. The key for establishing our convergence result is the *hidden smoothness of the optimal critic variable* we revealed, which have not been justified before and is non-trivial. Please refer to our manuscript’s introduction section for a more comprehensive summary of our work’s novelty and contribution.
>
> **B. “The results are for convergence to stationary point”.** The objective function we consider is a non-convex function (see an example in Appendix A of [Agarwal et al., 2019]). Although there are works which prove the gradient domination property (which means any stationary point will be global optimum; see Lemma 3 of [Fazel et al., 2018]) for particular problem setting such as linear quadratic regulator with linear policy, such a property cannot be established in our setting without additional assumptions. Hence, convergence to a stationary point is perhaps the only thing we can expect for Actor-Critic algorithm. Nevertheless, we have shown that the natural Actor-Critic can converge to a global optimum (with standard compatible function approximation error) by utilizing the structure of value function; see discussions in section 4.5 and Appendix E.
>
> **C. The $\varepsilon_{app}$ term in the convergence result.** Since we use critic to estimate the value function, the error between the true value function and the *best* critic’s approximated value function is inevitable, and thereby introduces the approximation error term $\varepsilon_{app}$, namely, $\varepsilon_{app}$ is not caused by our analysis. To the best of our knowledge, this term appears in almost all convergence results of Actor-Critic algorithms under *function approximation* setting; see e.g. [Xu et al., 2020], [Qiu et al., 2019]. The term $\varepsilon_{app}$ automatically becomes zero when the critic’s function class contains the true value function. The work [Konda et al., 1999] mentioned by the reviewer essentially assumes this condition so that $\varepsilon_{app} = 0$ in their case:
> > The features $\phi_{\theta}^{j}, j = 1, ... ,m$, used by the critic are dependent on the actor parameter vector $\theta$ and are chosen such that their span in $\mathbb{R}^{|\mathcal{S}||\mathcal{A}|}$, denoted by $\Phi_\theta$, contains $\Psi_\theta$.
>
> **D. The $\varepsilon_{sp}$ term in the convergence result.** The term $\varepsilon_{sp}$ captures the difference between the stationary distribution $\mu_{\pi_\theta}$ and the discounted state visitation distribution $d_{\pi_\theta}$. By the policy gradient expression (3) (here (3) refers to our manuscript), the states should be sampled from $d_{\pi_\theta}$ in order to obtain unbiased gradient estimation. Nevertheless, we do not assume the access to sample states from $d_{\pi_\theta}$ since it is often unknown in practice. By contrast, we analyze the practical implementation where the states are collected from an online trajectory. Since the state distribution converges to $\mu_{\pi_\theta}$ due to Markov chain’s mixing, the error between $\mu_{\pi_\theta}$ and $d_{\pi_\theta}$ is inevitable when analyzing the bias of the policy gradient, which again is not caused by our analysis. The term $\varepsilon_{sp}$ also appears in [Zeng et al., 2021], [Shen et al., 2020]. Some works assume that sampling from $ d_{\pi_\theta}$ is permitted, thus they have $\varepsilon_{sp}=0$; see e.g. [Xu et al., 2020], [Chen et al., 2021]. Please refer to Section 4.2 of the latest manuscript for more information.

---

> ### Author Response · Authors · 2022-11-10
> **Author's response: Part II**
>
> **E. "The bound is on the average of the gradient of the points observed so far, which means we need a uniform sampling of time step from 1 to K”.** Since our objective function is non-convex, we can only obtain bounds for the averaged gradients. Such a result is actually quite standard in non-convex optimization. We are not aware of any last iterate convergence result for general smooth non-convex optimization (without additional error bound-type conditions) even in the optimization society. If there is, we highly appreciate it if the reviewer could refer to us.
>
> **F. “The choice of $\alpha_k$ depends on the horizon”.** The horizon $T$ is not a tuned parameter. It is a free user-determined parameter, i.e., it is free to determine how many iterations one would like to run our algorithm.
>
> **G. Explanation on Assumption 5.** The second largest singular value of the communication matrix is required to lie in the range $\nu \in [0, 1)$. This is a standard and widely adopted assumption in decentralized reinforcement learning and decentralized optimization; see e.g. [Sun et al., 2020], [Singh et al., 2020]. The agents do not need a complete synchronization through the whole network. Instead, they only need to synchronize with its neighbors for every $K_c$ step. That is, for agent $i$, it only needs to exchange information with the agents $j$ where $W^{ij} \neq 0$. In practice, the communication matrix $W$ is usually sparse, which means that agents only need to communicate with very few neighbors, and the cost is much less than the complete synchronization. Please refer to equation (4) of the latest manuscript for more information.
>
> **H. How the $\sigma^2$ influences the Algorithm 2’s error?** $\sigma^2$ is the variance of the noise added to the local reward of each agent. When $\sigma^2$ increases, Algorithm 2 will have better privacy preservation of local reward. In the meantime, the estimation on the true average reward $\bar{r}(s_k,a_k)$ will have higher variance. That is, $\mathbb{E}[|\tilde{r}_{k, K_r}^{i} - \bar{r}(s_k, a_k)|^2]$ will increase. Hence, the algorithm will converge slower. In fact, the iteration complexity in Theorem 2 is of order $\mathcal{O}(\frac{\sigma^2}{\sqrt{T}})$, where we omit the $\sigma^2$ term for brevity. Please see the Section 3.2 of the latest manuscript for more information.
>
> **I. How to choose projection radius?** The projection radiuses $R_\omega$ and $R_\lambda$ are formally defined in Appendix C: $R_\omega:=\frac{r_\max}{\lambda_\phi}$, $R_\lambda:= \frac{r_\max}{\lambda_\varphi}$. In fact, our convergence results will hold for any $R_\omega \geq \frac{r_\max}{\lambda_\phi}$ and $ R_\lambda \geq \frac{r_\max}{\lambda_\varphi}$. In practice, one can estimate $R_\omega$ and $R_\lambda$ online; see Section 8.2 of [Bhandari et al., 2018] for one approach. Please see the Appendix C of the latest manuscript for detailed explanation.
>
> We hope that these revisions and responses are satisfactory to you. Once again, we truly appreciate the very detailed and constructive feedback, which has helped us greatly in improving the quality of the manuscript.
>
> **References**
>
> [Fazel et al., 2018] Global convergence of policy gradient methods for the linear quadratic regulator.
>
> [Agarwal et al., 2019] On the theory of policy gradient methods: optimality, approximation, and distribution shift.
>
> [Xu et al., 2020] Improving sample complexity bounds for (natural) actor-critic algorithms.
>
> [Qiu et al., 2019] On the finite-time convergence of actor-critic algorithm.
>
> [Konda et al., 1999] Actor-critic algorithms.
>
> [Zeng et al., 2021] Learning to coordinate in multi-agent systems: A coordinated actor-critic algorithm and finite-time guarantees.
>
> [Shen et al., 2020] Asynchronous advantage actor critic: Non-asymptotic analysis and linear speedup.
>
> [Chen et al., 2021] Tighter analysis of alternating stochastic gradient methods for bilevel problems.
>
> [Sun et al., 2020] Finite-sample analysis of decentralized temporal-difference learning with linear function approximation.
>
> [Singh et al., 2020] Squarm-sgd: Communication-efficient momentum sgd for decentralized optimization.
>
> [Bhandari et al., 2018] A finite-time analysis of temporal difference learning with linear function approximation.

---

> ### Author Response · Authors · 2022-12-13
> **Response to follow-up comments: Part I**
>
> Dear reviewer VYyQ,
>
> Thank you for your comments and your careful reading of our response. We now address your follow-up concerns.
>
> **A. “[Konda et al., 1999] does not require the function approximation error to be zero. So my question regarding function approximation error for convergence to the stationary point remains”.** Let us provide more details about their paper. In [Konda et al., 1999]’s paper, they consider using a linearly parameterized critic to estimate the Q-value:
> >$$Q_r^\theta(x,u)=\sum_{j=1}^{m}r^j\phi_{\theta}^{j}(x,u).$$
>
> Please see their paper for the definition of the notations. They assume that the feature vectors are selected such that their span contains the space $\Psi_{\theta}$:
> > The features $\phi_{\theta}^{j}, j = 1, ... ,m$, used by the critic are dependent on the actor parameter vector $\theta$ and are chosen such that their span in $\mathbb{R}^{|\mathcal{S}||\mathcal{A}|}$, denoted by $\Phi_\theta$, contains $\Psi_\theta$.
>
> Therefore, for any (projected) Q-value $\prod_{\theta}q_{\theta} \in \Psi_\theta$, there exists a critic $r \in \mathbb{R}^{m}$ such that the (projected) Q-value can be perfectly estimated: $\prod_{\theta}q_{\theta}=\Phi_{\theta} r$, namely, the function approximation error is zero. In this sense, for any policy $\theta$, there exists a critic variable such that the policy gradient can be calculated accurately based equation (2) of their paper and the policy gradient theorem:
> >$$\frac{\partial}{\partial \theta_i}\lambda(\theta) = \langle q_\theta, \psi_{\theta}^{i} \rangle_\theta.$$
>
> Hence, there is no term related to function approximation error in their convergence result.
>
> In our paper, we do not assume such a condition that feature vectors are related to $\theta$ and span the space $\Psi_{\theta}$. Consequently, the function approximation error cannot be zero and a related error term will appear in the convergence result, which also widely exists in the literature [Xu et al., 2020], [Qiu et al., 2019], [Chen et al., 2022], [Wu et al., 2020], [Shen et., 2020], [Hairi et al., 2021]. In other words, we consider the setting where even the best critic may not approximate the value function perfectly.
> We remark that we can adopt the same assumption as [Konda et al., 1999] and achieve zero approximation error.
>
> **B. “The argument provided by the authors for why $\varepsilon_{sp}$ is inevitable, is not satisfactory. There might be ways to get rid of this term. Either by changing the algorithm, or by a finer analysis”.** We first argue that such a term cannot be avoided by developing different analysis techniques. This is because policy gradient approaches use stochastic gradient ascent to update the policy, which requires that the stochastic gradient is unbiased in order to converge based on standard analysis. However, as shown in our analysis, the stochastic gradient is biased when updating actor. This is because the unbiased policy gradient should be calculated using states sampled from the state visitation distribution $d_{\pi_\theta}$, while the real implementations calculates the gradient using states that are sampled from a different distribution: a distribution that will converge to $\mu_{\pi_\theta}$. This will induce bias when evaluating the policy gradient. In other words, the expectation of the stochastic gradient
> will not be the true gradient of our objective function. Consequently, the algorithm will converge to a stationary point of a biased objective. Again, we emphasize that this error cannot be avoided by using different analysis technique as it is the intrinsic error of the problem. Our analysis characterizes the bias in terms of the discounted factor $\gamma$.
>
> To the best of our knowledge, we are not aware of works which can eliminate such a distribution mismatch error under our considered discounted reward setting and the alternating update scheme. Since the main focus of our work is to provide a finite-time convergence analysis for the practical single-timescale AC, designing an algorithm for avoiding the sampling bias goes beyond the scope of our work.

---

> ### Author Response · Authors · 2022-12-13
> **Response to follow-up comments: Part II**
>
> **C. “In the case you need to know the horizon before hand to tune the parameters, then we do require careful design of the parameters, and the point mentioned in the paper for the advantage of single time scale algorithm is no longer valid”.** In our theorem statement, we let $\alpha_k=\frac{\bar\alpha}{\sqrt{K}}$. To avoid the dependence on total iteration number $K$ when choosing the step size, we can replace $K$ with the current iteration $k$ without affecting the sample complexity. We use $K$ here only for simplifying the proof, as this type of convergence result is quite standard for finite-time analysis [Ghadimi et al., 2013], [Chen et al., 2021], [Zeng et al., 2022]. We have added the analysis under the diminishing step size at the end of Appendix D.1. We now present the basic idea for such an extension. Divide both sides of equation (94) by $\alpha_k$, rearrange terms and take summation, we can get:
> $$\frac{1}{K}\sum_{k=1}^{K}\sum_{i=1}^{N}||\nabla_{\theta_i}J(\theta_k)||^2 \leq \mathcal{O}(\sum_{k=1}^{K}\frac{1}{\alpha_k}(\mathbb{E}[\mathbb{V}_k]- \mathbb{E}[\mathbb V _k+ {}_1])) +\mathcal{O}(\frac{1}{K} \sum_k \alpha_k+ \beta_k + \eta_k)+ \mathcal{O}(\varepsilon_s {}_p + \varepsilon_a {}_p {}_p) $$
>
> When using diminishing step size, we have $\mathcal{O}(\alpha_k) = \mathcal{O}(\beta_k) = \mathcal{O}(\eta_k) = \mathcal{O}(1/\sqrt{k})$.
> We can bound the first term of right hand side (RHS) by $\mathcal{O}(\frac{1}{K\alpha_K}) = \mathcal{O}(1/\sqrt{K})$. The second term of the RHS is of order $\mathcal{O}(\log K/\sqrt{K})$ since $\sum_{k=1}^{K} \frac{1}{k} = \mathcal{O}(\log(K))$. In this sense, we recover the iteration complexity of $\mathcal{\tilde O}(1/\sqrt{T})$ in Theorem 1, or equivalently, the sample complexity of $\mathcal{\tilde O}(\varepsilon^{-2})$. The
> analysis applies to Theorem 2 by following exactly the same arguments. We have adjusted the $K$ to $k$ in the statement of Theorem 1 and 2, and provided the detailed derivation at the end of Appendix D.1; see the latest manuscript.
>
> We hope that our response is satisfactory to you. Once again, thank you so much for your reply. If there is any further concern, please let us know.
>
> **References**
>
> [Wu et al., 2020] A finite-time analysis of two time-scale actor-critic methods.
>
> [Shen et al., 2020] Asynchronous advantage actor critic: Non-asymptotic analysis and linear speedup.
>
> [Xu et al., 2020] Improving sample complexity bounds for (natural) actor-critic algorithms.
>
> [Qiu et al., 2019] On the finite-time convergence of actor-critic algorithm.
>
> [Chen et al., 2021] Tighter analysis of alternating stochastic gradient methods for bilevel problems.
>
> [Chen et al., 2022] Sample and communication-efficient decentralized Actor-Critic algorithms with finite-time analysis.
>
> [Zeng et al., 2022] Learning to coordinate in multi-agent systems: A coordinated actor-critic algorithm and finite-time guarantees.
>
> [Hairi et al., 2021] Finite-time convergence and sample complexity of multi-agent actor-critic reinforcement learning with average reward.
>
> [Ghadimi et al., 2013] Stochastic first-and zeroth-order methods for nonconvex stochastic programming.

---

> ### Author Response · Authors · 2022-12-19
> **Thank you for your feedback**
>
> Dear reviewer VYyQ,
>
> Thank you so much for your careful reading of our response and your feedback. We are more than happy to know that your concerns have been addressed. Should you have any other remarks, please feel free to let us know.
>
> Best,
>
> Authors.

---

### Review · Reviewer_tG2W · 2022-11-21

**Summary Of Contributions:**

This paper studies decentralized actor-critic algorithm which uses the same learning rate for actor and critic. By using linear approximation for value and reward estimation, they obtained $1/\epsilon^2$ sample complexity guarantee.



**Audience:**

Yes

**Broader Impact Concerns:**

I don't foresee any such concerns.


=============

Post rebuttal: I appreciate the authors' detailed response to my comments. After reading the author response and other reviewers' comments, I agree that there is indeed something new in this work compared to the existing ones. Nonetheless, I am still quite suspicious about the overall novelty of the analysis, and feel (just my own research taste) this single-timescale feature is a rather minor point due to its inconsistency with common empirical RL. As a result, my overall recommendation is borderline reject.

**Claims And Evidence:**

No

**Requested Changes:**

I think this submission is just another A+B paper which doesn't provide much useful information to the community. And I don't think any revision could change such nature.

**Strengths And Weaknesses:**

I do not think this paper provides convincing motivation for studying single-time scale actor-critic algorithm. In practice, people often train actor and critic for a different number of steps per update, even if they use the same-scale learning rate. Therefore, I don't think it is useful to study the setting where the actor and critic have the same-scale learning rate and the same number of updates per inner loop.

The proof techniques are quite standard and well-known. The analysis simply blends the existing ideas for distributed optimization and policy optimization, under the assumption that the Markov chain is irreducible and aperiodic for any policy and every policy has a lower bounded probability to pick any action at any state.

There is also a considerable number of typos in the submission, e.g., in Theorem 1 function $F$ is used without defining it.

---

> ### Author Response · Authors · 2022-12-09
> **Author's response**
>
> Dear reviewer tG2W,
>
> Thanks for your comment. We now address your concerns in a point-by-point manner below. Major changes in the manuscript are highlighted in blue.
>
> **A. “In practice, people often train actor and critic for a different number of steps per update, even if they use the same-scale learning rate. Therefore, I don't think it is useful to study the setting where the actor and critic have the same-scale learning rate and the same number of updates per inner loop”.** We have to clarify that our single-timescale framework allows critic and actor to update any constant number of steps at each iteration while maintains the same optimal sample complexity. For brevity, we let actor and critic update once at each iteration. This is because when actor updates $C_a$ steps at each iteration, $||\theta_{k+1} - \theta_k||=\tilde{\mathcal{O}}(C_a\alpha_k)$ by the bounded gradient property and Cauchy-Schwartz inequality. Hence, $||\theta_{k+1} - \theta_k||$ will remain the order $\tilde{\mathcal{O}}(\alpha_k)$. Thus, we can recover the actor’s error bound of one step update scheme. When critic updates $C_c>1$ steps per iteration, the expected temporal difference error will decrease. Therefore, the critic error bounds still apply. We have provided more details on analyzing the arbitrary constant steps update scheme in the latest revised manuscript; see the paragraph above Section 4.5.
>
> As mentioned by the reviewer, the single-timescale update rule with arbitrary constant number of updates for actor and critic matches the practical implementation of AC algorithm, yet the corresponding convergence is nearly unexplored. Therefore, we believe it is meaningful to study its finite time complexity bound as this may provide useful insights on the success of AC’s practical performance, and hence we believe that our study is well-motivated. We refer to Section 1 of our manuscript for more motivational statements.
>
> **B. “The proof techniques are quite standard and well-known. The analysis simply blends the existing ideas for distributed optimization and policy optimization, under the assumption that the Markov chain is irreducible and aperiodic for any policy and every policy has a lower bounded probability to pick any action at any state".** Let us clarify the analysis of AC algorithm is different from the traditional policy optimization algorithm, policy gradient. The former has an alternating updating scheme, which is the main difficulty. Our main technical challenge lies in analyzing the descent of actor’s gradient norm under the single-timescale update, in which the ciritc only updates for constant steps. This is different from the existing double-loop based analysis that solves the policy evaluation problem up to $\varepsilon$- accuracy. Hence, to ensure convergence of single-timescale scheme, one has to bound the error of policy evaluation (i.e., critic’s error) with constant number of updates as the same order as the actor's gradient norm , which is naturally more challenging (this error is automatically $\varepsilon$ in double-loop scheme). Such a difficulty is the reason for the lack of formal theoretical understanding of single-timescale AC in the literature. To this end, one of our key achievements is to establish the smoothness of the optimal critic variable with respect to policy parameter (see Lemma 14 in our manuscript), so that we can derive the above mentioned approximate descent property of the error of policy evaluation under the single-timescale update. As a consequence, we provide the first finite-time complexity results for single-timescale AC under standard assumptions used in [Chen et al., 2022]. Hence, the contribution of formally establishing smoothness is essential to the analysis and is non-trivial. Moreover, to analyze the single-timescale AC under the fully-decentralized setting , we construct a Lyapunov function which differs from existing literatures; see our reply to reviewer Lyfg for more details. In summary, we would argue that our techniques and theoretical results are nontrivial.
>
> **C. Typos in the manuscript.** Thank you for your careful review, we have corrected the typo; see our latest revised manuscript.
>
> We hope that these revisions and responses are satisfactory to you. Once again, we truly appreciate the very detailed and constructive feedback, which has helped us greatly in improving the quality of the manuscript.
>
> **References**
>
> [Chen et al., 2022] Sample and communication-efficient decentralized Actor-Critic algorithms with finite-time analysis.

---

> ### Author Response · Authors · 2022-12-21
> **Looking forward to your feedback (if any)**
>
> Dear reviewer tG2W,
>
> We hope that our response clarifies our work's contribution and addresses most of your concerns. In particular, we show that our analysis framework naturally incorporates the scheme where actor and critic are updated with different steps for each iteration (last paragraph of page 10 in the latest manuscript), which justifies the motivation of our setting. We also highlighted the difficulty of analyzing the *single-timescale* AC using existing analysis frameworks, and provided more details on the non-trivial part of our analysis (the last paragraph of page 2 in the latest manuscript).
>
> Since the deadline of the discussion phase is approaching, we would highly appreciate to receive feedback from you. If you have any further questions or remarks, please let us know so that we will have enough time to address your concerns. We will be more than happy to provide additional clarifications and details.
>
> Best,
>
> Authors.

---

> ### Author Response · Authors · 2022-12-26
> **Request for your feedback**
>
> Dear reviewer tG2W,
>
> Since the deadline of the open-discussion phase is within two days, we eargely look forward to your feedback on our response. If you have any further concerns, please let us know. We would be more than happy to provide more clarifications.
>
> Best,
>
> Authors.

---

> ### Author Response · Authors · 2023-01-12
> **Further clarification to follow-up comments**
>
> Dear reviewer tG2W,
>
> Thank you for your careful reading of our response and your comments. We would like to provide more references in supporting that our *single-timescale* setting is widely utilized in practice.
>
> In fact, the previous work [Fu et al., 2021] have pointed out that most practical implementations of AC algorithms use single-timescale update (the second paragraph on the page 2):
>
> > "most practical implementations of actor-critic are under the single-timescale setting, where the actor and critic are simultaneously updated, and particularly, the actor is updated without the critic reaching an approximate solution to the policy evaluation sub-problem."
>
> However, the work [Fu et al., 2021] does not analyze the practical single-timescale setting. Instead, it is based on an algorithm where the critic optimization step is formulated as a least-square temporal difference (LSTD) at each iteration, where they need to sample the transition tuples for $O(\varepsilon^{-1})$ times to form the data matrix in the LSTD problem. We refer to the second paragraph on page 2 in our revised manuscript for more discussions on [Fu et al., 2021]. By contrast, we formally establish the first sample complexity results under the practical single-timescale AC setting --- where the actor and critic are alternatively updated for any different constant number of steps at each iteration --- via deriving the smoothness property of the optimal critic's variable. Therefore, we would like to argue that our setting is of practical interest and that our theoretical results are non-trivial.
>
> We hope that this further clarification justifies thoroughly the motivation of our study.
>
> Best,
>
> Authors.
>
> **References**
>
> [Fu et al., 2021] Single-timescale Actor-Critic provably finds globally optimal policy, International Conference on Learning Representation (ICLR) 2021.

---

### Review · Reviewer_LYfg · 2022-11-29

**Summary Of Contributions:**

This paper studies the actor-critic methods in fully decentralized MARL. Compared with existing works, this work makes main contributions in the following sense:

A. Single-time algorithm instead of double-loop or two-timescale fashion
B. Considers the more practical markovian sampling and i.i.d sampling

It maintains the state-of-the-art sample complexity O(\frac{1}{\epsilon^2}) and demonstrates the effectiveness of the algorithm through experiments in the grounded communication environment, showing improved sample and communication efficiency.

**Audience:**

No

**Broader Impact Concerns:**

No.

**Claims And Evidence:**

Yes

**Requested Changes:**

In order to highlight the contributions of this paper, I suggest the authors discuss the main technical difficulties for (1) extending the algorithm for single-agent RL to fully decentralized MARL (2) using markov sampling. Discussing how the standard techniques for (1) and (2) will or will not be different when using a single-timescale algorithm can make the contribution of this paper clearer.

**Post rebuttal**: I would like to thank the author(s) for providing the detailed rebuttal, and I think they generally make sense. I have also checked other reviewers' comments, and still have a bit of concern about the technical novelty and significance of the work. I would suggest a borderline acceptance post-rebuttal.

**Strengths And Weaknesses:**

Strength:

To my best knowledge, it seems the first paper achieving the sample complexity of O(\frac{1}{\epsilon^2}) using a single-timescale algorithm with either iid or Markov sampling in the fully decentralized cooperative MARL.

Weakness:

It seems the proof techniques are not very new. From my understanding, the key analysis is from [1], and the main non-trivial contribution compared with [1] is to verify that the Jacobian of the stationary distribution is indeed Lipschitz, which is the assumption 10 (ii) and not verified in [1]. Other than that, the algorithmic techniques to extend the algorithm for single-agent [1] to fully decentralized cooperative MARL such as consensus update and reward estimator/noisy rewards seems common in existing literature [2, 3].


[1]. Chen, Tianyi, Yuejiao Sun, and Wotao Yin. "Tighter analysis of alternating stochastic gradient method for stochastic nested problems." arXiv preprint arXiv:2106.13781 (2021)
[2] Zhang, Kaiqing, Zhuoran Yang, Han Liu, Tong Zhang, and Tamer Basar. "Fully decentralized multi-agent reinforcement learning with networked agents." In International Conference on Machine Learning, pp. 5872-5881. PMLR, 2018.
[3] Chen, Ziyi, Yi Zhou, Rong-Rong Chen, and Shaofeng Zou. "Sample and communication-efficient decentralized actor-critic algorithms with finite-time analysis." In International Conference on Machine Learning, pp. 3794-3834. PMLR, 2022.

---

> ### Author Response · Authors · 2022-12-09
> **Author's response**
>
> Dear reviewer LYfg,
>
> Thank you for your acknowledgement of our contribution. We now address your concerns on our proof techniques. Major changes in the manuscript are highlighted in blue.
>
> **A. On the technical contribution of our analysis.** As highlighted by the reviewer, one of our main technical contributions is to formally establish the smoothness of the stationary distribution with respect to the policy parameter, which has not been justified by previous works. Rely on this smoothness property, we are able to characterize the sufficient descent of critic’s error under the single-timescale update. As a consequence, we provide the first finite-time complexity results for single-timescale AC under standard assumptions for analyzing AC [Chen et al., 2022]. Hence, this contribution is essential to the analysis and is non-trivial.
>
> In addition, analyzing the single-timescale AC under the fully-decentralized setting requires constructing a Lyapunov function, which differs from the existing works. More specifically, in order to analyze the approximate descent of gradient norm for each iteration, [Chen et al., 2021] constructs the Lyapunov function as $\mathbb{V}_k:= -J(\theta_k) + ||\omega_k - \omega^*(\theta_k)||^2$, while our Lyapunov function is $\mathbb{V}_k:= -J(\theta_k) + ||\bar{\omega}_k - \omega^*(\theta_k)||^2 + ||\bar{\lambda}_k - \lambda^*(\theta_k)||^2$. In comparison with [Chen et al., 2021], we have to analyze the convergence of averaged critic’s error $||\bar{\omega}_k - \omega^*(\theta_k)||^2$ under the decentralized setting, which requires additional treatment for the consensus error. Moreover, to cancel the gradient bias caused by the reward estimation, we plug the averaged reward estimation error $||\bar{\lambda}_k - \lambda^*(\theta_k)||^2$ into the Lyapunov function, which does not appear in [Chen et al., 2021], [Chen et al., 2022], and [Zhang et al., 2018]’s analyses.
>
> As mentioned by the reviewer, the consensus error’s analysis resort to the technique of standard distributed optimization, and the analysis of Markovian sampling error borrows the idea of constructing an auxiliary Markov chain. Hence, we did not list these analyses as technical contributions in our manuscript. Our contribution lies in constructing an analysis framework for the fully-decentralized single-timescale AC so that these existing techniques can be readily  applied. We have added more discussions in the latest manuscript; see the last paragraph of page 2.
>
> We hope that these revisions and responses are satisfactory to you. Once again, we truly appreciate the very detailed and constructive feedback, which has helped us greatly in improving the quality of the manuscript.
>
> **References**
>
> [Chen et al., 2021] Tighter analysis of alternating stochastic gradient methods for bilevel problems.
>
> [Chen et al., 2022] Sample and communication-efficient decentralized Actor-Critic algorithms with finite-time analysis.
>
> [Zhang et al., 2018] Fully decentralized multi-agent reinforcement learning with networked agents.

---

> ### Author Response · Authors · 2022-12-21
> **Looking forward to your feedback (if any)**
>
> Dear reviewer LYfg,
>
> We hope that our response clarifies our work's contribution and addresses most of your concerns. Specifically, we provided more discussions on the significance for verifying the smoothness property of stationary distribution. We also highlighted our design of Lyapunov function, which differs from existing works and is necessary for analyzing *single-timescale* AC under *decentralized* setting (the last paragraph of page 2 in the revised manuscript).
>
> Since the deadline of the discussion phase is approaching, we would highly appreciate to receive feedback from you. If you have any further questions or remarks, please let us know so that we will have enough time to address your concerns. We will be more than happy to provide additional clarifications and details.
>
> Best,
>
> Authors.

---

> ### Author Response · Authors · 2022-12-29
> **Response to follow-up comments**
>
> Dear reviewer LYfg,
>
> Thank you so much for your careful reading of our response and your acknowledgement of our work. To address your remaining concerns, we would like to provide more clarifications on our technical contributions.
>
> As highlighted before, verifying the Lipschitz property of the stationary distribution allows us to establish the smoothness property of the optimal critic variable. Consequently, we can resort to relatively standard analysis technique of alternating optimization. In fact, in [Chen et al., 2021]'s paper, the revealing of a smiliar smoothness property for the specific bilevel problem is listed as one main contribution:
>
> > "By leveraging the hidden smoothness, we present a tighter analysis of ALSET for the stochastic bilevel problems."
>
> The significance for the smoothness property is also justified by the review of [Chen et al., 2021]'s submission; see https://openreview.net/forum?id=OItvP2-i9j:
>
> > "The smoothness of the gradients proved in Lemma 2 opens the way to the improved rate."
>
> Thus, we believe establishing smoothness property for the specific reinforcement learning problem is non-trivial. When analyzing the AC algorithm, [Chen et al., 2021] does not justify the Lipschitz property of the stationary distribution when proving the smoothness property. Our work instead derives it, which has not been shown in the literature. This allows to fully formally establish the first finite-time convergene results for single-timescale AC. In addition, we believe that this smoothness property we derived may serve as a blueprint for the analysis of other single-timescale AC variants, such as single-timescale natural AC, etc.
>
> Additionally, we constructs a new Lyapunov function as discussed before in order to deal with multi-agent setting. Therefore, we would argue that our technical contributions are not minor compared to the existing literature.
>
> **References**
>
> [Chen et al., 2021] Tighter analysis of alternating stochastic gradient methods for bilevel problems.

---

### Decision · Action_Editors · 2023-01-18

**Recommendation:** Accept as is

**Comment:**

The paper studies finite-time convergence of decentralized actor-critic in multi-agent reinforcement learning, which uses single-timescale updates. This kind of update is simpler than two-timescale updates, but has limited theoretical understanding in its convergence behavior. The work is solid, although there are concerns with technical novelty. The author response clarified some of the issues. Furthermore, given popularity and simplicity of single-timescale updates, the results are interesting and relevant, and I’m happy to recommend acceptance.

Minor suggestion for the final version: There is no need to submit the main paper and appendix in separate files. They can be merged into a single file.

**Audience:**

Given popularity and simplicity of single-timescale updates, the results are interesting to TMLR audience.

**Claims And Evidence:**

The claims are supported by proofs and numerical experiments in a simulated domain.